# Decentralized Langevin Dynamics for Bayesian Learning

Anjaly Parayil[1], He Bai[2], Jemin George[1], and Prudhvi Gurram[1,3]

[1]CCDC Army Research Laboratory, Adelphi, MD 20783, USA
[2]Oklahoma State University, Stillwater, OK 74078, USA
[3]Booz Allen Hamilton, McLean, VA 22102, USA
[1]panjaly05@gmail.com, jemin.george.civ@mail.mil, pkgurram@ieee.org
[2]he.bai@okstate.edu

## Abstract

Motivated by decentralized approaches to machine learning, we propose a collaborative Bayesian learning algorithm taking the form of decentralized Langevin dynamics in a non-convex setting. Our analysis show that the initial KL-divergence between the Markov Chain and the target posterior distribution is exponentially decreasing while the error contributions to the overall KL-divergence from the additive noise is decreasing in polynomial time. We further show that the polynomial-term experiences speed-up with number of agents and provide sufficient conditions on the time-varying step-sizes to guarantee convergence to the desired distribution. The performance of the proposed algorithm is evaluated on a wide variety of machine learning tasks. The empirical results show that the performance of individual agents with locally available data is on par with the centralized setting with considerable improvement in the convergence rate.

## 1 Introduction

With the recent advances in computational infrastructure, there has been an increase in the use of larger machine learning models with millions of parameters. Even though there is a parallel increase in the size of training datasets for these models, there is a significant disparity between the amount of existing data and the data required to train the large models to avoid overfitting and provide good generalization performance. Such models trained in point estimate settings such as Maximum A Posteriori (MAP) neglect any associated epistemic uncertainties and make overconfident predictions. Bayesian learning framework provides a principled way to avoid over-fitting and model uncertainties by estimating the posterior distribution of the model parameters. However, analytical solutions of exact posterior or sampling from the exact posterior is often impossible due to the intractability of the evidence. Therefore, one needs to resort to approximate Bayesian methods such as Markov Chain Monte Carlo (MCMC) sampling techniques. To this effect, we focus on a specific class of MCMC methods, called Langevin dynamics to sample from the posterior distribution and perform Bayesian machine learning.

Langevin dynamics derives motivation from diffusion approximations and uses the information of a target density to efficiently explore the posterior distribution over parameters of interest [1]. Langevin dynamics, in essence, is the steepest descent flow of the relative entropy functional or the KL-divergence with respect to the Wasserstein metric [2–4]. Just as the gradient flow converges exponentially fast under a gradient-domination condition, Langevin dynamics converges exponentially fast to the stationary target distribution if the relative entropy functional satisfies the log-Sobolev inequality [3–5].

The Unadjusted Langevin Algorithm (ULA) is a popular inexact first-order discretized implementation of the Langevin dynamics without an acceptance/rejection criteria. Analysis of convergence properties of the ULA and other Langevin approximations has been a topic of active research over past several years [6–12]. Reference [3] shows that a bias exists in the ULA for any arbitrarily small (fixed) step size, even for a Gaussian target distribution. Controlling the bias and exponential convergence of KL divergence for strongly log-concave smooth target distributions using ULA is discussed in [3, 6, 7, 9–11]. Non-asymptotic bounds on variation error of the Langevin approximations for smooth log-concave target distributions have been established by [6] and [8]. Assuming a Lipschitz continuous Hessian, [6] introduces a modified version of the Langevin algorithm requiring fewer iterations to achieve the same precision level. Tight relations between the Langevin Monte Carlo for sampling and the gradient descent for optimization for (strongly) log-concave target distributions are presented in [7]. Similarly, using the notion of gradient flows over probability space and KL-divergence, [9] analyzes the non-asymptotic convergence of discretized Langevin diffusion. These results were improved and extended with particular emphasis on scalability of the approach with dimension, smoothness, and curvature of the function of interest in [10–12].

Compared to log concave ULA settings where local properties replicate the global behavior and optimal values are attained in a single pass, non-convex objective functions naturally require multiple passes through training data. Analysis of ULA in such cases often requires assuming that the negative log of the target distribution satisfies some dissipative property [13–17], contractivity condition [18], or limiting the non-convexity to a local region [19, 20]. In particular, [13] makes the first attempt in analyzing non-asymptotic convergence in a nonconvex setting and shows SGLD tracks continuous Langevin diffusion in quadratic Wasserstein distance for empirical risk minimization. Recent work [14, 19, 21] reports computational efficiency of sampling algorithm to optimization methods in the nonconvex setting. The approach is extended to relaxed dissipativity conditions, to evaluate dependent data streams and provides sharper convergence estimates uniform in the number of iterations in [15, 16]. More recently, it is shown that the convergence is polynomial in terms of dimension and error tolerance [17, 18, 20].

Besides the ULA, higher-order Langevin diffusion for accelerated sampling algorithms are presented in [22, 23]. Analysis of "leapfrog" implementation of Hamiltonian Monte-Carlo (HMC) for strongly log-concave target distributions is presented in [24] and [25]. Following the introduction of a stochastic gradient-based Langevin approach for Bayesian inference in [26], stochastic gradient based Langevin diffusion and other HMC schemes are presented in [27–31].

**Related Work:** The approaches discussed so far assume a centralized entity to process large datasets. However, communication challenges associated with transferring large amounts of data to a central location and the associated privacy issues motivate a decentralized implementation over its centralized counterparts [32, 33]. Master-slave architecture for distributed MCMC via moment sharing is presented [34]. Data-parallel MCMC algorithms for large-scale Bayesian posterior sampling are presented in [35–38]. These parallel MCMC schemes [39–42] are not applicable in decentralized setting since they require a central node to aggregate and combine the samples from individual chains generated by the computing nodes in a final post-processing step to generate an approximation of the true posterior. Recently, decentralized Stochastic gradient Langevin dynamics (SGLD) and stochastic gradient Hamiltonian Monte Carlo (SGHMC) methods for strongly log-concave posterior distribution are presented in [43].

**Contribution:** In this paper, we draw on the recent ULA literature and develop a decentralized learning algorithm based on the centralized ULA in a nonconvex setting. We consider the problem of collaboratively inferencing the global posterior of a parameter of interest based on independent data sets distributed among a network of $n$ agents. The communication topology between the agents can be any undirected connected graph, including the master-slave topology as a special case. We propose a decentralized ULA (D-ULA) that incorporates an average consensus process into the ULA with time-varying step-sizes. In this algorithm, each agent shares its current Markov Chain sample with neighboring agents at each time step. We show that the resulting distribution of the averaged sample converges to the true posterior asymptotically. We provide theoretical analysis of the convergence rate and step-size conditions to achieve speed up of convergence with respect to the number of agents. Empirical results show that the performance of our proposed algorithm is on par with centralized ULA with considerable improvement in the convergence rate, for three different machine learning tasks.

**Notation:** Let $\mathbb{R}^{n \times m}$ denote the set of $n \times m$ real matrices. For a vector $\boldsymbol{\phi}$, $\phi_i$ is the $i-$th entry of $\boldsymbol{\phi}$. An $n \times n$ identity matrix is denoted as $I_n$ and $\mathbf{1}_n$ denotes an $n$-dimensional vector of all ones. For $p \in [1, \infty]$, the $p$-norm of a vector $\mathbf{x}$ is denoted as $\|\mathbf{x}\|_p$. For matrices $A \in \mathbb{R}^{m \times n}$ and $B \in \mathbb{R}^{p \times q}$, $A \otimes B \in \mathbb{R}^{mp \times nq}$ denotes their Kronecker product. For a graph $\mathcal{G}(\mathcal{V}, \mathcal{E})$ of order $n$, $\mathcal{V} \triangleq \{v_1, \ldots, v_n\}$ represents the agents or nodes and the communication links between the agents are represented as $\mathcal{E} \triangleq \{e_1, \ldots, e_\ell\} \subseteq \mathcal{V} \times \mathcal{V}$. Let $\mathcal{A} = [a_{i,j}] \in \mathbb{R}^{n \times n}$ be the *adjacency matrix* with entries of $a_{i,j} = 1$ if $(v_i, v_j) \in \mathcal{E}$ and zero otherwise. Define $\Delta = \text{diag}(\mathcal{A}\mathbf{1}_n)$ as the in-degree matrix and $\mathcal{L} = \Delta - \mathcal{A}$ as the graph *Laplacian*.

## 2 Problem formulation

Consider a connected network of $n$ agents, each with a randomly distributed set of $m_i$ data items, $\boldsymbol{X}_i = \left\{ \boldsymbol{x}_i^j \right\}_{j=1}^{j=m_i}, \forall\, i = 1, \ldots, n$. Here $\boldsymbol{x}_i^j \in \mathbb{R}^{d_x}$ is the $j$-th data element in a set of $m_i$ data items available to the $i$-th agent. Let $\boldsymbol{w} \in \mathbb{R}^{d_w}$ be the parameter vector associated with the model and $p(\boldsymbol{w})$ is the prior associated with the model parameters. The *global posterior* distribution of $\boldsymbol{w}$ given the $n$ independent data sets distributed among the agents can be expressed as

$$p(\boldsymbol{w}|\boldsymbol{X}_1, \ldots, \boldsymbol{X}_n) \propto p(\boldsymbol{w}) \prod_{i=1}^{n} p(\boldsymbol{X}_i|\boldsymbol{w}) = \prod_{i=1}^{n} \underbrace{p(\boldsymbol{X}_i|\boldsymbol{w})p(\boldsymbol{w})^{\frac{1}{n}}}_{\text{local posterior}}. \tag{1}$$

In the optimization literature, the prior, $p(\boldsymbol{w})$, regularizes the parameter and the likelihood, $p(\boldsymbol{X}_i|\boldsymbol{w})$, represents the local cost function available to each agent. Here, the set of $n$ independent data sets are distributed among $n$ agents each with a size of $m_i$, $i = 1, \ldots, n$. At the risk of abusing the notation, define $p(\boldsymbol{w}|\boldsymbol{X}_i)$ as the *local posterior* distribution. Thus the global posterior can be written as the product of local posteriors as

$$p(\boldsymbol{w}|\boldsymbol{X}_1, \ldots, \boldsymbol{X}_n) \propto \prod_{i=1}^{n} p(\boldsymbol{w}|\boldsymbol{X}_i). \tag{2}$$

The main issue with point estimates obtained from optimization schemes like maximum likelihood and maximum a posteriori estimation is that they fail to capture the parameter uncertainty and they can potentially over-fit the data. This paper is aimed at developing a method for collaborative Bayesian learning from large scale datasets distributed among a networked set of agents as a solution to the numerous issues associated with the point estimation schemes. In particular, we present a decentralized version of the unadjusted Langevin algorithm to distributedly obtain samples from the global posterior $p(\boldsymbol{w}|\boldsymbol{X}_1, \ldots, \boldsymbol{X}_n)$. For the ease of notation, we use $\mathbf{X}$ to denote the entire data set. Thus the global posterior can be written as $p(\boldsymbol{w}|\mathbf{X})$.

## 3 Decentralized unadjusted Langevin algorithm

To efficiently explore the global posterior $p(\boldsymbol{w}|\boldsymbol{X})$, we first rewrite the target distribution in terms of an energy function $U$ as follows [1, 44, 45]:

$$p(\boldsymbol{w}|\mathbf{X}) \propto \exp(-U(\boldsymbol{w})), \tag{3}$$

where $U$ is the analogue of potential energy given by

$$U(\boldsymbol{w}) \propto -\log p(\boldsymbol{w}|\mathbf{X}). \tag{4}$$

The Langevin algorithm is a well known family of gradient based Monte Carlo sampling algorithms. The sample obtained using Unadjusted Langevin Algorithm (ULA) at a given time instant $k$ is given by [19]

$$\boldsymbol{w}(k+1) = \boldsymbol{w}(k) - \alpha_k \nabla U(\boldsymbol{w}(k)) + \sqrt{2\alpha_k}\boldsymbol{v}(k) \tag{5}$$

where $\alpha_k$ is the algorithm step-size, $\boldsymbol{w}(k)$ represents the sample obtained at the $k$-th time instant and $\boldsymbol{v}(k)$ is a $d_w$-dimensional, independent, zero-mean, unit variance, Gaussian sequence, i.e., $\boldsymbol{v}(k) \sim \mathcal{N}(\mathbf{0}, I_{d_w}), \forall k \geq 0$. Now substituting (4) yields

$$\boldsymbol{w}(k+1) = \boldsymbol{w}(k) + \alpha_k \nabla \log p(\boldsymbol{w}(k)|\mathbf{X}) + \sqrt{2\alpha_k}\boldsymbol{v}(k). \tag{6}$$

Substituting (1) yields

$$\boldsymbol{w}(k+1) = \boldsymbol{w}(k) + \alpha_k \sum_{i=1}^{n} \left( \nabla \log p(\boldsymbol{X}_i|\boldsymbol{w}(k)) + \frac{1}{n} \nabla \log p(\boldsymbol{w}(k)) \right) + \sqrt{2\alpha_k}\boldsymbol{v}(k). \tag{7}$$

The samples obtained using the continuous-time version of the centralized ULA given in (7) have shown to exponentially converge to the target posterior distribution [46] for a certain class of distributions with exponential tails. Convergence properties of the ULA had been widely studied for log-concave target distributions [6,8–12]. Non-asymptotic analysis of centralized ULA without the strong log-concavity assumption on target distribution is presented in [13–20].

However, when the data is distributed among $n$ agents and there is no central agent to pool all the local gradients, exact implementation of the above ULA is difficult, if not impossible. Therefore we propose the following decentralized ULA:

$$
\begin{aligned}
\boldsymbol{w}_i(k+1) = \boldsymbol{w}_i(k) &- \beta_k \sum_{j=1}^{n} a_{i,j} \left( \boldsymbol{w}_i(k) - \boldsymbol{w}_j(k) \right) \\
&+ \alpha_k n \left( \nabla \log p(\boldsymbol{X}_i | \boldsymbol{w}_i(k)) + \frac{1}{n} \nabla \log p(\boldsymbol{w}_i(k)) \right) + \sqrt{2\alpha_k} \boldsymbol{v}_i(k),
\end{aligned}
\tag{8}
$$

where $a_{i,j}$ denotes the entries of the adjacency matrix corresponding to the communication network $\mathcal{G}(\mathcal{V},\mathcal{E})$, $\beta_k$ is the consensus step-size and $\boldsymbol{v}_i(k)$ are $d_w$-dimensional, independent, zero-mean, Gaussian sequence with variance $n$, i.e., $\boldsymbol{v}_i(k) \sim \mathcal{N}(\boldsymbol{0}_{d_w}, nI_{d_w})$, $\forall i \in \mathcal{I}$.

**Remark 1.** *Compared to the parallel MCMC setting, our formulation do not require a central coordinator and each computing nodes reconstruct the approximation to the posterior simply relying on individually available data set and prior information (incorporated as $\nabla \log p(\boldsymbol{X}_i | \boldsymbol{w}_i(k)) + \frac{1}{n} \nabla \log p(\boldsymbol{w}_i(k))$ into the algorithm as shown in (8)) and by interacting with their one-hop neighbors as dictated by the undirected communication graph $\mathcal{G}(\mathcal{V},\mathcal{E})$ (as denoted as $\sum_{j=1}^{n} a_{i,j} \left( \boldsymbol{w}_i(k) - \boldsymbol{w}_j(k) \right)$ in (8)). Here $a_{i,j}$ is the $(i,j)$-th entry of the $n \times n$ adjacency matrix $\mathcal{A}$. $a_{i,j} = 1$ if the $i$-th node can communicate with the $j$-th node and zero otherwise. Similar technique is used in decentralized supervised learning [32, 47, 48].*

Define $\mathbf{w}(k) \triangleq \begin{bmatrix} \boldsymbol{w}_1^\top(k) & \cdots & \boldsymbol{w}_n^\top(k) \end{bmatrix}^\top \in \mathbb{R}^{nd_w}$ and $\mathbf{v}(k) \triangleq \begin{bmatrix} \boldsymbol{v}_1^\top(k) & \cdots & \boldsymbol{v}_n^\top(k) \end{bmatrix}^\top \in \mathbb{R}^{nd_w}$. Now (8) can be written as

$$
\mathbf{w}(k+1) = \mathbf{w}(k) - \beta_k \left( \mathcal{L} \otimes I_{d_w} \right) \mathbf{w}(k) - \alpha_k n \mathbf{g}(\mathbf{w}(k), \mathbf{X}) + \sqrt{2\alpha_k} \mathbf{v}(k), \tag{9}
$$

where $\mathcal{L}$ is the network Laplacian and

$$
\mathbf{g}(\mathbf{w}(k), \mathbf{X}) \triangleq \begin{bmatrix} \mathbf{g}_1 \left( \boldsymbol{w}_1(k), \boldsymbol{X}_1 \right) \\ \vdots \\ \mathbf{g}_n \left( \boldsymbol{w}_n(k), \boldsymbol{X}_n \right) \end{bmatrix} = \begin{bmatrix} \nabla U_1 \left( \boldsymbol{w}_1(k), \boldsymbol{X}_1 \right) \\ \vdots \\ \nabla U_n \left( \boldsymbol{w}_n(k), \boldsymbol{X}_n \right) \end{bmatrix}
$$

where $U_i(\boldsymbol{w}, \boldsymbol{X}_i) = -\log p(\boldsymbol{w}|\boldsymbol{X}_i)$ and $p(\boldsymbol{w}|\boldsymbol{X}_i)$ is the local posterior, given in (1). Define the network weight-matrix $\mathcal{W}_k = (I_n - \beta_k \mathcal{L})$. Thus the proposed decentralized ULA can be written as

$$
\mathbf{w}(k+1) = \left( \mathcal{W}_k \otimes I_{d_w} \right) \mathbf{w}(k) - \alpha_k n \mathbf{g}(\mathbf{w}(k), \mathbf{X}) + \sqrt{2\alpha_k} \mathbf{v}(k). \tag{10}
$$

If we ignore the additive noise term, then the decentralized ULA of (10) can be considered a consensus optimization algorithm aimed at solving the problem, $\min_{\boldsymbol{w}} \boldsymbol{U}(\boldsymbol{w}, \mathbf{X})$, where

$$
\boldsymbol{U}(\boldsymbol{w}, \mathbf{X}) = \sum_{i=1}^{n} U_i(\boldsymbol{w}, \boldsymbol{X}_i). \tag{11}
$$

Denote by $p^*$ the stationary probability distribution corresponding to the global posterior distribution, i.e., the target distribution. It then follows from (3) that

$$
p^*(\cdot) = \exp\left( -\boldsymbol{U}(\cdot, \mathbf{X}) + C \right), \tag{12}
$$

for some positive constant $C$ corresponding to the normalizing constant. Now note that the centralized ULA for generating samples from the target distribution $p^*(\bar{\boldsymbol{w}}^*)$ of (12) is given as [9]

$$
\bar{\boldsymbol{w}}^*(k+1) = \bar{\boldsymbol{w}}^*(k) - \alpha_k \nabla \boldsymbol{U}(\bar{\boldsymbol{w}}^*(k), \mathbf{X}) + \sqrt{2\alpha_k} \bar{\boldsymbol{v}}(k). \tag{13}
$$

The continuous-time limit of (13) can be obtained as the following Stochastic Differential Equation (SDE) known as the Langevin equation [49]:

$$
d\bar{\boldsymbol{w}}^*(t) = -\nabla \boldsymbol{U}(\bar{\boldsymbol{w}}^*(t), \mathbf{X})dt + \sqrt{2}dB_t, \tag{14}
$$

where $B_t$ is a $d_w$-dimensional Brownian motion. The pseudocode of the proposed decentralized ULA is given in Algorithm 1, where $\alpha_k = \frac{a}{(k+1)^{\delta_2}}$ and $\beta_k = \frac{b}{(k+1)^{\delta_1}}$ (see Condition 1 in S1). We refer materials from supplementary sections with the prefix S.

# 4 Main Results

Though our proposed algorithm is built on ULA, analysis of even the centralized ULA (C-ULA) for non-log-concave target distributions requires assuming that the negative log of the target distribution satisfies some dissipative property [13–17], contractivity condition [18], or limiting the non-convexity to a local region [19, 20]. Given analysis of D-ULA is novel/non-trivial compared to the existing non-convex consensus-optimization and non-log-concave ULA literature because: $(i)$ the consensus analysis and the results in Theorem 1 are novel since we use time-varying step-sizes $\alpha_k$ and $\beta_k$ and provide an explicit consensus rate in term of step-size decay rates (see (25)), $(ii)$ compared to existing C-ULA analysis for non-log-concave target distributions, the continuous-time approximation to the D-ULA contains an additional consensus error term $\zeta(\cdot)$ in (21) that complicates the analysis. Requirements on the time-varying step sizes are also not straightforward to obtain as the existing literature is focused on fixed step-sizes.

Analysis of the proposed distributed ULA given in (10) requires that the sequences $\{\alpha_k\}$ and $\{\beta_k\}$ be selected as (see Condition 1 in S1)

$$\alpha_k = \frac{a}{(k+1)^{\delta_2}} \quad \text{and} \quad \beta_k = \frac{b}{(k+1)^{\delta_1}}, \tag{15}$$

where $0 < a$, $0 < b$, $0 \le \delta_1$ and $\frac{1}{2} + \delta_1 < \delta_2 < 1$. Furthermore, we make the following three assumptions (formally stated in S1): $(i)$ the gradients $\nabla U_i$ are Lipschitz continuous with Lipschitz constant $L_i > 0$, $\forall i = 1, \ldots, n$; $(ii)$ the communication network is given as a connected undirected graph; and $(iii)$ there exists a positive constant $\mu_g < \infty$ such that the disagreement on the gradient among the distributed agents, denoted as $\tilde{\mathbf{g}}(\mathbf{w}_k, \mathbf{X})$, satisfies $\mathbb{E}\left[\|\tilde{\mathbf{g}}(\mathbf{w}_k, \mathbf{X})\|_2^2 \,|\, \mathbf{w}_k\right] \le n\mu_g(1+k)^{\delta_2}$ a.s., where $\tilde{\mathbf{g}}(\mathbf{w}_k, \mathbf{X}) = \mathbf{g}(\mathbf{w}_k, \mathbf{X}) - \left(\frac{1}{n}\mathbf{1}_n\mathbf{1}_n^\top \otimes I_{d_w}\right)\mathbf{g}(\mathbf{w}_k, \mathbf{X})$.

From the proposed distributed ULA given in (10), the average dynamics is given as

$$\bar{\boldsymbol{w}}(k+1) = \bar{\boldsymbol{w}}(k) - \alpha_k \sum_{i=1}^{n} \nabla U_i\left(\boldsymbol{w}_i(k), \boldsymbol{X}_i\right) + \sqrt{2\alpha_k}\,\bar{\boldsymbol{v}}(k), \tag{16}$$

where $\bar{\boldsymbol{w}}(k) = \frac{1}{n}\sum_{i=1}^{n} \boldsymbol{w}_i(k)$ and $\bar{\boldsymbol{v}}(k) = \frac{1}{n}\sum_{i=1}^{n} \boldsymbol{v}_i(k)$ is a zero-mean, unit-variance Gaussian random vector. Now adding and subtracting $\alpha_k \sum_{i=1}^{n} \nabla U_i\left(\bar{\boldsymbol{w}}(k), \boldsymbol{X}_i\right) = \alpha_k \nabla \boldsymbol{U}\left(\bar{\boldsymbol{w}}(k), \mathbf{X}\right)$ yields

$$\bar{\boldsymbol{w}}(k+1) = \bar{\boldsymbol{w}}(k) - \alpha_k \nabla \boldsymbol{U}\left(\bar{\boldsymbol{w}}(k), \mathbf{X}\right) - \alpha_k \zeta(\bar{\boldsymbol{w}}(k), \tilde{\mathbf{w}}(k)) + \sqrt{2\alpha_k}\,\bar{\boldsymbol{v}}(k), \tag{17}$$

where $\boldsymbol{U}\left(\bar{\boldsymbol{w}}(k), \mathbf{X}\right)$ is defined in (11), the consensus error $\tilde{\mathbf{w}}(k)$ is defined as $\tilde{\mathbf{w}}(k) = \left(\left(I_n - \frac{1}{n}\mathbf{1}_n\mathbf{1}_n^\top\right) \otimes I_{d_w}\right)\mathbf{w}(k)$ and $\zeta(\bar{\boldsymbol{w}}(k), \tilde{\mathbf{w}}(k))$ is defined as

$$\zeta(\bar{\boldsymbol{w}}(k), \tilde{\mathbf{w}}(k)) = \sum_{i=1}^{n} \left(\nabla U_i\left(\bar{\boldsymbol{w}}(k) + \tilde{\boldsymbol{w}}_i(k), \boldsymbol{X}_i\right) - \nabla U_i\left(\bar{\boldsymbol{w}}(k), \boldsymbol{X}_i\right)\right). \tag{18}$$

For all $k \ge 0$, let $[t_k, \ t_{k+1})$ denote the current time-interval, i.e., $t \in [t_k, \ t_{k+1})$, where $t_k$ is defined as $t_k = \sum_{j=0}^{k-1} \alpha_j$. Here, $t_{k+1} = t_k + \alpha_k$. Define $\tilde{\omega}(t)$ as

$$\tilde{\omega}(t) = \tilde{\mathbf{w}}(k), \quad \forall t \in [t_k, \ t_{k+1}), \quad k \ge 0. \tag{19}$$

Now (17) can be written as

$$\bar{\boldsymbol{w}}(t_{k+1}) = \bar{\boldsymbol{w}}(t_k) - \alpha_k \nabla \boldsymbol{U}\left(\bar{\boldsymbol{w}}(t_k), \mathbf{X}\right) - \alpha_k \zeta(\bar{\boldsymbol{w}}(t_k), \tilde{\mathbf{w}}(t_k)) + \sqrt{2}\left(B_{t_{k+1}} - B_{t_k}\right), \tag{20}$$

where $B_t$ is a $d_w$-dimensional Brownian motion. Thus, for $t_k \le t < t_{k+1}$, the discretized equation of (17) is given by

$$d\bar{\boldsymbol{w}}(t) = -\nabla \boldsymbol{U}\left(\bar{\boldsymbol{w}}(t_k), \mathbf{X}\right) dt + \sqrt{2}dB_t - \zeta(\bar{\boldsymbol{w}}(t_k), \tilde{\omega}(t))dt. \tag{21}$$

Let $\bar{\boldsymbol{w}}(t)$ in (21) admits a probability distribution $p_t(\bar{\boldsymbol{w}})$ for $t_k \le t < t_{k+1}$. Here we aim to show that $p_{t_k}(\bar{\boldsymbol{w}}) \to p^*$ as $k \to \infty$.

## 4.1 Kullback-Leibler (KL) divergence and log-Sobolev inequality

Sampling can be viewed as optimization in the space of measures, where the objective function in the space of measures attains its minimum at the target distribution. Following [3, 4, 19, 20], we use the relative entropy or the KL-divergence of $p_t(\bar{\boldsymbol{w}})$ to the target distribution $p^*$, denoted by $F(p_t(\bar{\boldsymbol{w}}))$, as the objective, i.e.,

$$F(p_t(\bar{\boldsymbol{w}})) = \int p_t(\bar{\boldsymbol{w}}) \log\left(\frac{p_t(\bar{\boldsymbol{w}})}{p^*(\bar{\boldsymbol{w}})}\right) d\bar{\boldsymbol{w}}. \tag{22}$$

---

**Algorithm 1** Decentralized ULA (D-ULA)

---
1: *Initialization* : $\mathbf{w}(0) = \begin{bmatrix} \boldsymbol{w}_1^\top(0) & \dots & \boldsymbol{w}_n^\top(0) \end{bmatrix}^\top$
2: *Input* : $a, b, \delta_1$ and $\delta_2$
3: **for** $k \geq 0$ **do**
4:    **for** $i = 1$ to $n$ **do**
5:       *Sample* $\boldsymbol{v}_i(k) \sim \mathcal{N}(\mathbf{0}, nI_{d_w})$ *& compute* $\mathbf{g}_i(\boldsymbol{w}_i(k), \boldsymbol{X}_i)$
6:       *Compute* $\hat{\boldsymbol{w}}_i(k) = \sum_{j=1}^n a_{i,j}(\boldsymbol{w}_i(k) - \boldsymbol{w}_j(k))$
7:       *Update* $\boldsymbol{w}_i(k+1) = \boldsymbol{w}_i(k) - \beta_k \hat{\boldsymbol{w}}_i(k) - \alpha_k n \, \mathbf{g}_i(\boldsymbol{w}_i(k), \boldsymbol{\xi}_i(k)) + \sqrt{2\alpha_k} \boldsymbol{v}_i(k)$
8:    **end for**
9: **end for**

---

KL-divergence is non-negative and it is minimized at the target distribution, i.e., $F(p_t(\bar{\boldsymbol{w}})) \geq 0$ and $F(p_t(\bar{\boldsymbol{w}})) = 0$ if and only if $p_t = p^*$. The property of $p^*$ that we rely on to show convergence of the proposed algorithm is that it satisfies a log-*Sobolev inequality*. Consider a Sobolev space defined by the weighted norm: $\int g(\bar{\boldsymbol{w}})^2 p^*(\bar{\boldsymbol{w}}) \, d\bar{\boldsymbol{w}}$, where $p^*(\bar{\boldsymbol{w}}) \propto \exp(-U(\bar{\boldsymbol{w}}))$. We say that $p^*(\bar{\boldsymbol{w}})$ satisfies a log-Sobolev inequality if there exists a constant $\rho_U > 0$ such that for any smooth function $g$ satisfying $\int g(\bar{\boldsymbol{w}}) p^*(\bar{\boldsymbol{w}}) \, d\bar{\boldsymbol{w}} = 1$, we have:

$$\int g(\bar{\boldsymbol{w}}) \log g(\bar{\boldsymbol{w}}) p^*(\bar{\boldsymbol{w}}) \, d\bar{\boldsymbol{w}} \leq \frac{1}{2\rho_U} \int \frac{\|\nabla g(\bar{\boldsymbol{w}})\|^2}{g(\bar{\boldsymbol{w}})} p^*(\bar{\boldsymbol{w}}) \, d\bar{\boldsymbol{w}}, \tag{23}$$

where $\rho_U$ is the log-Sobolev constant. Let $g(\bar{\boldsymbol{w}}) = \dfrac{p_t(\bar{\boldsymbol{w}})}{p^*(\bar{\boldsymbol{w}})}$. Thus we have

$$F(p_t(\bar{\boldsymbol{w}})) = \mathbb{E}_{p_t(\bar{\boldsymbol{w}})} \left[ \log \left( \frac{p_t(\bar{\boldsymbol{w}})}{p^*(\bar{\boldsymbol{w}})} \right) \right] \leq \frac{1}{2\rho_U} \mathbb{E}_{p_t(\bar{\boldsymbol{w}})} \left[ \left\| \nabla \log \left( \frac{p_t(\bar{\boldsymbol{w}})}{p^*(\bar{\boldsymbol{w}})} \right) \right\|_2^2 \right]. \tag{24}$$

Now we present our first result, which shows that the average-consensus error $\tilde{\mathbf{w}}_k$ is decreasing at the rate $\mathcal{O}\left( \frac{1}{(k+1)^{\delta_2 - 2\delta_1}} \right)$ (see (S96) for an explicit expression). This implies that the individual samples $\boldsymbol{w}_i(k)$ are converging to $\bar{\boldsymbol{w}}_k$ and this is possible only because of the decaying step-size $\alpha_k$, which also multiplies the additive Gaussian noise.

**Theorem 1.** *Consider the decentralized ULA (D-ULA) given in Algorithm 1 under Assumptions 1-3. Then, for the average-consensus error defined as $\tilde{\mathbf{w}}_k = \left( I_{nd_w} - \frac{1}{n} \mathbf{1}_n \mathbf{1}_n^\top \otimes I_{d_w} \right) \mathbf{w}_k$, there holds:*

$$\mathbb{E}\left[ \|\tilde{\mathbf{w}}_{k+1}\|_2^2 \right] \leq \frac{W_3}{\exp\left( W_1 (k+1)^{1-\delta_1} \right)} + \frac{W_2}{(k+1)^{\delta_2 - 2\delta_1}}, \tag{25}$$

*where $W_1$, $W_2$ and $W_3$ are positive constants defined in* (S87), (S88), (S89) *and* (S90).

Detailed proof of Theorem 1 is given in S3. Now we present our main result which shows that the KL-divergence between $p_t$ and $p^*$ is in fact decreasing.

**Theorem 2.** *Consider the decentralized ULA (D-ULA) given in Algorithm 1 under Assumptions 1-3 with $\alpha_k$, $\beta_k$ given in Condition 1 and $a$ in $\alpha_k$ selected as*

$$a = \frac{1}{n^\gamma} \left( \frac{\rho_U (3\delta_2 - 1)}{25 L^4 \delta_2} \right)^{\frac{1}{3}}, \quad \gamma > 2. \tag{26}$$

*Given that the target distribution satisfies the log-Sobolev inequality (24) with a constant $\rho_U > 0$, and has a bounded second moment, i.e., $\int \|\bar{\boldsymbol{w}}\|_2^2 \, p^*(\bar{\boldsymbol{w}}) \, d\bar{\boldsymbol{w}} \leq c_1$ for some bounded positive constant $c_1$, then for all initial distributions $p_{t_0}(\bar{\boldsymbol{w}})$ satisfying $F(p_{t_0}(\bar{\boldsymbol{w}})) \leq c_2$, we have*

$$F(p_{t_{k+1}}(\bar{\boldsymbol{w}})) \leq \frac{F(p_{t_0}(\bar{\boldsymbol{w}})) + \bar{C}_{F_1}}{\exp\left( \rho_U \sum_{\ell=0}^k \alpha_\ell \right)} + \frac{1}{n^{\gamma-2}} \frac{\bar{C}_{F_2}}{(k+1)^{\delta_2 - 2\delta_1}} + \frac{\bar{C}_{F_3}}{\exp\left( \frac{\rho_U a}{1-\delta_2} (k+1)^{1-\delta_2} \right)} \tag{27}$$

*where the positive constants $\bar{C}_{F_1}$, $\bar{C}_{F_2}$, $\bar{C}_{F_3}$ and associated parameters are defined in* (S224)-(S232).

Proof of Theorem 2 is given in S4. Compared to the existing results, by using a decaying step-size, we are able to remove the constant bias term present in the KL-divergence due to the additive noise. In (27), the constants $\bar{C}_{F_1}$ and $\bar{C}_{F_3}$ are dominated by the consensus-error while the additive noise contributes most to $\bar{C}_{F_2}$. Note that the exponential convergence rate for the initial KL-divergence

$F(p_{t_0}(\bar{\boldsymbol{w}}))$ is similar to what is currently known in the literature [4, 9, 19]. More importantly, the constant bias-term present in the existing results for the KL-divergence between the actual and target distribution, which is absorbed into the constant $\bar{C}_{F_2}$, is decreasing and we do see a speed-up for decay with the number of agents due to the $n^{\gamma-2}$-term. Even though this speed-up increases with $\gamma$, an increasing $\gamma$ in fact decreases the exponential rates of the first and the third terms in (27). Furthermore, constants $\bar{C}_{F_1}, \bar{C}_{F_2}$ are $\bar{C}_{F_3}$ are polynomial in the problem dimension $d_w$.

**Corollary 1.** *For the decentralized ULA (D-ULA) given in Algorithm 1 under the conditions of Theorem 2 and error tolerance $\epsilon \in (0, \ 1)$, there holds*

$$F(p_{t_k}(\bar{\boldsymbol{w}})) \leq \epsilon, \qquad \forall k \geq k^* \tag{28}$$

*where*

$$k^* = \max \left\{ \left( \frac{1-\delta_2}{a\rho_U} \log\left(\frac{2Q_1}{\epsilon}\right) \right)^{\frac{1}{1-\delta_2}}, \ \left( \frac{2Q_2}{\epsilon} \right)^{\frac{1}{\delta_2 - 2\delta_1}} \right\},$$

$Q_1 = \left( F(p_{t_0}(\bar{\boldsymbol{w}})) + \bar{C}_{F_1} \right) \exp\left( \frac{a\rho_U}{1-\delta_2} \right) + \bar{C}_{F_3}$ *and* $Q_2 = \frac{\bar{C}_{F_2}}{n^{\gamma-2}}$.

Corollary 1 follows from Theorem 2 and the proof is given in S5. Corollary 1 provides the minimum number of iterations required to decrease the KL-divergence below a given error-tolerance $\epsilon$.

## 5   Numerical experiments

We apply the proposed algorithm to perform decentralized Bayesian learning for Gaussian mixture modeling, logistic regression, and classification and empirically compare our proposed algorithm to centralized ULA (C-ULA). In all the experiments, we have used a network of five agents in an undirected unweighted ring topology for the decentralized setting. Additional details of all the experiments including step sizes and number of epochs are provided in the Supplementary material (see S6).

### 5.1   Parameter estimation for Gaussian mixture

In this section, we compare the efficiency of D-ULA against the C-ULA for parameter estimation of a multimodal Gaussian mixture with tied means [26]. The Gaussian mixture is given by

$$\theta_1 \sim \mathcal{N}(0, \sigma_1^2); \quad \theta_2 \sim \mathcal{N}(0, \sigma_2^2) \quad \text{and} \quad x_i \sim \frac{1}{2}\mathcal{N}(\theta_1, \sigma_x^2) + \frac{1}{2}\mathcal{N}(\theta_1 + \theta_2, \sigma_x^2)$$

where $\sigma_1^2 = 10, \sigma_2^2 = 1, \sigma_x^2 = 2$ and $\boldsymbol{w} \triangleq [\theta_1, \theta_2]^\top \in \mathbb{R}^2$. For the centralized setting, similar to [26], 100 data samples are drawn from the model with $\theta_1 = 0$ and $\theta_2 = 1$. Available 100 data samples are randomly divided into 5 sets of 20 samples that are made available to each agent in the decentralized network. The posterior distribution of the parameters is bimodal with negatively correlated modes at $\boldsymbol{w} = [0, \ 1]$ and $\boldsymbol{w} = [1, \ -1]$. As shown in Figure 1, the posteriors estimated by D-ULA and C-ULA replicate the true posterior distribution of parameters. Quality of estimated posteriors are compared using an approximate Wasserstein measure [50]. With accurate metric being computational complex, we resort to the Sinkhorn distance and Sinkhorn's algorithm introduced in [50] which in essence defines the cost incurred while mapping the estimated posterior to the true posterior using a transport matrix. The regularization parameter, $\lambda$ in Sinkhorn algorithm is set to 0.1. The experiments are performed for networks of size 1, 5 and 10 and the corresponding Sinkhorn distances, $d_M$, are given by 0.259, 0.251 and 0.244, respectively.

### 5.2   Bayesian logistic regression

We compare the performance of D-ULA and C-ULA for Bayesian inference of logistic regression models using $a9a$ dataset available at the UCI machine learning repository [1]. The dataset contains 32561 observations and 123 parameters. We use a Laplace prior with a scale of 1 on the parameters. Test accuracy averaged over 50 runs for both approaches are shown in Figure 2. During each run, we chose random 80% of data for training and the remaining 20% for testing as in [26]. For D-ULA, we consider networks with 5, 10, and 25 agents. The training data for each case is divided into random sets of equal sizes and made available to agents in the decentralized network. During each run, the same 20% partition is used to test the performance of C-ULA and D-ULA. Test results over ten epochs averaged over 50 runs indicate that the performance of D-ULA is comparable to that of C-ULA. Figures 2b, 2c, 2c, and 2d are zoomed into first 1200 iterations to better show faster

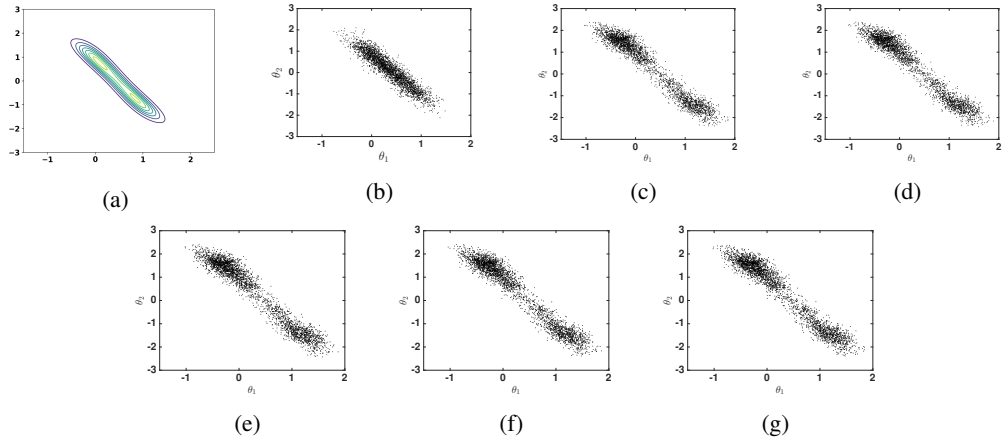

Figure 1: (a) True posterior (b) Estimated posterior by C-ULA (c)-(g) Posteriors estimated by D-ULA

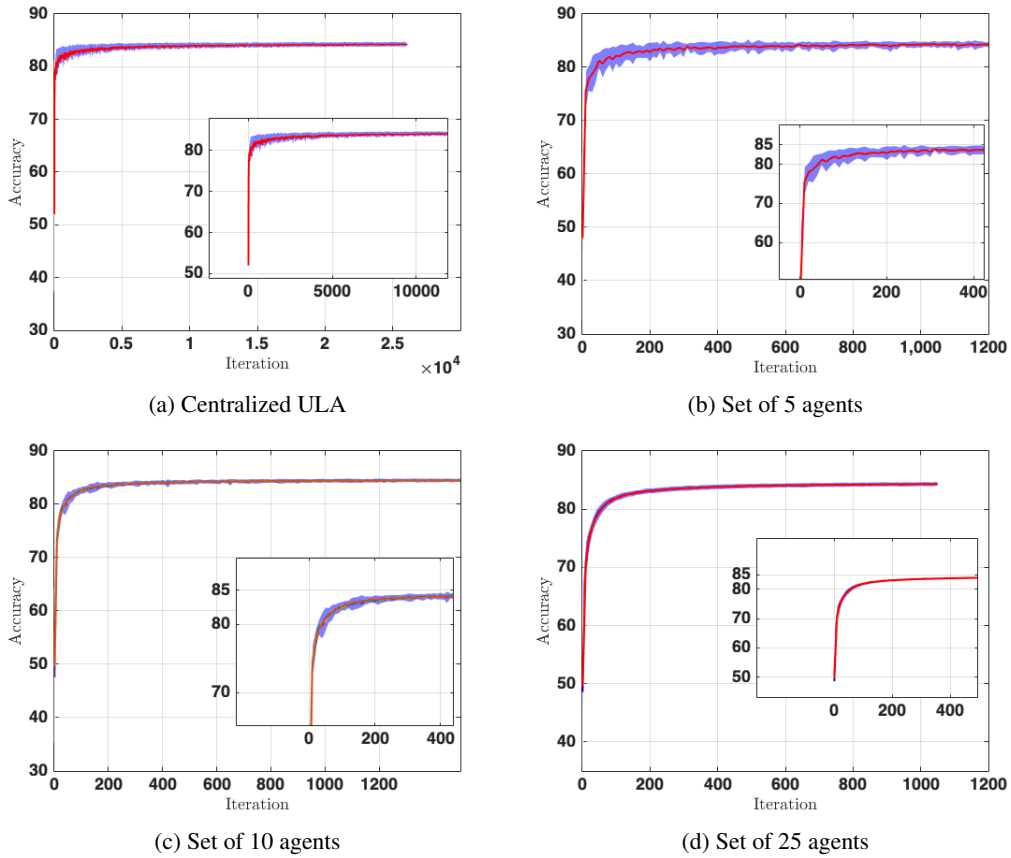

(a) Centralized ULA

(b) Set of 5 agents

(c) Set of 10 agents

(d) Set of 25 agents

Figure 2: Test accuracy averaged over 50 runs

convergence with increase in network size. Corresponding accuracy values for C-ULA and D-ULA networks with agents 5, 10, and 25 are 83.89 %, 84.38%, 84.5637%, and 84.5637%. Insets of Figures 2b, 2c, 2c, and 2d indicates faster convergence of D-ULA compared to C-ULA. Test accuracy of all the agents in D-ULA networks settle to the same accuracy level as shown in the insets. The shaded region in the figures indicates one standard deviation.

Table 1: Probability of predicted labels (mean/standard deviation)

|  |  | SGD | C-ULA | Agent 1 | Agent 2 | Agent 3 | Agent 4 | Agent 5 |
|---|---|---|---|---|---|---|---|---|
| MNIST | Mean | 0.974 | 0.968 | 0.973 | 0.972 | 0.972 | 0.973 | 0.973 |
|  | Std. dev. | 0.078 | 0.086 | 0.079 | 0.08 | 0.08 | 0.08 | 0.08 |
| SVHN | Mean | 0.849 | 0.604 | 0.659 | 0.6588 | 0.653 | 0.663 | 0.651 |
|  | Std. dev | 0.154 | 0.169 | 0.188 | 0.189 | 0.188 | 0.19 | 0.187 |

## 5.3 Bayesian learning for handwritten digit classification and OOD detection

In this section, we present decentralized Bayesian learning as a potential strategy to recognize handwritten digits in images. For this, we use the MNIST data set containing 60000 gray scale images of 10 digits (0-9) for training and 10000 images for testing. Each agent in D-ULA aims to train its own neural network, which is a randomly initialized LeNet-5 [51] with Kaiming uniform prior [52] on the parameters of the network. Each agent has access to 12000 randomly chosen training samples. Test accuracy obtained using stochastic gradient descent (SGD), C-ULA, and 5 agents of D-ULA after 10 epochs are 98.15%, 98.16%, 98.52%, 98.52%, 98.39%, 98.45% and 98.47%, respectively.

Next, we explore the efficacy of the proposed algorithm to detect out-of-distribution (OOD) samples or outliers in the datasets. We train each LeNet-5 neural network on the MNIST training data set and test it on MNIST test data set for normalcy class and Street View House Numbers (SVHN)[2] test data set for OOD data. SVHN data set is similar to MNIST, but with color images of 10 digits (0-9) and extra confusing digits around the central digit of interest. We converted them to gray scale for this experiment. Networks trained on MNIST are expected to give relatively low prediction probabilities for SVHN data samples. Table 1 summarizes the mean and standard deviation of probabilities of predicted labels obtained for all the approaches. Since SGD is a maximum a posteriori point estimate, it fails to recognize out of sample data sets, and gives high prediction probabilities even for OOD SVHN data. One the other hand, C-ULA and D-ULA show an improved performance in detecting OOD SVHN data by giving lower prediction probabilities for SVHN data, but giving high prediction probablities for MNIST test data as seen in Table 1. The plots of probability density of predicted labels corresponding to all the approaches are provided in the Supplementary material.

The decentralized ULA results in Section 5.2 and 5.3 were obtained using a "mini-batch" version of the proposed D-ULA algorithm, where the log-likelihood was obtained from random mini-batches of $X_i$ for agent $i, i = 1, \cdots, n$. Although our theoretical analysis is based on the likelihood from the entire $X_i$, the empirical results in these two sections show that the "mini-batch" D-ULA algorithm is also effective. This is plausible since the additive noise $\sqrt{2\alpha_k}\mathbf{v}_i(k)$ in (9) will dominate the noise in the local posterior term as $k$ increases.

## 6 Conclusion

In this paper, we present a decentralized collaborative approach for a group of agents to sample the posterior distribution of a parameter of interest with locally available data sets. We assume an undirected connected communication topology between the agents. We propose a decentralized unadjusted Langevin algorithm with time-varying step-sizes and establish conditions on the step-sizes for asymptotic convergence to the target distribution. The algorithm also exhibits a guaranteed speed-up in convergence in the number of agents. We conducted three experiments on Gaussian mixtures, logistic regression, and image classification. The experimental results demonstrated that the proposed algorithm offers improved accuracy with enhanced speed of convergence. The results from the last experiment also suggest a potential application of the proposed algorithm for outlier detection.

## Broader Impact

This work presents a basic line of research on reducing computational complexity, enhancing speed of convergence, and addressing potential privacy issues associated with centralized Bayesian learning. Experiments and empirical results cover a broad set of applications including parameter estimation for local non-convex models, logistic regression, image classification and outlier detection. We have

used publicly available datasets, which have no implications on machine learning bias, fairness or ethics. Hence, we believe that this section about potential negative impact of our work on society is not applicable to the proposed work.

## Acknowledgement

This work was supported by the CCDC Army Research Laboratory under Cooperative Agreement W911NF-16-2-0008. The work of the second author was supported in part by the National Science Foundation under Grant No. 1925147. The views and conclusions contained in this document are those of the authors and should not be interpreted as representing the official policies, either expressed or implied, of the Army Research Laboratory or the U.S. Government. The U.S. Government is authorized to reproduce and distribute reprints for Government purposes not withstanding any copyright notation here on.

## Footnotes

[1]http://www.csie.ntu.edu.tw/ cjlin/libsvmtools/ datasets/binary/a9a

[2]http://ufldl.stanford.edu/housenumbers/

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
