[Supplementary Material · NeurIPS_2020_full_07Jan2021.pdf]

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

# Decentralized Langevin Dynamics for Bayesian Learning
## (Supplementary Material)

## S1 Assumptions

We first make the following assumption regarding the $U_i$:

**Assumption 1.** *The gradients[3] $\nabla U_i$ are Lipschitz continuous with Lipschitz constant $L_i > 0$, $\forall i = 1, \ldots, n$, i.e., $\forall\; \boldsymbol{w}_a, \boldsymbol{w}_b \in \mathbb{R}^{d_w}$*

$$\|\nabla U_i(\boldsymbol{w}_a, \boldsymbol{X}_i) - \nabla U_i(\boldsymbol{w}_b, \boldsymbol{X}_i)\|_2 \leq L_i \|\boldsymbol{w}_a - \boldsymbol{w}_b\|_2. \tag{S1}$$

Let

$$\boldsymbol{U}(\boldsymbol{w}, \mathbf{X}) = \sum_{i=1}^{n} U_i(\boldsymbol{w}, \boldsymbol{X}_i). \tag{S2}$$

Following Assumption 1, the function $\boldsymbol{U}$ is continuously differentiable and the gradient $\nabla \boldsymbol{U}$ is Lipschitz continuous, i.e., $\forall\; \boldsymbol{w}_a, \boldsymbol{w}_b \in \mathbb{R}^{d_w}$, there exists a positive constant $\bar{L}$ such that

$$\|\nabla \boldsymbol{U}(\boldsymbol{w}_a, \mathbf{X}) - \nabla \boldsymbol{U}(\boldsymbol{w}_b, \mathbf{X})\|_2 \leq \bar{L} \|\boldsymbol{w}_a - \boldsymbol{w}_b\|_2, \tag{S3}$$

Now we introduce $F : \mathbb{R}^{nd_w} \times \mathbb{R}^{\sum_i m_i d_x} \mapsto \mathbb{R}$, an aggregate potential function of local variables $\boldsymbol{w}_i(k)$ and local data $\boldsymbol{X}_i$

$$F(\mathbf{w}(k), \mathbf{X}) = \sum_{i=1}^{n} U_i\left(\boldsymbol{w}_i(k), \boldsymbol{X}_i\right). \tag{S4}$$

Following Assumption 1, the function $F$ is continuously differentiable and the gradient $\nabla F$ is Lipschitz continuous, i.e., $\forall\; \mathbf{w}_a, \mathbf{w}_b \in \mathbb{R}^{nd_w}$

$$\|\nabla F(\mathbf{w}_a, \mathbf{X}) - \nabla F(\mathbf{w}_b, \mathbf{X})\|_2 \leq L \|\mathbf{w}_a - \mathbf{w}_b\|_2, \tag{S5}$$

where $L = \max_i \{L_i\}$ and $\nabla F(\mathbf{w}, \mathbf{X}) \triangleq \left[\nabla U_1(\boldsymbol{w}_1, \boldsymbol{X}_1)^\top \quad \ldots \quad \nabla U_n(\boldsymbol{w}_n, \boldsymbol{X}_n)^\top\right]^\top \in \mathbb{R}^{nd_w}$.

**Assumption 2.** *The interaction topology of $n$ networked agents is given as a connected undirected graph $\mathcal{G}\left(\mathcal{V}, \mathcal{E}\right)$.*

For the connected undirected graph $\mathcal{G}\left(\mathcal{V}, \mathcal{E}\right)$, the graph Laplacian $\mathcal{L}$ is a positive semi-definite matrix with one eigenvalue at 0 corresponding to the eigenvector $\mathbf{1}_n$. Furthermore, it follows from Lemma 3 [53] that for all $\mathbf{x} \in \mathbb{R}^n$, such that $\mathbf{1}_n^T \mathbf{x} = 0$, we have $\mathbf{x}^T \mathcal{L}\left(\mathcal{L}\right)^+ \mathbf{x} = \mathbf{x}^T \mathbf{x}$.

Here we require the following condition on $\{\alpha_k\}$ and $\{\beta_k\}$:

**Condition 1.** *Sequences $\{\alpha_k\}$ and $\{\beta_k\}$ are selected as*

$$\alpha_k = \frac{a}{(k+1)^{\delta_2}} \quad \text{and} \quad \beta_k = \frac{b}{(k+1)^{\delta_1}}, \tag{S6}$$

*where $0 < a$, $0 < b$, $0 \leq \delta_1$ and $\frac{1}{2} + \delta_1 < \delta_2 < 1$. Also, the parameter $b = \beta_0$ in sequence $\{\beta_k\}$ is selected such that $\mathcal{W}_0 = (I_n - b\mathcal{L})$ has a single eigenvalue at 1 corresponding to the right and left eigenvectors $\mathbf{1}_n$ and $\mathbf{1}_n^\top$, respectively. Furthermore, the remaining $n-1$ eigenvalues of $\mathcal{W}_0$ are strictly inside the unit circle.*

For sequences $\{\alpha_k\}$ and $\{\beta_k\}$ that satisfy Condition 1, we have $\sum_{k=0}^{\infty} \alpha_k = \infty$, $\sum_{k=0}^{\infty} \beta_k^3 = \infty$ and $\sum_{k=0}^{\infty} \alpha_k^2 < \infty$. Thus $\alpha_k$, $\beta_k$, $\beta_k^2$ and $\beta_k^3$ are not summable sequences while $\alpha_k$ is square-summable. Also note that $\beta_k$ is allowed to be a constant $b$ for all $k \geq 0$. However, $b$ is selected such that $b < 1/\sigma_{\max}(\mathcal{L})$, where $\sigma_{\max}(\cdot)$ denotes the largest singular value. Thus, $b\sigma_{\max}(\mathcal{L}) < 1$.

Let $\mathcal{F}_k$ denotes a filtration generated by the sequence $\{\mathbf{w}_0, \ldots, \mathbf{w}_k\}$, i.e., $\mathbb{E}[\mathbf{v}_k | \mathcal{F}_k] = 0$

$$\mathbb{E}[\mathbf{w}_{k+1} | \mathcal{F}_k] = (\mathcal{W}_k \otimes I_{d_w})\mathbf{w}_k - \alpha_k n \mathbb{E}[\mathbf{g}(\mathbf{w}_k, \mathbf{X}) | \mathcal{F}_k] \quad \text{a.s.}, \tag{S7}$$

where a.s. (almost surely) denotes events that occur with probability one. Let

$$\tilde{\mathbf{g}}(\mathbf{w}_k, \mathbf{X}) = \mathbf{g}(\mathbf{w}_k, \mathbf{X}) - \left(\frac{1}{n}\mathbf{1}_n\mathbf{1}_n^\top \otimes I_{d_w}\right)\mathbf{g}(\mathbf{w}_k, \mathbf{X}). \tag{S8}$$

Note that $\tilde{\mathbf{g}}(\mathbf{w}_k, \mathbf{X}) \triangleq \left[\tilde{\mathbf{g}}_1\left(\boldsymbol{w}_1(k), \boldsymbol{X}_1\right)^\top \quad \ldots \quad \tilde{\mathbf{g}}_n\left(\boldsymbol{w}_n(k), \boldsymbol{X}_n\right)^\top\right]^\top$ denotes the disagreement on the gradient among the distributed agents. Here we make the following assumption regarding $\tilde{\mathbf{g}}(\mathbf{w}_k, \mathbf{X})$:

**Assumption 3.** *There exists a positive constant $\mu_g < \infty$ such that*

$$\sup_{i=1\ldots,n} \mathbb{E}\left[\|\tilde{\mathbf{g}}_i\left(\boldsymbol{w}_i(k), \boldsymbol{X}_i\right)\|_2 |\mathcal{F}_k\right] \leq \sqrt{\mu_g}(1+k)^{\delta_2/2} \quad \text{a.s.,} \tag{S9}$$

*or equivalently*

$$\mathbb{E}\left[\|\tilde{\mathbf{g}}(\mathbf{w}_k, \mathbf{X})\|_2^2 |\mathcal{F}_k\right] \leq n\,\mu_g(1+k)^{\delta_2} \quad \text{a.s.,} \tag{S10}$$

*where $\delta_2$ is defined in Condition 1.*

Note that Assumption 3 does not uniformly bound $\mathbb{E}\left[\|\tilde{\mathbf{g}}(\mathbf{w}_k, \mathbf{X})\|_2^2 |\mathcal{F}_k\right]$. In fact $\mathbb{E}\left[\|\tilde{\mathbf{g}}(\mathbf{w}_k, \mathbf{X})\|_2^2 |\mathcal{F}_k\right]$ can grow unbounded with time, i.e., as $k \to \infty$.

## S2   Useful Lemmas

**Lemma S1.** *Given Assumption 1, for $U$ defined in (11), we have $\forall \boldsymbol{w}_a, \boldsymbol{w}_b \in \mathbb{R}^{d_w}$,*

$$U(\boldsymbol{w}_b, \mathbf{X}) \leq U(\boldsymbol{w}_a, \mathbf{X}) + \nabla U(\boldsymbol{w}_a, \mathbf{X})^\top (\boldsymbol{w}_b - \boldsymbol{w}_a) + \frac{1}{2}nL\|\boldsymbol{w}_b - \boldsymbol{w}_a\|_2^2. \tag{S11}$$

**Proof:** Proof follows from the mean value theorem. ∎

**Lemma S2.** *Given Assumption 2, we have*

$$M \triangleq \left(I_n - \frac{1}{n}\mathbf{1}_n\mathbf{1}_n^\top\right) = \mathcal{L}\left(\mathcal{L}\right)^+, \tag{S12}$$

*where $(\cdot)^+$ denotes the generalized inverse. Furthermore, for all $\mathbf{x} \in \mathbb{R}^n$ such that $\mathbf{x} \notin \mathbb{R}_\mathbf{1}^n$, we have*

$$\tilde{\mathbf{x}}^\top \mathcal{L}\tilde{\mathbf{x}} = \mathbf{x}^\top \mathcal{L}\mathbf{x} > \lambda_2(\mathcal{L})\mathbf{x}^\top \mathbf{x}, \tag{S13}$$

*where $\tilde{\mathbf{x}} = M\mathbf{x}$ is the average-consensus error and $\lambda_2(\mathcal{L})$ denotes the second smallest eigenvalue of $\mathcal{L}$.*

**Proof:** For the connected undirected graph, $\mathcal{L}$ is a positive semi-definite matrix with one eigenvalue at 0 corresponding to the eigenvector $\mathbf{1}_n$. Thus

$$\mathcal{L}\tilde{\mathbf{x}} = \mathcal{L}\left(I_n - \frac{1}{n}\mathbf{1}_n\mathbf{1}_n^\top\right)\mathbf{x} = \mathcal{L}\mathbf{x},$$

and for $\mathbf{x} \in \mathbb{R}^n$ such that $\mathbf{x} \notin \mathbb{R}_\mathbf{1}^n$, we have $\mathbf{x}^\top \mathcal{L}\mathbf{x} > \lambda_2(\mathcal{L})\mathbf{x}^\top \mathbf{x}$. See Lemma 3 of [53] for a detailed proof of (S12). ∎

**Lemma S3.** *Let $f(k)$ is be a non-negative and decreasing sequence for all $k \geq k_0$. Then for all $k \leq K$, we have*

$$\int_k^K f(x)\,dx \leq \sum_{t=k}^K f(t) \leq \int_{k-1}^K f(x)\,dx. \tag{S14}$$

*Furthermore, if $f(k)$ is non-negative and increasing, then for all $k \leq K$ we have*

$$\int_{k-1}^K f(x)\,dx \leq \sum_{t=k}^K f(t) \leq \int_k^{K+1} f(x)\,dx. \tag{S15}$$

**Proof:** See Appendix A2 in [54]. ∎

**Lemma S4.** *For all $k \geq 0$, let $y_k$ be a nonnegative sequence satisfying:*

$$y_{k+1} \leq \left(1 - \frac{\mu_\beta}{(k+1)^{\delta_1}}\right) y_k + \frac{\mu_\zeta}{(k+1)^{\delta_4}}, \tag{S16}$$

*where $0 < \mu_\beta \leq 1$, $0 < \mu_\zeta$, $0 \leq \delta_1 < 1$ and $\delta_1 < \delta_4$ are positive constants. Then we have*

$$y_{k+1} \leq \frac{Y_3}{\exp\left(Y_1(k+1)^{1-\delta_1}\right)} + \frac{Y_2}{(k+1)^{\delta_4-\delta_1}} \tag{S17}$$

*where the constants $Y_1$, $Y_2$ and $Y_3$ are defined as*

$$Y_1 = \frac{\mu_\beta}{1-\delta_1} \tag{S18}$$

$$Y_2 = \frac{\mu_\zeta \delta_4}{\mu_\beta \delta_1} \exp\left(Y_1 2^{1-\delta_1}\right) \tag{S19}$$

$$Y_3 = \exp\left(Y_1\right) \left(y_0 + \sum_{t=0}^{\bar{k}} \left(\frac{1}{(1-\mu_\beta)^t} \frac{\mu_\zeta}{(t+1)^{\delta_4}}\right)\right) \tag{S20}$$

*where $y_0$ is the initial condition and $\bar{k} > 0$ is defined as*

$$\bar{k} = \left\lceil \left(\frac{\delta_4}{\mu_\beta}\right)^{\frac{1}{1-\delta_1}} \right\rceil. \tag{S21}$$

**Proof:** Let

$$\beta_k = \frac{\mu_\beta}{(k+1)^{\delta_1}} \tag{S22}$$

$$\zeta_k = \frac{\mu_\zeta}{(k+1)^{\delta_4}} \tag{S23}$$

and

$$\eta_k = (1 - \beta_k). \tag{S24}$$

Now (S16) can be written as

$$y_{k+1} \leq \eta_k\, y_k + \zeta_k = \zeta_k + y_0 \prod_{t=0}^{k} \eta_t + \sum_{t=0}^{k-1} \zeta_t \left(\prod_{i=t+1}^{k} \eta_i\right) \tag{S25}$$

Since empty product is 1, we have

$$y_{k+1} \leq y_0 \prod_{t=0}^{k} \eta_t + \sum_{t=0}^{k} \zeta_t \left(\prod_{i=t+1}^{k} \eta_i\right) \tag{S26}$$

Note that $\eta_k \leq 1$ and $\eta_k \to 1$ as $k \to \infty$. Then, using

$$1 - \varphi \leq \exp -\varphi, \quad 0 \leq \varphi \leq 1 \tag{S27}$$

yields

$$\prod_{t=0}^{k} \eta_t = \prod_{t=0}^{k} (1 - \beta_t) \leq \exp\left(-\sum_{t=0}^{k} \beta_t\right) \tag{S28}$$

Since $\beta_k$ is monotonically decreasing, from Lemma S3 we have

$$\sum_{t=0}^{k} \beta_t \geq \int_0^k \frac{\mu_\beta}{(t+1)^{\delta_1}}\, dt = \frac{\mu_\beta(k+1)^{1-\delta_1}}{1-\delta_1} - \frac{\mu_\beta}{1-\delta_1} \tag{S29}$$

Thus

$$\prod_{t=0}^{k} \eta_t \le \exp\left(-\sum_{t=0}^{k} \beta_t\right) \le \frac{\exp(Y_1)}{\exp\left(Y_1(k+1)^{1-\delta_1}\right)}, \tag{S30}$$

where

$$Y_1 = \frac{\mu_\beta}{1-\delta_1}.$$

Similarly

$$\prod_{i=t+1}^{k} \eta_i = \prod_{i=t+1}^{k} (1-\beta_i) \le \exp\left(-\sum_{i=t+1}^{k} \beta_i\right) \tag{S31}$$

and

$$\sum_{i=t+1}^{k} \beta_i \ge \int_{t+1}^{k} \frac{\mu_\beta}{(x+1)^{\delta_1}} \, dx = \frac{\mu_\beta(k+1)^{1-\delta_1}}{1-\delta_1} - \frac{\mu_\beta(t+2)^{1-\delta_1}}{1-\delta_1} \tag{S32}$$

Thus

$$\exp\left(-\sum_{i=t+1}^{k} \beta_i\right) \le \exp\left(-\frac{\mu_\beta(k+1)^{1-\delta_1}}{(1-\delta_1)} + \frac{\mu_\beta(t+2)^{1-\delta_1}}{(1-\delta_1)}\right), \tag{S33}$$

$$= \exp\left(-Y_1(k+1)^{1-\delta_1} + Y_1(t+2)^{1-\delta_1}\right). \tag{S34}$$

Note that for some $\bar{t} \in (0, k)$, we have

$$\sum_{t=0}^{k} \zeta_t \left(\prod_{i=t+1}^{k} \eta_i\right) = \sum_{t=0}^{\bar{k}} \zeta_t \left(\prod_{i=t+1}^{k} \eta_i\right) + \sum_{t=\bar{k}+1}^{k} \zeta_t \left(\prod_{i=t+1}^{k} \eta_i\right) \tag{S35}$$

and

$$\sum_{t=\bar{t}+1}^{k} \zeta_t \left(\prod_{i=t+1}^{k} \eta_i\right) \le \sum_{t=\bar{t}+1}^{k} \frac{\mu_\zeta \exp\left(-Y_1(k+1)^{1-\delta_1} + Y_1(t+2)^{1-\delta_1}\right)}{(t+1)^{\delta_4}} \tag{S36}$$

$$= \mu_\zeta \exp\left(-Y_1(k+1)^{1-\delta_1}\right) \sum_{t=\bar{t}+1}^{k} \frac{\exp\left(Y_1(t+2)^{1-\delta_1}\right)}{(t+1)^{\delta_4}} \tag{S37}$$

$$\le \mu_\zeta \exp\left(-Y_1(k+1)^{1-\delta_1}\right) \sum_{t=\bar{t}+1}^{k} \frac{\exp\left(Y_1 t^{1-\delta_1} + Y_1 2^{1-\delta_1}\right)}{(t+1)^{\delta_4}} \tag{S38}$$

$$\le \mu_\zeta \exp\left(-Y_1(k+1)^{1-\delta_1} + Y_1 2^{1-\delta_1}\right) \sum_{t=\bar{t}+1}^{k} \frac{\exp\left(Y_1 t^{1-\delta_1}\right)}{t^{\delta_4}} \tag{S39}$$

Now it follows from Lemma S3 that

$$\sum_{t=\bar{t}+1}^{k} \frac{\exp\left(Y_1 t^{1-\delta_1}\right)}{t^{\delta_4}} \le \int_{\bar{t}}^{k+1} \frac{\exp\left(Y_1 t^{1-\delta_1}\right)}{t^{\delta_4}} \, dt. \tag{S40}$$

Thus we have

$$\sum_{t=\bar{t}+1}^{k} \zeta_t \left(\prod_{i=t+1}^{k} \eta_i\right) \le \mu_\zeta \frac{\exp\left(Y_1 2^{1-\delta_1}\right)}{\exp\left(Y_1(k+1)^{1-\delta_1}\right)} \int_{\bar{t}}^{k+1} \frac{\exp\left(Y_1 t^{1-\delta_1}\right)}{t^{\delta_4}} \, dt \tag{S41}$$

We note

$$\frac{d\left(\exp\left(Y_1 t^{1-\delta_1}\right) t^{-\delta_4+\delta_1}\right)}{dt} = Y_1(1-\delta_1)\exp\left(Y_1 t^{1-\delta_1}\right)t^{-\delta_4} - (\delta_4-\delta_1)\exp\left(Y_1 t^{1-\delta_1}\right)t^{-\delta_4+\delta_1-1} \tag{S42}$$

$$= \left(Y_1(1-\delta_1) - (\delta_4-\delta_1)t^{\delta_1-1}\right)\exp\left(Y_1 t^{1-\delta_1}\right)t^{-\delta_4} \tag{S43}$$

Thus for

$$t \geq \bar{t} = \left(\frac{\delta_4}{Y_1(1-\delta_1)}\right)^{\frac{1}{1-\delta_1}} = \left(\frac{\delta_4}{\mu_\beta}\right)^{\frac{1}{1-\delta_1}} \tag{S44}$$

we have

$$\left(Y_1(1-\delta_1) - \frac{(\delta_4-\delta_1)}{t^{1-\delta_1}}\right) \geq \frac{\mu_\beta \delta_1}{\delta_4} \tag{S45}$$

and

$$\frac{d\left(\exp\left(Y_1 t^{1-\delta_1}\right) t^{-\delta_4+\delta_1}\right)}{dt} \geq \frac{\mu_\beta \delta_1}{\delta_4} \exp\left(Y_1 t^{1-\delta_1}\right)t^{-\delta_4}. \tag{S46}$$

Thus we have

$$\frac{\exp\left(Y_1 t^{1-\delta_1}\right)}{t^{\delta_4}} \leq \frac{\delta_4}{\mu_\beta \delta_1}\frac{d\left(\exp\left(Y_1 t^{1-\delta_1}\right) t^{-\delta_4+\delta_1}\right)}{dt} \tag{S47}$$

and

$$\int_{\bar{t}}^{k+1} \frac{\exp\left(Y_1 t^{1-\delta_1}\right)}{t^{\delta_4}}\,dt \leq \frac{\delta_4}{\mu_\beta \delta_1}\left.\left(\frac{\exp\left(Y_1 t^{1-\delta_1}\right)}{t^{\delta_4-\delta_1}}\right)\right|_{\bar{t}}^{k+1} \tag{S48}$$

$$= \frac{\delta_4}{\mu_\beta \delta_1}\left(\frac{\exp\left(Y_1(k+1)^{1-\delta_1}\right)}{(k+1)^{\delta_4-\delta_1}} - \frac{\exp\left(Y_1(\bar{t})^{1-\delta_1}\right)}{(\bar{t})^{\delta_4-\delta_1}}\right) \tag{S49}$$

Therefore we have

$$\sum_{t=\bar{t}+1}^{k} \zeta_t\left(\prod_{i=t+1}^{k} \eta_i\right) \leq \frac{\mu_\zeta \exp\left(Y_1 2^{1-\delta_1}\right)}{\exp\left(Y_1(k+1)^{1-\delta_1}\right)}\frac{\delta_4}{\mu_\beta \delta_1}\left(\frac{\exp\left(Y_1(k+1)^{1-\delta_1}\right)}{(k+1)^{\delta_4-\delta_1}} - \frac{\exp\left(Y_1(\bar{t})^{1-\delta_1}\right)}{(\bar{t})^{\delta_4-\delta_1}}\right) \tag{S50}$$

$$= \frac{\mu_\zeta \exp\left(Y_1 2^{1-\delta_1}\right)}{\exp\left(Y_1(k+1)^{1-\delta_1}\right)}\left(\frac{\delta_4\,\exp\left(Y_1(k+1)^{1-\delta_1}\right)}{\mu_\beta \delta_1\,(k+1)^{\delta_4-\delta_1}} - Y_4\right) \tag{S51}$$

where $Y_4$ is a positive constant defined as

$$Y_4 = \frac{\delta_4\,\exp\left(Y_1(\bar{t})^{1-\delta_1}\right)}{\mu_\beta \delta_1\,(\bar{t})^{\delta_4-\delta_1}} \tag{S52}$$

Therefore

$$\sum_{t=\bar{t}+1}^{k} \zeta_t\left(\prod_{i=t+1}^{k} \eta_i\right) \leq \frac{\mu_\zeta \exp\left(Y_1 2^{1-\delta_1}\right)}{\exp\left(Y_1(k+1)^{1-\delta_1}\right)}\left(\frac{\delta_4\,\exp\left(Y_1(k+1)^{1-\delta_1}\right)}{\mu_\beta \delta_1\,(k+1)^{\delta_4-\delta_1}} - Y_4\right) \tag{S53}$$

$$= \left(\frac{\mu_\zeta \delta_4\,\exp\left(Y_1 2^{1-\delta_1}\right)}{\mu_\beta \delta_1\,(k+1)^{\delta_4-\delta_1}} - \frac{Y_4\mu_\zeta \exp\left(Y_1 2^{1-\delta_1}\right)}{\exp\left(Y_1(k+1)^{1-\delta_1}\right)}\right) \tag{S54}$$

Thus we have

$$\sum_{t=\bar{t}+1}^{k} \zeta_t\left(\prod_{i=t+1}^{k} \eta_i\right) \leq \frac{\mu_\zeta \delta_4\,\exp\left(Y_1 2^{1-\delta_1}\right)}{\mu_\beta \delta_1\,(k+1)^{\delta_4-\delta_1}} \tag{S55}$$

Now going back to (S26), we can write

$$y_{k+1} \leq y_0 \prod_{t=0}^{k} \eta_t + \sum_{t=0}^{k} \zeta_t \left( \prod_{i=t+1}^{k} \eta_i \right) \tag{S56}$$

$$= y_0 \prod_{t=0}^{k} \eta_t + \sum_{t=0}^{\bar{t}} \zeta_t \left( \prod_{i=t+1}^{k} \eta_i \right) + \sum_{t=\bar{t}+1}^{k} \zeta_t \left( \prod_{i=t+1}^{k} \eta_i \right) \tag{S57}$$

$$= y_0 \prod_{t=0}^{k} \eta_t + \sum_{t=0}^{\bar{t}} \frac{\zeta_t}{\prod_{i=0}^{t} \eta_i} \left( \prod_{i=0}^{k} \eta_i \right) + \sum_{t=\bar{t}+1}^{k} \zeta_t \left( \prod_{i=t+1}^{k} \eta_i \right) \tag{S58}$$

$$= \left( y_0 + \sum_{t=0}^{\bar{t}} \frac{\zeta_t}{\prod_{i=0}^{t} \eta_i} \right) \prod_{t=0}^{k} \eta_t + \sum_{t=\bar{t}+1}^{k} \zeta_t \left( \prod_{i=t+1}^{k} \eta_i \right). \tag{S59}$$

Note that since $\eta_k \leq 1$ and $\eta_k \to 1$ as $k \to \infty$, we have

$$\prod_{i=0}^{t} \eta_i \geq \prod_{i=0}^{t} \eta_0 = (1 - \mu_\beta)^t \tag{S60}$$

Thus

$$y_0 + \sum_{t=0}^{\bar{t}} \frac{\zeta_t}{\prod_{i=0}^{t} \eta_i} \leq y_0 + \sum_{t=0}^{\bar{t}} \frac{1}{(1-\mu_\beta)^t} \frac{\mu_\zeta}{(t+1)^{\delta_4}} \tag{S61}$$

Now define a bounded constant

$$Y_5 \triangleq \left( y_0 + \sum_{t=0}^{\bar{t}} \frac{1}{(1-\mu_\beta)^t} \frac{\mu_\zeta}{(t+1)^{\delta_4}} \right) \tag{S62}$$

Thus we have

$$y_{k+1} \leq \frac{\exp(Y_1) Y_5}{\exp\left( Y_1(k+1)^{1-\delta_1} \right)} + \frac{\mu_\zeta \delta_4 \exp\left( Y_1 2^{1-\delta_1} \right)}{\mu_\beta \delta_1 (k+1)^{\delta_4-\delta_1}} \tag{S63}$$

Now (S17) follows from noting that $Y_3 = \exp(Y_1) Y_5$ and substituting for $Y_2$. ∎

## S3  Proof of Threorem 1

Consider the DULA given in (10)

$$\mathbf{w}_{k+1} = (\mathcal{W}_k \otimes I_{d_w}) \mathbf{w}_k - \alpha_k n \mathbf{g}(\mathbf{w}_k, \mathbf{X}) + \sqrt{2\alpha_k} \mathbf{v}_k. \tag{S64}$$

Define the average-consensus error as $\tilde{\mathbf{w}}_k = (M \otimes I_{d_w}) \mathbf{w}_k$, where $M = I_n - \frac{1}{n} \mathbf{1}_n \mathbf{1}_n^\top$. Thus we have

$$\tilde{\mathbf{w}}_{k+1} = (\mathcal{W}_k \otimes I_{d_w}) \tilde{\mathbf{w}}_k - \alpha_k n \tilde{\mathbf{g}}(\mathbf{w}_k, \mathbf{X}) + \sqrt{2\alpha_k} \tilde{\mathbf{v}}_k \tag{S65}$$

where $\tilde{\mathbf{g}}(\mathbf{w}_k, \mathbf{X}) = (M \otimes I_{d_w}) \mathbf{g}(\mathbf{w}_k, \mathbf{X})$, $\tilde{\mathbf{v}}_k = (M \otimes I_{d_w}) \mathbf{v}_k$ and we used the identities $M (I_n - \beta_k \mathcal{L}) = M - \beta_k \mathcal{L}$ and $(\mathcal{L} \otimes I_{d_w}) \mathbf{w}_k = (\mathcal{L} \otimes I_{d_w}) \tilde{\mathbf{w}}_k$. Taking the norm on both sides yields

$$\|\tilde{\mathbf{w}}_{k+1}\|_2 \leq \| ((I_n - \beta_k \mathcal{L}) \otimes I_{d_w}) \tilde{\mathbf{w}}_k\|_2 + \alpha_k n \|\tilde{\mathbf{g}}(\mathbf{w}_k, \mathbf{X})\|_2 + \sqrt{2\alpha_k} \|\tilde{\mathbf{v}}_k\|_2. \tag{S66}$$

Since $\mathbf{1}_{nd_w}^\top \tilde{\mathbf{w}}_k = 0$, it follows from [55, Lemma 4.4] that

$$\| ((I_n - \beta_k \mathcal{L}) \otimes I_{d_w}) \tilde{\mathbf{w}}_k\|_2 \leq (1 - \beta_k \lambda_2(\mathcal{L})) \|\tilde{\mathbf{w}}_k\|_2, \tag{S67}$$

where $\lambda_2(\cdot)$ denotes the second smallest eigenvalue. Thus we have

$$\|\tilde{\mathbf{w}}_{k+1}\|_2 \leq (1 - \beta_k \lambda_2(\mathcal{L})) \|\tilde{\mathbf{w}}_k\|_2 + \sqrt{2\alpha_k} \|\tilde{\mathbf{v}}_k\|_2 + \alpha_k n \|\tilde{\mathbf{g}}(\mathbf{w}_k, \mathbf{X})\|_2. \tag{S68}$$

Now we use the following inequality

$$(x + y)^2 \le (1 + \theta)x^2 + \left(1 + \frac{1}{\theta}\right)y^2, \tag{S69}$$

for all $x, y, \in \mathbb{R}$ and $\theta > 0$. Since $\beta_k \lambda_2(\mathcal{L}) < 1$ for all $k \ge 0$, selecting

$$\theta = (1 - \beta_k \lambda_2(\mathcal{L}))^{-\frac{1}{2}} - 1$$

yields

$$
\begin{aligned}
\|\tilde{\mathbf{w}}_{k+1}\|_2^2 &\le (1 - \beta_k \lambda_2(\mathcal{L}))^{-\frac{1}{2}} \left((1 - \beta_k \lambda_2(\mathcal{L}))\|\tilde{\mathbf{w}}_k\|_2 + \sqrt{2\alpha_k}\|\tilde{\mathbf{v}}_k\|_2\right)^2 \\
&\quad + n^2 \alpha_k^2 \left(\frac{(1 - \beta_k \lambda_2(\mathcal{L}))^{-\frac{1}{2}}}{(1 - \beta_k \lambda_2(\mathcal{L}))^{-\frac{1}{2}} - 1}\right) \|\tilde{\mathbf{g}}(\mathbf{w}_k, \mathbf{X})\|_2^2
\end{aligned} \tag{S70}
$$

$$
\begin{aligned}
&= (1 - \beta_k \lambda_2(\mathcal{L}))^{-\frac{1}{2}} \left((1 - \beta_k \lambda_2(\mathcal{L}))\|\tilde{\mathbf{w}}_k\|_2 + \sqrt{2\alpha_k}\|\tilde{\mathbf{v}}_k\|_2\right)^2 \\
&\quad + n^2 \alpha_k^2 \left(\frac{1}{1 - (1 - \beta_k \lambda_2(\mathcal{L}))^{\frac{1}{2}}}\right) \|\tilde{\mathbf{g}}(\mathbf{w}_k, \mathbf{X})\|_2^2
\end{aligned} \tag{S71}
$$

Since $\beta_k \lambda_2(\mathcal{L}) < 1$ for all $k \ge 0$, we have

$$(1 - \beta_k \lambda_2(\mathcal{L}))^{\frac{1}{2}} \le \left(1 - \frac{\beta_k \lambda_2(\mathcal{L})}{2}\right), \tag{S72}$$

which results in

$$\left(\frac{1}{1 - (1 - \beta_k \lambda_2(\mathcal{L}))^{\frac{1}{2}}}\right) \le \left(\frac{1}{1 - \left(1 - \frac{\beta_k \lambda_2(\mathcal{L})}{2}\right)}\right) = \frac{2}{\beta_k \lambda_2(\mathcal{L})} \tag{S73}$$

Now it follows from (S71) that

$$
\begin{aligned}
\|\tilde{\mathbf{w}}_{k+1}\|_2^2 &\le (1 - \beta_k \lambda_2(\mathcal{L}))^{-\frac{1}{2}} \left((1 - \beta_k \lambda_2(\mathcal{L}))\|\tilde{\mathbf{w}}_k\|_2 + \sqrt{2\alpha_k}\|\tilde{\mathbf{v}}_k\|_2\right)^2 \\
&\quad + \left(\frac{2n^2 \alpha_k^2}{\beta_k \lambda_2(\mathcal{L})}\right) \|\tilde{\mathbf{g}}(\mathbf{w}_k, \mathbf{X})\|_2^2
\end{aligned} \tag{S74}
$$

Again applying (S69) with the same $\theta$ yields

$$
\begin{aligned}
\left((1 - \beta_k \lambda_2(\mathcal{L}))\|\tilde{\mathbf{w}}_k\|_2 + \sqrt{2\alpha_k}\|\tilde{\mathbf{v}}_k\|_2\right)^2 &\le (1 - \beta_k \lambda_2(\mathcal{L}))^{-\frac{1}{2}} (1 - \beta_k \lambda_2(\mathcal{L}))^2 \|\tilde{\mathbf{w}}_k\|_2^2 \\
&\quad + \left(\frac{(1 - \beta_k \lambda_2(\mathcal{L}))^{-\frac{1}{2}}}{(1 - \beta_k \lambda_2(\mathcal{L}))^{-\frac{1}{2}} - 1}\right) 2\alpha_k \|\tilde{\mathbf{v}}_k\|_2^2
\end{aligned} \tag{S75}
$$

$$
= (1 - \beta_k \lambda_2(\mathcal{L}))^{\frac{3}{2}} \|\tilde{\mathbf{w}}_k\|_2^2 + \left(\frac{(1 - \beta_k \lambda_2(\mathcal{L}))^{-\frac{1}{2}}}{(1 - \beta_k \lambda_2(\mathcal{L}))^{-\frac{1}{2}} - 1}\right) 2\alpha_k \|\tilde{\mathbf{v}}_k\|_2^2 \tag{S76}
$$

$$
\le (1 - \beta_k \lambda_2(\mathcal{L}))^{\frac{3}{2}} \|\tilde{\mathbf{w}}_k\|_2^2 + \left(\frac{4\alpha_k}{\beta_k \lambda_2(\mathcal{L})}\right) \|\tilde{\mathbf{v}}_k\|_2^2 \tag{S77}
$$

Combining (S74) and (S77) yields

$$
\begin{aligned}
\|\tilde{\mathbf{w}}_{k+1}\|_2^2 &\le (1 - \beta_k \lambda_2(\mathcal{L}))\|\tilde{\mathbf{w}}_k\|_2^2 + \left(\frac{4\alpha_k (1 - \beta_k \lambda_2(\mathcal{L}))^{-\frac{1}{2}}}{\beta_k \lambda_2(\mathcal{L})}\right) \|\tilde{\mathbf{v}}_k\|_2^2 \\
&\quad + \left(\frac{2n^2 \alpha_k^2}{\beta_k \lambda_2(\mathcal{L})}\right) \|\tilde{\mathbf{g}}(\mathbf{w}_k, \mathbf{X})\|_2^2
\end{aligned} \tag{S78}
$$

$$
= (1 - \beta_k \lambda_2(\mathcal{L}))\|\tilde{\mathbf{w}}_k\|_2^2 + \frac{2\alpha_k}{\beta_k \lambda_2(\mathcal{L})} \left(\frac{2\|\tilde{\mathbf{v}}_k\|_2^2}{(1 - \beta_k \lambda_2(\mathcal{L}))^{\frac{1}{2}}} + n^2 \alpha_k \|\tilde{\mathbf{g}}(\mathbf{w}_k, \mathbf{X})\|_2^2\right) \tag{S79}
$$

Now taking the conditional expectation $\mathbb{E}\left[\cdot|\mathcal{F}_k\right]$ yields

$$\mathbb{E}\left[\|\tilde{\mathbf{w}}_{k+1}\|_2^2|\mathcal{F}_k\right] \leq (1-\beta_k\lambda_2(\mathcal{L}))\|\tilde{\mathbf{w}}_k\|_2^2$$
$$+\frac{2\alpha_k}{\beta_k\lambda_2(\mathcal{L})}\left(\frac{2\mathbb{E}\left[\|\tilde{\mathbf{v}}_k\|_2^2|\mathcal{F}_k\right]}{(1-\beta_k\lambda_2(\mathcal{L}))^{\frac{1}{2}}}+n^2\alpha_k\mathbb{E}\left[\|\tilde{\mathbf{g}}(\mathbf{w}_k,\mathbf{X})\|_2^2|\mathcal{F}_k\right]\right) \quad \text{(S80)}$$

$$\leq (1-\beta_k\lambda_2(\mathcal{L}))\|\tilde{\mathbf{w}}_k\|_2^2 + \frac{2\alpha_k n^2}{\beta_k\lambda_2(\mathcal{L})}\left(\frac{2d_w}{(1-b\lambda_2(\mathcal{L}))^{\frac{1}{2}}}+an\mu_g\right) \quad \text{(S81)}$$

where we used Assumption 3 and the fact that

$$\|\tilde{\mathbf{v}}_k\|_2^2 = \tilde{\mathbf{v}}_k^\top\tilde{\mathbf{v}}_k = \mathbf{v}_k^\top\left(M\otimes I_{d_w}\right)^\top\left(M\otimes I_{d_w}\right)\mathbf{v}_k = \mathbf{v}_k^\top\left(M\otimes I_{d_w}\right)\mathbf{v}_k \quad \text{(S82)}$$

$$= \mathbf{v}_k^\top\mathbf{v}_k - \frac{1}{n}\mathbf{v}_k^\top\mathbf{1}_{nd_w}\mathbf{1}_{nd_w}^\top\mathbf{v}_k = \mathbf{v}_k^\top\mathbf{v}_k - \frac{1}{n}\mathbf{1}_{nd_w}^\top\mathbf{v}_k\mathbf{v}_k^\top\mathbf{1}_{nd_w}. \quad \text{(S83)}$$

Thus taking the expectation yields

$$\mathbb{E}\left[\|\tilde{\mathbf{v}}_k\|_2^2\right] = \mathbb{E}\left[\mathbf{v}_k^\top\mathbf{v}_k\right] - \frac{1}{n}\mathbf{1}_{nd_w}^\top\mathbb{E}\left[\mathbf{v}_k\mathbf{v}_k^\top\right]\mathbf{1}_{nd_w} = n^2d_w - nd_w \leq n^2d_w. \quad \text{(S84)}$$

Now taking the total expectation of (S81) gives

$$\mathbb{E}\left[\|\tilde{\mathbf{w}}_{k+1}\|_2^2\right] \leq \left(1-\frac{b\lambda_2(\mathcal{L})}{(1+k)^{\delta_1}}\right)\mathbb{E}\left[\|\tilde{\mathbf{w}}_k\|_2^2\right] + \frac{2n^2a}{b\lambda_2(\mathcal{L})}\left(\frac{2d_w}{(1-b\lambda_2(\mathcal{L}))^{\frac{1}{2}}}+an\mu_g\right)\frac{1}{(1+k)^{\delta_2-\delta_1}} \quad \text{(S85)}$$

Now (S85) can be written in the form of (S16) with $\mu_\beta = b\lambda_2(\mathcal{L})$, $\delta_4 = \delta_2 - \delta_1$ and $\mu_\zeta = \frac{2n^2a}{b\lambda_2(\mathcal{L})}\left(\frac{2d_w}{(1-b\lambda_2(\mathcal{L}))^{\frac{1}{2}}}+an\mu_g\right)$. Thus it follows from Lemma S4 that

$$\mathbb{E}\left[\|\tilde{\mathbf{w}}_{k+1}\|_2^2\right] \leq \frac{W_3}{\exp\left(W_1(k+1)^{1-\delta_1}\right)} + \frac{W_2}{(k+1)^{\delta_2-2\delta_1}} \quad \text{(S86)}$$

where $W_1$, $W_2$ and $W_3$ are positive constants defined as

$$W_1 = \frac{b\lambda_2(\mathcal{L})}{(1-\delta_1)}, \quad \text{(S87)}$$

$$W_2 = \frac{2n^2a\left(\frac{2d_w}{(1-b\lambda_2(\mathcal{L}))^{\frac{1}{2}}}+na\mu_g\right)(\delta_2-\delta_1)}{b^2\lambda_2(\mathcal{L})^2\delta_1}\exp\left(W_12^{1-\delta_1}\right), \quad \text{(S88)}$$

$$W_3 = \exp\left(W_1\right)\left(\mathbb{E}\left[\|\tilde{\mathbf{w}}_0\|_2^2\right] + \frac{2n^2a\left(\frac{2d_w}{(1-b\lambda_2(\mathcal{L}))^{\frac{1}{2}}}+na\mu_g\right)}{b\lambda_2(\mathcal{L})}\sum_{\ell=0}^{\bar{k}}\left(\frac{1}{(1-b\lambda_2(\mathcal{L}))^\ell}\frac{1}{(\ell+1)^{\delta_2-\delta_1}}\right)\right) \quad \text{(S89)}$$

in which $\lambda_2(\mathcal{L})$ denotes the second smallest eigenvalue of $\mathcal{L}$ and $\bar{k} > 0$ is defined as

$$\bar{k} = \left\lceil\left(\frac{\delta_2-\delta_1}{b\lambda_2(\mathcal{L})}\right)^{\frac{1}{1-\delta_1}}\right\rceil. \quad \text{(S90)}$$

This concludes the proof of Theorem 1.

∎

## S3.1 Consensus rate

Note that

$$\frac{W_3}{\exp\left(W_1(k+1)^{1-\delta_1}\right)} = \frac{W_3}{(k+1)^{\delta_2-2\delta_1}} \frac{(k+1)^{\delta_2-2\delta_1}}{\exp\left(W_1(k+1)^{1-\delta_1}\right)} \tag{S91}$$

$$\leq \frac{W_3}{(k+1)^{\delta_2-2\delta_1}} \max_{\forall t \geq 0} \left(\frac{(t+1)^{\delta_2-2\delta_1}}{\exp\left(W_1(t+1)^{1-\delta_1}\right)}\right). \tag{S92}$$

We have

$$\max_{\forall t \geq 0} \left(\frac{(t+1)^{\delta_2-2\delta_1}}{\exp\left(W_1(t+1)^{1-\delta_1}\right)}\right) = \exp\left(-\frac{\delta_2-2\delta_1}{1-\delta_1}\right) \left(\frac{\delta_2-2\delta_1}{W_1(1-\delta_1)}\right)^{\frac{\delta_2-2\delta_1}{1-\delta_1}}, \tag{S93}$$

which is attained when

$$\frac{\delta_2-2\delta_1}{1-\delta_1} = W_1(t+1)^{1-\delta_1}. \tag{S94}$$

Note $\frac{\delta_2-2\delta_1}{W_1(1-\delta_1)} = \frac{\delta_2-2\delta_1}{b\lambda_2(\mathcal{L})}$. Define

$$W_4 = W_3 \exp\left(-\frac{\delta_2-2\delta_1}{1-\delta_1}\right) \left(\frac{\delta_2-2\delta_1}{b\lambda_2(\mathcal{L})}\right)^{\frac{\delta_2-2\delta_1}{1-\delta_1}}. \tag{S95}$$

Then

$$\mathbb{E}\left[\|\tilde{\mathbf{w}}_{k+1}\|_2^2\right] \leq \frac{W_2 + W_4}{(k+1)^{\delta_2-2\delta_1}}. \tag{S96}$$

## S4   Proof of Theorem 2

Denote $\bar{w}(t_k)$ and $\tilde{\omega}(t_k)$ by $Y_{k,1}$ and $Y_{k,2}$, respectively. Let $Y_k = [Y_{k,1}^\top \ Y_{k,2}^\top]^\top$ and $X_k(s) = [Y_k^\top \ \bar{w}^\top(s)]^\top$. Then from (21) we have for $s \in [t_k, t_{k+1})$

$$dX_k(s) = \begin{pmatrix} 0 \\ 0 \\ -\nabla U\left(Y_{k,1}, \mathbf{X}\right) - \zeta(Y_{k,2}, Y_{k,1}) \end{pmatrix} ds + \begin{pmatrix} 0 \\ 0 \\ \sqrt{2}dB_s \end{pmatrix}. \tag{S97}$$

Let $X_k$ admit a distribution $p_t(X_k)$. The time evolution of $p_t(X_k)$ is given by the following Fokker-Planck (FP) equation (see 4.1 in [56])

$$\frac{\partial p_t(X_k)}{\partial t} = -\nabla_{\bar{w}} \cdot \left[p_t(X_k)\left(-\nabla U(Y_{k,1}, \mathbf{X}) - \nabla_{\bar{w}} \log p_t(X_k) - \zeta(Y_{k,2}, Y_{k,1})\right)\right] \tag{S98}$$

where $\nabla \cdot [\mathbf{u}(\cdot)]$ denotes the divergence of a vector field $\mathbf{u}(\cdot)$.

We next marginalize out $Y_k$ from $p_t(X_k)$ to obtain $p_t(\bar{w}) = \int p_t(X_k)\,dY_k$ and

$$\frac{\partial p_t(\bar{w})}{\partial t} = -\nabla_{\bar{w}} \cdot \left[\int p_t(\bar{w}, Y_k)\left(-\nabla U(Y_{k,1}, \mathbf{X}) - \nabla_{\bar{w}} \log p_t(\bar{w}, Y_k) - \zeta(Y_{k,2}, Y_{k,1})\right) dY_k\right]. \tag{S99}$$

Note that

$$\int p_t(\bar{w}, Y_k)\nabla_{\bar{w}} \log p_t(\bar{w}, Y_k)\,dY_k = p_t(\bar{w})\nabla \log p_t(\bar{w}) = \nabla p_t(\bar{w}). \tag{S100}$$

We further write (S99) as

$$\frac{\partial p_t(\bar{w})}{\partial t} = \nabla_{\bar{w}} \cdot \left[p_t(\bar{w})\nabla \log p_t(\bar{w})\right]$$

$$- \nabla_{\bar{w}} \cdot \left[\int p_t(\bar{w}, Y_k)\left(-\nabla U(Y_{k,1}, \mathbf{X}) - \zeta(Y_{k,2}, Y_{k,1})\right) dY_k\right] \tag{S101}$$

$$= \nabla_{\bar{\boldsymbol{w}}} \cdot [p_t(\bar{\boldsymbol{w}}) \nabla \log p_t(\bar{\boldsymbol{w}})]$$
$$- \nabla_{\bar{\boldsymbol{w}}} \cdot \left[ \int p_t(\bar{\boldsymbol{w}}, Y_k) \left( -\nabla \boldsymbol{U}(Y_{k,1}, \mathbf{X}) \pm \nabla \boldsymbol{U}(\bar{\boldsymbol{w}}, \mathbf{X}) - \zeta(Y_{k,2}, Y_{k,1}) \right) dY_k \right]$$
$$\text{(S102)}$$

$$= \nabla_{\bar{\boldsymbol{w}}} \cdot [p_t(\bar{\boldsymbol{w}}) \nabla \log p_t(\bar{\boldsymbol{w}})] - \nabla_{\bar{\boldsymbol{w}}} \cdot \left[ \int p_t(\bar{\boldsymbol{w}}, Y_k) \left( -\nabla \boldsymbol{U}(\bar{\boldsymbol{w}}, \mathbf{X}) \right) dY_k \right]$$
$$- \nabla_{\bar{\boldsymbol{w}}} \cdot \left[ \int p_t(\bar{\boldsymbol{w}}, Y_k) \left( \nabla \boldsymbol{U}(\bar{\boldsymbol{w}}, \mathbf{X}) - \nabla \boldsymbol{U}(Y_{k,1}, \mathbf{X}) - \zeta(Y_{k,2}, Y_{k,1}) \right) dY_k \right]$$
$$\text{(S103)}$$

$$= \nabla_{\bar{\boldsymbol{w}}} \cdot [p_t(\bar{\boldsymbol{w}}) \left( \nabla \log p_t(\bar{\boldsymbol{w}}) + \nabla \boldsymbol{U}(\bar{\boldsymbol{w}}, \mathbf{X}) \right)]$$
$$- \nabla_{\bar{\boldsymbol{w}}} \cdot \left[ \int p_t(\bar{\boldsymbol{w}}, Y_k) \left( \nabla \boldsymbol{U}(\bar{\boldsymbol{w}}, \mathbf{X}) - \nabla \boldsymbol{U}(Y_{k,1}, \mathbf{X}) - \zeta(Y_{k,2}, Y_{k,1}) \right) dY_k \right]$$
$$\text{(S104)}$$

Let

$$f_t(\bar{\boldsymbol{w}}) = p_t(\bar{\boldsymbol{w}}) \left( \nabla \log p_t(\bar{\boldsymbol{w}}) + \nabla \boldsymbol{U}(\bar{\boldsymbol{w}}, \mathbf{X}) \right) = p_t(\bar{\boldsymbol{w}}) \nabla \log \left( \frac{p_t(\bar{\boldsymbol{w}})}{p^*(\bar{\boldsymbol{w}})} \right), \qquad \text{(S105)}$$

where we used $\nabla \boldsymbol{U}(\bar{\boldsymbol{w}}, \boldsymbol{X}) = -\nabla \log p^*(\bar{\boldsymbol{w}})$ and let

$$\tilde{f}_t(\bar{\boldsymbol{w}}) = \int p_t(\bar{\boldsymbol{w}}, Y_k) \left( \nabla \boldsymbol{U}(\bar{\boldsymbol{w}}, \mathbf{X}) - \nabla \boldsymbol{U}(Y_{k,1}, \mathbf{X}) - \zeta(Y_{k,2}, Y_{k,1}) \right) dY_k. \qquad \text{(S106)}$$

Thus,

$$\frac{\partial p_t(\bar{\boldsymbol{w}})}{\partial t} = \nabla_{\bar{\boldsymbol{w}}} \cdot \left[ f_t - \tilde{f}_t \right]. \qquad \text{(S107)}$$

We next derive the evolution of the KL divergence between $p_t(\bar{\boldsymbol{w}})$ and $p^*(\bar{\boldsymbol{w}})$, denoted by $F(p_t(\bar{\boldsymbol{w}}))$, i.e.,

$$F(p_t(\bar{\boldsymbol{w}})) = \int p_t(\bar{\boldsymbol{w}}) \log \left( \frac{p_t(\bar{\boldsymbol{w}})}{p^*(\bar{\boldsymbol{w}})} \right) d\bar{\boldsymbol{w}}. \qquad \text{(S108)}$$

Taking the time derivative of $F(p_t(\bar{\boldsymbol{w}}))$ leads to

$$\dot{F}(p_t(\bar{\boldsymbol{w}})) = \frac{d}{dt} \int p_t(\bar{\boldsymbol{w}}) \log \left( \frac{p_t(\bar{\boldsymbol{w}})}{p^*(\bar{\boldsymbol{w}})} \right) d\bar{\boldsymbol{w}} \qquad \text{(S109)}$$

$$= \int \frac{\partial}{\partial t} \left( p_t(\bar{\boldsymbol{w}}) \log \left( p_t(\bar{\boldsymbol{w}}) \right) - p_t(\bar{\boldsymbol{w}}) \log \left( p^*(\bar{\boldsymbol{w}}) \right) \right) d\bar{\boldsymbol{w}} \qquad \text{(S110)}$$

$$= \int \left( \log \left( \frac{p_t(\bar{\boldsymbol{w}})}{p^*(\bar{\boldsymbol{w}})} \right) + 1 \right) \frac{\partial p_t(\bar{\boldsymbol{w}})}{\partial t} d\bar{\boldsymbol{w}}. \qquad \text{(S111)}$$

Let

$$\kappa(\bar{\boldsymbol{w}}) = \log \left( \frac{p_t(\bar{\boldsymbol{w}})}{p^*(\bar{\boldsymbol{w}})} \right) + 1. \qquad \text{(S112)}$$

Using (S107), we further obtain

$$\dot{F}(p_t(\bar{\boldsymbol{w}})) = \int \kappa(\bar{\boldsymbol{w}}) \frac{\partial p_t(\bar{\boldsymbol{w}})}{\partial t} d\bar{\boldsymbol{w}}$$
$$= \int \kappa(\bar{\boldsymbol{w}}) \left( \nabla_{\bar{\boldsymbol{w}}} \cdot [f_t(\bar{\boldsymbol{w}})] \right) d\bar{\boldsymbol{w}} - \int \kappa(\bar{\boldsymbol{w}}) \left( \nabla_{\bar{\boldsymbol{w}}} \cdot \left[ \tilde{f}_t(\bar{\boldsymbol{w}}) \right] \right) d\bar{\boldsymbol{w}}. \qquad \text{(S113)}$$

The first term in (S113) corresponds to the continuous time Langevin dynamics. Using Lemma S5 (an alternative version of Lemma 10.4.1 in [57]) it can be shown that

$$\int \kappa(\bar{\boldsymbol{w}}) \left( \nabla_{\bar{\boldsymbol{w}}} \cdot [f_t(\bar{\boldsymbol{w}})] \right) d\bar{\boldsymbol{w}} = - \int \nabla \log \left( \frac{p_t(\bar{\boldsymbol{w}})}{p^*(\bar{\boldsymbol{w}})} \right)^\top f_t(\bar{\boldsymbol{w}}) d\bar{\boldsymbol{w}}. \qquad \text{(S114)}$$

Substituting (S105), we further get

$$\int \nabla \log \left( \frac{p_t(\bar{\boldsymbol{w}})}{p^*(\bar{\boldsymbol{w}})} \right)^\top f_t(\bar{\boldsymbol{w}}) \, d\bar{\boldsymbol{w}} = \int \left\| \nabla \log \left( \frac{p_t(\bar{\boldsymbol{w}})}{p^*(\bar{\boldsymbol{w}})} \right) \right\|_2^2 p_t(\bar{\boldsymbol{w}}) \, d\bar{\boldsymbol{w}} \qquad (\text{S115})$$

$$= \mathbb{E}_{p_t(\bar{\boldsymbol{w}})} \left[ \left\| \nabla \log \left( \frac{p_t(\bar{\boldsymbol{w}})}{p^*(\bar{\boldsymbol{w}})} \right) \right\|_2^2 \right]. \qquad (\text{S116})$$

For the second term in (S113), we have

$$\int \kappa(\bar{\boldsymbol{w}}) \left( \nabla_{\bar{\boldsymbol{w}}} \cdot \left[ \tilde{f}_t(\bar{\boldsymbol{w}}) \right] \right) d\bar{\boldsymbol{w}}$$

$$= \int \kappa(\bar{\boldsymbol{w}}) \left( \nabla_{\bar{\boldsymbol{w}}} \cdot \left[ \int p_t(\bar{\boldsymbol{w}}, Y_k) \left( \nabla \boldsymbol{U}(\bar{\boldsymbol{w}}, \mathbf{X}) - \nabla \boldsymbol{U}(Y_{k,1}, \mathbf{X}) - \zeta(Y_{k,2}, Y_{k,1}) \right) dY_k \right] \right) d\bar{\boldsymbol{w}}$$
$$(\text{S117})$$

$$= \iint \kappa(\bar{\boldsymbol{w}}) \left( \nabla_{\bar{\boldsymbol{w}}} \cdot \left[ p_t(\bar{\boldsymbol{w}}, Y_k) \left( \nabla \boldsymbol{U}(\bar{\boldsymbol{w}}, \mathbf{X}) - \nabla \boldsymbol{U}(Y_{k,1}, \mathbf{X}) - \zeta(Y_{k,2}, Y_{k,1}) \right) \right] \right) d\bar{\boldsymbol{w}} \, dY_k. \quad (\text{S118})$$

From Lemma S5, we further obtain

$$\iint \kappa(\bar{\boldsymbol{w}}) \left( \nabla_{\bar{\boldsymbol{w}}} \cdot \left[ p_t(\bar{\boldsymbol{w}}, Y_k) \left( \nabla \boldsymbol{U}(\bar{\boldsymbol{w}}, \mathbf{X}) - \nabla \boldsymbol{U}(Y_{k,1}, \mathbf{X}) - \zeta(Y_{k,2}, Y_{k,1}) \right) \right] \right) d\bar{\boldsymbol{w}} \, dY_k$$
$$(\text{S119})$$

$$= - \iint \nabla_{\bar{\boldsymbol{w}}} \kappa(\bar{\boldsymbol{w}})^\top \left( \nabla \boldsymbol{U}(\bar{\boldsymbol{w}}, \mathbf{X}) - \nabla \boldsymbol{U}(Y_{k,1}, \mathbf{X}) - \zeta(Y_{k,2}, Y_{k,1}) \right) p_t(\bar{\boldsymbol{w}}, Y_k) \, d\bar{\boldsymbol{w}} \, dY_k$$

$$= - \int \nabla \log \left( \frac{p_t(\bar{\boldsymbol{w}})}{p^*(\bar{\boldsymbol{w}})} \right)^\top \tilde{f}_t(\bar{\boldsymbol{w}}) \, d\bar{\boldsymbol{w}}. \qquad (\text{S120})$$

It then follows from (S113), (S116) and (S120) that

$$\dot{F}(p_t(\bar{\boldsymbol{w}})) = -\mathbb{E}_{p_t(\bar{\boldsymbol{w}})} \left[ \left\| \nabla \log \left( \frac{p_t(\bar{\boldsymbol{w}})}{p^*(\bar{\boldsymbol{w}})} \right) \right\|_2^2 \right] + \int \nabla \log \left( \frac{p_t(\bar{\boldsymbol{w}})}{p^*(\bar{\boldsymbol{w}})} \right)^\top \tilde{f}_t(\bar{\boldsymbol{w}}) \, d\bar{\boldsymbol{w}}. \qquad (\text{S121})$$

To bound the second term in (S121), we note from (S106) that

$$\int \nabla \log \left( \frac{p_t(\bar{\boldsymbol{w}})}{p^*(\bar{\boldsymbol{w}})} \right)^\top \tilde{f}_t(\bar{\boldsymbol{w}}) \, d\bar{\boldsymbol{w}}$$

$$= \iint \nabla \log \left( \frac{p_t(\bar{\boldsymbol{w}})}{p^*(\bar{\boldsymbol{w}})} \right)^\top \left( \nabla \boldsymbol{U}(\bar{\boldsymbol{w}}, \mathbf{X}) - \nabla \boldsymbol{U}(Y_{k,1}, \mathbf{X}) - \zeta(Y_{k,2}, Y_{k,1}) \right) p_t(\bar{\boldsymbol{w}}, Y_k) \, dY_k \, d\bar{\boldsymbol{w}}$$
$$(\text{S122})$$

$$\leq \frac{1}{2} \iint \left\| \nabla \log \left( \frac{p_t(\bar{\boldsymbol{w}})}{p^*(\bar{\boldsymbol{w}})} \right) \right\|_2^2 p_t(\bar{\boldsymbol{w}}, Y_k) \, dY_k \, d\bar{\boldsymbol{w}}$$
$$+ \frac{1}{2} \iint \| \nabla \boldsymbol{U}(\bar{\boldsymbol{w}}, \mathbf{X}) - \nabla \boldsymbol{U}(Y_{k,1}, \mathbf{X}) - \zeta(Y_{k,2}, Y_{k,1}) \|_2^2 \, p_t(\bar{\boldsymbol{w}}, Y_k) \, dY_k \, d\bar{\boldsymbol{w}}$$
$$(\text{S123})$$

$$\leq \frac{1}{2} \mathbb{E}_{p_t(\bar{\boldsymbol{w}})} \left[ \left\| \nabla \log \left( \frac{p_t(\bar{\boldsymbol{w}})}{p^*(\bar{\boldsymbol{w}})} \right) \right\|_2^2 \right] + \iint \| \nabla \boldsymbol{U}(\bar{\boldsymbol{w}}, \mathbf{X}) - \nabla \boldsymbol{U}(Y_{k,1}, \mathbf{X}) \|_2^2 \, p_t(\bar{\boldsymbol{w}}, Y_k) \, dY_k \, d\bar{\boldsymbol{w}}$$
$$+ \iint \| \zeta(Y_{k,2}, Y_{k,1}) \|_2^2 \, p_t(\bar{\boldsymbol{w}}, Y_k) \, dY_k \, d\bar{\boldsymbol{w}}$$
$$(\text{S124})$$

$$\leq \frac{1}{2} \mathbb{E}_{p_t(\bar{\boldsymbol{w}})} \left[ \left\| \nabla \log \left( \frac{p_t(\bar{\boldsymbol{w}})}{p^*(\bar{\boldsymbol{w}})} \right) \right\|_2^2 \right] + \bar{L}^2 \iint \| \bar{\boldsymbol{w}} - Y_{k,1} \|_2^2 \, p_t(\bar{\boldsymbol{w}}, Y_k) \, dY_k \, d\bar{\boldsymbol{w}}$$
$$+ \iint \| \zeta(Y_{k,2}, Y_{k,1}) \|_2^2 \, p_t(\bar{\boldsymbol{w}}, Y_k) \, dY_k \, d\bar{\boldsymbol{w}}.$$
$$(\text{S125})$$

For the last term in (S125), it follows from Assumption 1 that

$$\|\zeta(\tilde{\omega}(t_k), \bar{\boldsymbol{w}}(t_k))\|_2^2 = \left\|\sum_{i=1}^n \left(\nabla U_i \left(\bar{\boldsymbol{w}}(t_k) + \tilde{\omega}_i(t_k), \boldsymbol{X}_i\right) - \nabla U_i \left(\bar{\boldsymbol{w}}(t_k), \boldsymbol{X}_i\right)\right)\right\|_2^2 \leq L^2 \|\tilde{\omega}(t_k)\|_2^2$$
(S126)

from which we obtain

$$\iint \|\zeta(Y_{k,2}, Y_{k,1})\|_2^2 \, p_t(\bar{\boldsymbol{w}}, Y_k) \, dY_k \, d\bar{\boldsymbol{w}} \leq L^2 \iint \|Y_{k,2}\|_2^2 \, p_t(\bar{\boldsymbol{w}}, Y_k) \, dY_k \, d\bar{\boldsymbol{w}} \qquad \text{(S127)}$$

$$= L^2 \iint \|Y_{k,2}\|_2^2 \, p_t(\bar{\boldsymbol{w}}, Y_{k,2}) \, dY_{k,2} \, d\bar{\boldsymbol{w}} \qquad \text{(S128)}$$

$$= L^2 \int \|Y_{k,2}\|_2^2 \, p(Y_{k,2}) \, dY_{k,2} \qquad \text{(S129)}$$

$$= L^2 \, \mathbb{E}_{p(Y_{k,2})} \|Y_{k,2}\|_2^2 = L^2 \, \mathbb{E}_{p_{t_k}(\tilde{\omega})} \left[\|\tilde{\omega}(t_k)\|_2^2\right].$$
(S130)

For the second term in (S125), since $Y_{k,1} = \bar{\boldsymbol{w}}(t_k)$, it follows from (S97) that $\forall t \in [t_k, t_{k+1}]$

$$\|\bar{\boldsymbol{w}}(t) - \bar{\boldsymbol{w}}(t_k)\|_2^2 = \left\|-\nabla \boldsymbol{U}(\bar{\boldsymbol{w}}(t_k), \mathbf{X})(t - t_k) + \sqrt{2}(B_t - B_{t_k}) - (t - t_k)\zeta(\tilde{\omega}(t_k), \bar{\boldsymbol{w}}(t_k))\right\|_2^2$$
(S131)

$$= 2\|B_t - B_{t_k}\|_2^2 + \|\nabla \boldsymbol{U}(\bar{\boldsymbol{w}}(t_k), \mathbf{X})(t - t_k) + (t - t_k)\zeta(\tilde{\omega}(t_k), \bar{\boldsymbol{w}}(t_k))\|_2^2$$
$$- 2\sqrt{2}(B_t - B_{t_k})^\top \left(\nabla \boldsymbol{U}(\bar{\boldsymbol{w}}(t_k), \mathbf{X})(t - t_k) + (t - t_k)\zeta(\tilde{\omega}(t_k), \bar{\boldsymbol{w}}(t_k))\right)$$
(S132)

$$\leq 2\|B_t - B_{t_k}\|_2^2 + 2\alpha_k^2 \|\nabla \boldsymbol{U}(\bar{\boldsymbol{w}}(t_k), \mathbf{X})\|_2^2 + 2\alpha_k^2 \|\zeta(\tilde{\omega}(t_k), \bar{\boldsymbol{w}}(t_k))\|_2^2$$
$$- 2\sqrt{2}(B_t - B_{t_k})^\top \left(\nabla \boldsymbol{U}(\bar{\boldsymbol{w}}(t_k), \mathbf{X})(t - t_k) + (t - t_k)\zeta(\tilde{\omega}(t_k), \bar{\boldsymbol{w}}(t_k))\right)$$
(S133)

$$\leq 2\|B_t - B_{t_k}\|_2^2 + 2\alpha_k^2 \bar{L}^2 \|\bar{\boldsymbol{w}}(t_k)\|_2^2 + 2\alpha_k^2 L^2 \|\tilde{\omega}(t_k)\|_2^2$$
$$- 2\sqrt{2}(B_t - B_{t_k})^\top \left(\nabla \boldsymbol{U}(\bar{\boldsymbol{w}}(t_k), \mathbf{X})(t - t_k) + (t - t_k)\zeta(\tilde{\omega}(t_k), \bar{\boldsymbol{w}}(t_k))\right).$$
(S134)

The last inequality follows from Lipschitz continuity of $\nabla \boldsymbol{U}$ and assuming $\nabla \boldsymbol{U}(\boldsymbol{0}, \mathbf{X}) = \boldsymbol{0}$. Note that assuming $\nabla \boldsymbol{U}(\boldsymbol{0}, \mathbf{X}) = \boldsymbol{0}$ is only to simplify the notation. Later we will bound the expectation of $\|\bar{\boldsymbol{w}}(t_k)\|_2^2$ in (S134). Then given any finite $\boldsymbol{w}^\star$ such that $\nabla \boldsymbol{U}(\boldsymbol{w}^\star, \mathbf{X}) = \boldsymbol{0}$, $\|\nabla \boldsymbol{U}(\bar{\boldsymbol{w}}(t_k), \mathbf{X})\|_2^2 \leq \bar{L}^2 \|\bar{\boldsymbol{w}}(t_k) - \boldsymbol{w}^\star\|_2^2$, whose expectation is also bounded.

Let $\tilde{B}_t = B_t - B_{t_k}$. Then $\tilde{B}_t$ follows a zero mean Gaussian distribution with a variance of $t - t_k$. Note that $\bar{\boldsymbol{w}}(t)$ depends on $\tilde{B}_t$ for $t_k < t \leq t_{k+1}$ while $Y_k$ is independent of $\tilde{B}_t$. Thus,

$$\iint \|\tilde{B}_t\|^2 p_t(\bar{\boldsymbol{w}}, Y_k) \, dY_k \, d\bar{\boldsymbol{w}} = \int \|\tilde{B}_t\|^2 p_t(\bar{\boldsymbol{w}}) \, d\bar{\boldsymbol{w}} \qquad \text{(S135)}$$

$$= \int \|\tilde{B}_t\|^2 \left[\int p_t(\bar{\boldsymbol{w}}|\tilde{B}_t) p(\tilde{B}_t) \, d\tilde{B}_t\right] d\bar{\boldsymbol{w}} \qquad \text{(S136)}$$

$$= \int \|\tilde{B}_t\|^2 p(\tilde{B}_t) \left[\int p_t(\bar{\boldsymbol{w}}|\tilde{B}_t) \, d\bar{\boldsymbol{w}}\right] d\tilde{B}_t \qquad \text{(S137)}$$

$$= \mathbb{E}_{p(\tilde{B}_t)} \|\tilde{B}_t\|^2 = d_w(t - t_k) \leq \alpha_k d_w. \qquad \text{(S138)}$$

Similarly, for any function $S$ of $Y_k$, we have

$$\iint \tilde{B}_t^\top S(Y_k) p_t(\bar{\boldsymbol{w}}, Y_k) \, dY_k \, d\bar{\boldsymbol{w}} = \iint \tilde{B}_t^\top S(Y_k) \left[\int p_t(\bar{\boldsymbol{w}}|Y_k, \tilde{B}_t) p(Y_k) p(\tilde{B}_t) \, d\tilde{B}_t\right] dY_k \, d\bar{\boldsymbol{w}}$$
(S139)

$$= \int \tilde{B}_t^\top S(Y_k) p(Y_k) p(\tilde{B}_t) \left[\int p_t(\bar{\boldsymbol{w}}|Y_k, \tilde{B}_t) \, d\bar{\boldsymbol{w}}\right] dY_k \, d\tilde{B}_t$$
(S140)

$$= \int p(\tilde{B}_t)\tilde{B}_t^\top S(Y_k)p(Y_k)\, dY_k\, d\tilde{B}_t = 0. \tag{S141}$$

Recall $Y_{k,1} = \bar{\boldsymbol{w}}(t_k)$ and $Y_{k,2} = \tilde{\omega}(t_k)$. Using (S138) and (S141), we further obtain

$$\iint \|\bar{\boldsymbol{w}}(t) - Y_{k,1}\|_2^2\, p_t(\bar{\boldsymbol{w}}, Y_k)\, dY_k\, d\bar{\boldsymbol{w}} \tag{S142}$$
$$\leq \iint \left( 2\alpha_k^2 \bar{L}^2 \|Y_{k,1}\|_2^2 + 2\alpha_k^2 L^2 \|Y_{k,2}\|_2^2 + 2\alpha_k d_w \right) p_t(\bar{\boldsymbol{w}}, Y_k)\, dY_k\, d\bar{\boldsymbol{w}}$$
$$= 2\alpha_k^2 \bar{L}^2 \mathbb{E}_{p(Y_{k,1})} \|Y_{k,1}\|_2^2 + 2\alpha_k^2 L^2 \mathbb{E}_{p(Y_{k,2})} \|Y_{k,2}\|_2^2 + 2\alpha_k d_w \tag{S143}$$
$$= 2\alpha_k^2 \bar{L}^2 \mathbb{E}_{p(\bar{\boldsymbol{w}}(t_k))} \|\bar{\boldsymbol{w}}(t_k)\|_2^2 + 2\alpha_k^2 L^2 \mathbb{E}_{p(\tilde{\omega}(t_k))} \|\tilde{\omega}(t_k)\|_2^2 + 2\alpha_k d_w. \tag{S144}$$

Substituting (S144) and (S130) into (S125) yields

$$\int \nabla \log\left( \frac{p_t(\bar{\boldsymbol{w}})}{p^*(\bar{\boldsymbol{w}})} \right)^\top \tilde{f}_t(\bar{\boldsymbol{w}})\, d\bar{\boldsymbol{w}} \leq \frac{1}{2}\, \mathbb{E}_{p_t(\bar{\boldsymbol{w}})}\left[ \left\| \nabla \log\left( \frac{p_t(\bar{\boldsymbol{w}})}{p^*(\bar{\boldsymbol{w}})} \right) \right\|_2^2 \right] + L^2\, \mathbb{E}_{p(\tilde{\omega}(t_k))}\left[ \|\tilde{\omega}(t_k)\|_2^2 \right]$$
$$+ \bar{L}^2 \left( 2\alpha_k^2 \bar{L}^2 \mathbb{E}_{p(\bar{\boldsymbol{w}}(t_k))} \|\bar{\boldsymbol{w}}(t_k)\|_2^2 + 2\alpha_k^2 L^2 \mathbb{E}_{p(\tilde{\omega}(t_k))} \|\tilde{\omega}(t_k)\|_2^2 + 2\alpha_k d_w \right). \tag{S145}$$

Now substituting (S145) into (S121) gives

$$\dot{F}(p_t(\bar{\boldsymbol{w}})) \leq -\frac{1}{2} \mathbb{E}_{p_t(\bar{\boldsymbol{w}})}\left[ \left\| \nabla \log\left( \frac{p_t(\bar{\boldsymbol{w}})}{p^*(\bar{\boldsymbol{w}})} \right) \right\|_2^2 \right] + L^2\, \mathbb{E}_{p(\tilde{\omega}(t_k))} \|\tilde{\omega}(t_k)\|_2^2 \tag{S146}$$
$$+ \bar{L}^2 \left( 2\alpha_k^2 \bar{L}^2 \mathbb{E}_{p(\bar{\boldsymbol{w}}(t_k))} \|\bar{\boldsymbol{w}}(t_k)\|_2^2 + 2\alpha_k^2 L^2 \mathbb{E}_{p(\tilde{\omega}(t_k))} \|\tilde{\omega}(t_k)\|_2^2 + 2\alpha_k d_w \right)$$
$$= -\frac{1}{2} \mathbb{E}_{p_t(\bar{\boldsymbol{w}})}\left[ \left\| \nabla \log\left( \frac{p_t(\bar{\boldsymbol{w}})}{p^*(\bar{\boldsymbol{w}})} \right) \right\|_2^2 \right] + \left( 2\alpha_k^2 \bar{L}^2 L^2 + L^2 \right) \mathbb{E}_{p(\tilde{\omega}(t_k))} \|\tilde{\omega}(t_k)\|_2^2 \tag{S147}$$
$$+ 2\alpha_k^2 \bar{L}^4\, \mathbb{E}_{p(\bar{\boldsymbol{w}}(t_k))} \|\bar{\boldsymbol{w}}(t_k)\|_2^2 + 2\alpha_k \bar{L}^2 d_w.$$

Recall the log-Sobolev inequality (24)

$$F(p_t(\bar{\boldsymbol{w}})) = \mathbb{E}_{p_t(\bar{\boldsymbol{w}})}\left[ \log\left( \frac{p_t(\bar{\boldsymbol{w}})}{p^*(\bar{\boldsymbol{w}})} \right) \right] \leq \frac{1}{2\rho_U} \mathbb{E}_{p_t(\bar{\boldsymbol{w}})}\left[ \left\| \nabla \log\left( \frac{p_t(\bar{\boldsymbol{w}})}{p^*(\bar{\boldsymbol{w}})} \right) \right\|_2^2 \right], \tag{S148}$$

where $\rho_U$ is the log-Sobolev constant. We then have

$$\dot{F}(p_t(\bar{\boldsymbol{w}})) \leq -\rho_U F(p_t(\bar{\boldsymbol{w}})) + 2\alpha_k^2 \bar{L}^4 \mathbb{E}_{p(\bar{\boldsymbol{w}}(t_k))} \|\bar{\boldsymbol{w}}(t_k)\|_2^2$$
$$+ (2\alpha_k^2 \bar{L}^2 L^2 + L^2) \mathbb{E}_{p(\tilde{\omega}(t_k))} \|\tilde{\omega}(t_k)\|_2^2 + 2\alpha_k \bar{L}^2 d_w \tag{S149}$$
$$= -\rho_U \left( F(p_t(\bar{\boldsymbol{w}})) - \frac{1}{\rho_U} \left( 2\alpha_k^2 \bar{L}^4 \mathbb{E}_{p(\bar{\boldsymbol{w}}(t_k))} \|\bar{\boldsymbol{w}}(t_k)\|_2^2 \right. \right.$$
$$\left. \left. + (2\alpha_k^2 \bar{L}^2 L^2 + L^2) \mathbb{E}_{p(\tilde{\omega}(t_k))} \|\tilde{\omega}(t_k)\|_2^2 + 2\alpha_k \bar{L}^2 d_w \right) \right), \tag{S150}$$

which means $\forall t \in [t_k, t_{k+1}]$

$$F(p_t(\bar{\boldsymbol{w}})) \leq \exp\left( -\rho_U(t - t_k) \right) F(p_{t_k}(\bar{\boldsymbol{w}})) + \frac{1 - \exp\left( -\rho_U(t - t_k) \right)}{\rho_U} \left( 2\alpha_k^2 \bar{L}^4 \mathbb{E}_{p(\bar{\boldsymbol{w}}(t_k))} \|\bar{\boldsymbol{w}}(t_k)\|_2^2 \right.$$
$$\left. + (2\alpha_k^2 \bar{L}^2 L^2 + L^2) \mathbb{E}_{p(\tilde{\omega}(t_k))} \|\tilde{\omega}(t_k)\|_2^2 + 2\alpha_k \bar{L}^2 d_w \right). \tag{S151}$$

Since $\frac{1-\exp(-\rho_U(t-t_k))}{\rho_U} \leq t - t_k \leq \alpha_k$, we further obtain

$$
F(p_t(\bar{\boldsymbol{w}})) \leq \exp\left(-\rho_U(t-t_k)\right) F(p_{t_k}(\bar{\boldsymbol{w}})) + \Bigg(2\alpha_k^3 \bar{L}^4 \mathbb{E}_{p(\bar{\boldsymbol{w}}(t_k))}\|\bar{\boldsymbol{w}}(t_k)\|_2^2
$$
$$
+ (2\alpha_k^3 \bar{L}^2 L^2 + L^2 \alpha_k)\mathbb{E}_{p(\tilde{\omega}(t_k))}\|\tilde{\omega}(t_k)\|_2^2 + 2\alpha_k^2 \bar{L}^2 d_w\Bigg). \tag{S152}
$$

In particular, at $t = t_{k+1}$, we have

$$
F(p_{t_{k+1}}(\bar{\boldsymbol{w}})) \leq \exp\left(-\rho_U \alpha_k\right) F(p_{t_k}(\bar{\boldsymbol{w}})) + \Bigg(2\alpha_k^3 \bar{L}^4 \mathbb{E}_{p(\bar{\boldsymbol{w}}(t_k))}\|\bar{\boldsymbol{w}}(t_k)\|_2^2
$$
$$
+ (2\alpha_k^3 \bar{L}^2 L^2 + L^2 \alpha_k)\mathbb{E}_{p(\tilde{\omega}(t_k))}\|\tilde{\omega}(t_k)\|_2^2 + 2\alpha_k^2 \bar{L}^2 d_w\Bigg). \tag{S153}
$$

We can then use (S153) to recursively bound the KL-divergence $F(p_{t_{k+1}}(\bar{\boldsymbol{w}}))$. Note that $\mathbb{E}_{p(\bar{\boldsymbol{w}}(t_k))}\left[\|\bar{\boldsymbol{w}}(t_k)\|_2^2\right] \leq C_{\bar{\boldsymbol{w}}}$ from Lemma S6 if we select $a$ in $\alpha_k = \frac{a}{(k+1)^{\delta_2}}$ as

$$
a = \frac{1}{n^\gamma}\left(\frac{\rho_U(3\delta_2 - 1)}{25L^4\delta_2}\right)^{\frac{1}{3}}, \quad \gamma > 2. \tag{S154}
$$

Here $\gamma > 2$ is a design parameter to be specified by the user. Also recall from Theorem 1 that

$$
\mathbb{E}\left[\|\tilde{\mathbf{w}}_k\|_2^2\right] \leq \frac{W_3}{\exp\left(W_1 k^{1-\delta_1}\right)} + \frac{W_2}{k^{\delta_2 - 2\delta_1}}. \tag{S155}
$$

Let

$$
Z_k = \frac{W_3(2\alpha_k^3 \bar{L}^2 L^2 + L^2 \alpha_k)}{\exp\left(W_1 k^{1-\delta_1}\right)}, \tag{S156}
$$

$$
C_a = \frac{\rho_U(3\delta_2 - 1)}{25L^4\delta_2}, \tag{S157}
$$

and

$$
\xi_k = 2\alpha_k^3 \bar{L}^4 C_{\bar{\boldsymbol{w}}} + (2\alpha_k^3 \bar{L}^2 L^2 + L^2 \alpha_k)\frac{W_2}{k^{\delta_2 - 2\delta_1}} + 2\alpha_k^2 \bar{L}^2 d_w + Z_k. \tag{S158}
$$

Also define

$$
\theta_k = \frac{\bar{W}_3}{\exp\left(W_1 k^{1-\delta_1}\right)}, \tag{S159}
$$

where

$$
\bar{W}_3 = W_3(2a^3 \bar{L}^2 L^2 + L^2 a). \tag{S160}
$$

Note that $Z_k \leq \theta_k$. Now substituting

$$
\alpha_k = \left(\frac{C_a}{n^{3\gamma}}\right)^{\frac{1}{3}}\frac{1}{(k+1)^{\delta_2}} \leq \left(\frac{C_a}{n^{3\gamma}}\right)^{\frac{1}{3}}\frac{1}{k^{\delta_2}} \tag{S161}
$$

into (S158) yields

$$
\xi_k \leq \frac{2C_a \bar{L}^4 C_{\bar{\boldsymbol{w}}}}{n^{3\gamma} k^{3\delta_2}} + \left(\frac{2C_a \bar{L}^2 L^2}{n^{3\gamma} k^{3\delta_2}} + \frac{C_a^{\frac{1}{3}} L^2}{n^\gamma k^{\delta_2}}\right)\frac{W_2}{k^{\delta_2 - 2\delta_1}} + \frac{2C_a^{\frac{2}{3}} \bar{L}^2 d_w}{n^{2\gamma} k^{2\delta_2}} + Z_k
$$

$$
= \frac{2C_a \bar{L}^4 C_{\bar{\boldsymbol{w}}}}{n^{3\gamma} k^{3\delta_2}} + \frac{2C_a \bar{L}^2 L^2 W_2}{n^{3\gamma} k^{4\delta_2 - 2\delta_1}} + \frac{C_a^{\frac{1}{3}} L^2 W_2}{n^\gamma k^{2\delta_2 - 2\delta_1}} + \frac{2C_a^{\frac{2}{3}} \bar{L}^2 d_w}{n^{2\gamma} k^{2\delta_2}} + Z_k \tag{S162}
$$

$$
= \frac{C_a^{\frac{1}{3}}}{n^\gamma k^{2\delta_2 - 2\delta_1}}\left(\frac{2C_a^{\frac{2}{3}} \bar{L}^4 C_{\bar{\boldsymbol{w}}}}{n^{2\gamma} k^{\delta_2 + 2\delta_1}} + \frac{2C_a^{\frac{2}{3}} \bar{L}^2 L^2 W_2}{n^{2\gamma} k^{2\delta_2}} + L^2 W_2 + \frac{2C_a^{\frac{1}{3}} \bar{L}^2 d_w}{n^\gamma k^{2\delta_1}}\right) + Z_k \tag{S163}
$$

$$
\leq \frac{C_\xi}{k^{2\delta_2 - 2\delta_1}} + \theta_k, \tag{S164}
$$

where

$$C_\xi = \frac{C_a^{\frac{1}{3}}}{n^\gamma} \left( \frac{2C_a^{\frac{2}{3}} \bar{L}^4 C_{\bar{w}}}{n^{2\gamma}} + \frac{2C_a^{\frac{2}{3}} \bar{L}^2 L^2 W_2}{n^{2\gamma}} + L^2 W_2 + \frac{2C_a^{\frac{1}{3}} \bar{L}^2 d_w}{n^\gamma} \right). \tag{S165}$$

Now we rewrite (S153) as

$$F(p_{t_{k+1}}(\bar{\boldsymbol{w}})) \leq \exp\left(-\rho_U \alpha_k\right) F(p_{t_k}(\bar{\boldsymbol{w}})) + \xi_k, \tag{S166}$$

which results in

$$F(p_{t_{k+1}}(\bar{\boldsymbol{w}})) \leq F(p_{t_0}(\bar{\boldsymbol{w}})) \exp\left(-\rho_U \sum_{\ell=0}^{k} \alpha_\ell\right) + \sum_{\ell=0}^{k} \xi_\ell \exp\left(-\rho_U \sum_{i=\ell+1}^{k} \alpha_i\right). \tag{S167}$$

When $\delta_2 \in (0.5, \ 1)$, from Lemma S3 we have

$$\sum_{\ell=0}^{k} \alpha_\ell \geq \int_0^k \frac{a}{(x+1)^{\delta_2}}\, dx = \frac{a(k+1)^{1-\delta_2}}{1-\delta_2} - \frac{a}{1-\delta_2}. \tag{S168}$$

Thus

$$\exp\left(-\rho_U \sum_{\ell=0}^{k} \alpha_\ell\right) \leq \exp\left(-\frac{a\rho_U}{1-\delta_2}(k+1)^{1-\delta_2} + \frac{a\rho_U}{1-\delta_2}\right) \tag{S169}$$

$$= \exp\left(-\frac{a\rho_U}{1-\delta_2}(k+1)^{1-\delta_2}\right) \exp\left(\frac{a\rho_U}{1-\delta_2}\right). \tag{S170}$$

Therefore from (S167) we obtain

$$F(p_{t_{k+1}}(\bar{\boldsymbol{w}})) \leq F(p_{t_0}(\bar{\boldsymbol{w}})) \exp\left(\frac{a\rho_U}{1-\delta_2}\right) \exp\left(-\frac{a\rho_U}{1-\delta_2}(k+1)^{1-\delta_2}\right)$$
$$+ \sum_{\ell=0}^{k} \xi_\ell \exp\left(-\rho_U \sum_{i=\ell+1}^{k} \alpha_i\right). \tag{S171}$$

Substituting (S164) yields

$$F(p_{t_{k+1}}(\bar{\boldsymbol{w}})) \leq F(p_{t_0}(\bar{\boldsymbol{w}})) \exp\left(\frac{a\rho_U}{1-\delta_2}\right) \exp\left(-\frac{a\rho_U}{1-\delta_2}(k+1)^{1-\delta_2}\right)$$
$$+ \sum_{\ell=0}^{k} \frac{C_\xi}{\ell^{2\delta_2-2\delta_1}} \exp\left(-\rho_U \sum_{i=\ell+1}^{k} \alpha_i\right) + \sum_{\ell=0}^{k} \theta_\ell \exp\left(-\rho_U \sum_{i=\ell+1}^{k} \alpha_i\right). \tag{S172}$$

We first consider the last term in (S172). We write

$$\theta_\ell \exp\left(-\rho_U \sum_{i=\ell+1}^{k} \alpha_i\right) = \theta_\ell \exp(\rho_U \alpha_\ell) \exp\left(-\rho_U \sum_{i=\ell}^{k} \alpha_i\right) \leq \theta_\ell \exp(\rho_U a) \exp\left(-\rho_U \sum_{i=\ell}^{k} \alpha_i\right). \tag{S173}$$

From Lemma S3, we have

$$\sum_{i=\ell}^{k} \alpha_i \geq \int_\ell^k \frac{a}{(x+1)^{\delta_2}}\, dx = \frac{a(k+1)^{1-\delta_2}}{1-\delta_2} - \frac{a(\ell+1)^{1-\delta_2}}{1-\delta_2}$$
$$= \frac{a\left((k+1)^{1-\delta_2} - (\ell+1)^{1-\delta_2}\right)}{1-\delta_2}. \tag{S174}$$

Using (S159), (S173) and (S174), we bound the last term in (S172) by

$$\sum_{\ell=0}^{k} \theta_\ell \exp\left(-\rho_U \sum_{i=\ell+1}^{k} \alpha_i\right) \leq \sum_{\ell=0}^{k} \frac{\bar{W}_3 \exp(\rho_U a) \exp\left(-\rho_U \frac{a\left((k+1)^{1-\delta_2} - (\ell+1)^{1-\delta_2}\right)}{1-\delta_2}\right)}{\exp\left(W_1 \ell^{1-\delta_1}\right)} \tag{S175}$$

$$= \bar{W}_3 \exp(\rho_U a) \exp\left(-\frac{\rho_U a}{1-\delta_2}(k+1)^{1-\delta_2}\right) \sum_{\ell=0}^{k} \exp\left(\frac{\rho_U a}{1-\delta_2}(\ell+1)^{1-\delta_2} - W_1 \ell^{1-\delta_1}\right) \tag{S176}$$

$$\leq \bar{W}_3 \exp(\rho_U a) \exp\left(-\frac{\rho_U a}{1-\delta_2}(k+1)^{1-\delta_2}\right) \sum_{\ell=0}^{k} \exp\left(\frac{\rho_U a}{1-\delta_2}(\ell^{1-\delta_2} + 1) - W_1 \ell^{1-\delta_1}\right) \tag{S177}$$

$$= \bar{W}_3 \exp(\rho_U a) \exp\left(-\frac{\rho_U a}{1-\delta_2}(k+1)^{1-\delta_2}\right) \exp\left(\frac{\rho_U a}{1-\delta_2}\right)$$

$$\times \sum_{\ell=0}^{k} \exp\left(\ell^{1-\delta_1}\left(\frac{\rho_U a}{1-\delta_2}\ell^{\delta_1-\delta_2} - W_1\right)\right). \tag{S178}$$

Since $\delta_1 < \delta_2$, $\frac{\rho_U a}{1-\delta_2}\ell^{\delta_1-\delta_2} - W_1$ decreases as $\ell$ increases. There exists a finite integer $\bar{\ell}$ such that $\frac{\rho_U a}{1-\delta_2}\ell^{\delta_1-\delta_2} - W_1 \geq 0$, $\forall \ell < \bar{\ell}$, and $\frac{\rho_U a}{1-\delta_2}\ell^{\delta_1-\delta_2} - W_1 < 0$, $\forall \ell \geq \bar{\ell}$. The sequence $\exp\left(\ell^{1-\delta_1}\left(\frac{\rho_U a}{1-\delta_2}\ell^{\delta_1-\delta_2} - W_1\right)\right)$ is decreasing after $\bar{\ell}$ because

$$\frac{d\left[\ell^{1-\delta_1}\left(\frac{\rho_U a}{1-\delta_2}\ell^{\delta_1-\delta_2} - W_1\right)\right]}{d\ell} < 0, \quad \forall \ell \geq \bar{\ell}. \tag{S179}$$

Note that $\bar{\ell}$ can be computed as

$$\bar{\ell} = \left\lceil \left(\frac{\rho_U a}{(1-\delta_2)W_1}\right)^{\frac{1}{\delta_2-\delta_1}} \right\rceil. \tag{S180}$$

It then follows that for $k > \bar{\ell}$

$$\sum_{\ell=0}^{k} \exp\left(\frac{\rho_U a}{1-\delta_2}\ell^{1-\delta_2} - W_1\ell^{1-\delta_1}\right) =$$

$$\sum_{\ell=0}^{\bar{\ell}} \exp\left(\frac{\rho_U a}{1-\delta_2}\ell^{1-\delta_2} - W_1\ell^{1-\delta_1}\right) \tag{S181}$$

$$+ \sum_{\ell=\bar{\ell}+1}^{k} \exp\left(\frac{\rho_U a}{1-\delta_2}\ell^{1-\delta_2} - W_1\ell^{1-\delta_1}\right).$$

Applying Lemma S3 yields

$$\sum_{\ell=\bar{\ell}+1}^{k} \exp\left(\frac{\rho_U a}{1-\delta_2}\ell^{1-\delta_2} - W_1\ell^{1-\delta_1}\right) \leq \tag{S182}$$

$$\int_{\bar{\ell}}^{k} \exp\left(t^{1-\delta_1}\left(\frac{\rho_U a}{1-\delta_2}t^{\delta_1-\delta_2} - W_1\right)\right) dt. \tag{S183}$$

Let $\kappa = -\left(\frac{\rho_U a}{1-\delta_2}\bar{\ell}^{\delta_1-\delta_2} - W_1\right)$. Note $\kappa > 0$. Then

$$\int_{\bar{\ell}}^{k} \exp\left(t^{1-\delta_1}\left(\frac{\rho_U a}{1-\delta_2}t^{\delta_1-\delta_2} - W_1\right)\right) dt \leq \int_{\bar{\ell}}^{k} \exp\left(-\kappa t^{1-\delta_1}\right) dt \tag{S184}$$

$$= \kappa^{-\frac{1}{1-\delta_1}} \frac{1}{1-\delta_1} \int_{\kappa\bar{\ell}^{1-\delta_1}}^{\kappa k^{1-\delta_1}} \exp\left(-z\right) z^{\frac{\delta_1}{1-\delta_1}} dz \tag{S185}$$

$$\leq \kappa^{-\frac{1}{1-\delta_1}} \frac{1}{1-\delta_1} \int_0^\infty \exp\left(-z\right) z^{\frac{\delta_1}{1-\delta_1}} dz \tag{S186}$$

$$\leq \kappa^{-\frac{1}{1-\delta_1}} \frac{1}{1-\delta_1} \Gamma\left(\frac{1}{1-\delta_1}\right) \tag{S187}$$

where $\Gamma(\cdot)$ is the Gamma function defined as

$$\Gamma(z) = \int_0^\infty x^{z-1} \exp(-x) dx, \quad \forall z > 0. \tag{S188}$$

We substitute (S181) into (S178) together with the bound in (S187) to get

$$\sum_{\ell=0}^k \theta_\ell \exp\left(-\rho_U \sum_{i=\ell+1}^k \alpha_i\right) \leq C_\theta \exp\left(-\frac{\rho_U a}{1-\delta_2}(k+1)^{1-\delta_2}\right) \tag{S189}$$

where

$$C_\theta = \bar{W}_3 \exp(\rho_U a) \exp\left(\frac{\rho_U a}{1-\delta_2}\right) \left(\sum_{\ell=0}^{\bar{\ell}} \exp\left(\frac{\rho_U a}{1-\delta_2}\ell^{1-\delta_2} - W_1\ell^{1-\delta_1}\right)\right.$$
$$\left. + \kappa^{-\frac{1}{1-\delta_1}} \frac{1}{1-\delta_1} \Gamma\left(\frac{1}{1-\delta_1}\right)\right). \tag{S190}$$

We next consider the second term on the right hand side of (S172). Note that for some $\bar{k} \in (0, k)$, we have

$$\sum_{\ell=0}^k \frac{C_\xi}{\ell^{2\delta_2-2\delta_1}} \exp\left(-\rho_U \sum_{i=\ell+1}^k \alpha_i\right) = \sum_{\ell=0}^{\bar{k}} \frac{C_\xi}{\ell^{2\delta_2-2\delta_1}} \exp\left(-\rho_U \sum_{i=\ell+1}^k \alpha_i\right)$$
$$+ \sum_{\ell=\bar{k}+1}^k \frac{C_\xi}{\ell^{2\delta_2-2\delta_1}} \exp\left(-\rho_U \sum_{i=\ell+1}^k \alpha_i\right). \tag{S191}$$

From Lemma S3, we have

$$\sum_{i=\ell+1}^k \alpha_i \geq \int_{\ell+1}^k \frac{a}{(x+1)^{\delta_2}} dx = \frac{a(k+1)^{1-\delta_2}}{1-\delta_2} - \frac{a(\ell+2)^{1-\delta_2}}{1-\delta_2}$$
$$= \frac{a\left((k+1)^{1-\delta_2} - (\ell+2)^{1-\delta_2}\right)}{1-\delta_2}. \tag{S192}$$

The second term in (S191) is bounded as follows

$$\sum_{\ell=\bar{k}+1}^k \frac{C_\xi}{\ell^{2\delta_2-2\delta_1}} \exp\left(-\rho_U \sum_{i=\ell+1}^k \alpha_i\right) \leq \sum_{\ell=\bar{k}+1}^k \frac{C_\xi \exp\left(-\rho_U \frac{a\left((k+1)^{1-\delta_2}-(\ell+2)^{1-\delta_2}\right)}{1-\delta_2}\right)}{\ell^{2\delta_2-2\delta_1}} \tag{S193}$$

$$= \sum_{\ell=\bar{k}+1}^k \frac{C_\xi \exp\left(-\frac{\rho_U a}{1-\delta_2}(k+1)^{1-\delta_2} + \frac{\rho_U a}{1-\delta_2}(\ell+2)^{1-\delta_2}\right)}{\ell^{2\delta_2-2\delta_1}} \tag{S194}$$

$$= C_\xi \exp\left(-\frac{\rho_U a}{1-\delta_2}(k+1)^{1-\delta_2}\right) \sum_{\ell=\bar{k}+1}^k \frac{\exp\left(\frac{\rho_U a}{1-\delta_2}(\ell+2)^{1-\delta_2}\right)}{\ell^{2\delta_2-2\delta_1}} \tag{S195}$$

$$\leq C_\xi \, \exp\left(-\frac{\rho_U a}{1-\delta_2}(k+1)^{1-\delta_2} + \frac{\rho_U a}{1-\delta_2}2^{1-\delta_2}\right) \sum_{\ell=\bar{k}+1}^{k} \frac{\exp\left(\frac{\rho_U a}{1-\delta_2}\ell^{1-\delta_2}\right)}{\ell^{2\delta_2-2\delta_1}}. \tag{S196}$$

Note that from Lemma S3 we have

$$\sum_{\ell=\bar{k}+1}^{k} \frac{\exp\left(\frac{\rho_U a}{1-\delta_2}\ell^{1-\delta_2}\right)}{\ell^{2\delta_2-2\delta_1}} \leq \int_{\bar{k}}^{k+1} \frac{\exp\left(\frac{\rho_U a}{1-\delta_2}\ell^{1-\delta_2}\right)}{\ell^{2\delta_2-2\delta_1}}\, d\ell. \tag{S197}$$

Now it follows from the proof of Lemma S4 (see (S48)) that for $\bar{k} = \left\lceil \left(\frac{2\delta_2-2\delta_1}{\rho_U a}\right)^{\frac{1}{1-\delta_2}} \right\rceil$, we have

$$\int_{\bar{k}}^{k+1} \frac{\exp\left(\frac{\rho_U a}{1-\delta_2}t^{1-\delta_2}\right)}{t^{2\delta_2-2\delta_1}}\, dt \leq \frac{2\delta_2-2\delta_1}{\rho_U a\delta_2}\left(\frac{\exp\left(\frac{\rho_U a}{1-\delta_2}t^{1-\delta_2}\right)}{t^{\delta_2-2\delta_1}}\right)\Bigg|_{\bar{k}}^{k+1} \tag{S198}$$

$$= \frac{2\delta_2-2\delta_1}{\rho_U a\delta_2}\left(\frac{\exp\left(\frac{\rho_U a}{1-\delta_2}(k+1)^{1-\delta_2}\right)}{(k+1)^{\delta_2-2\delta_1}} - \frac{\exp\left(\frac{\rho_U a}{1-\delta_2}\bar{k}^{1-\delta_2}\right)}{\bar{k}^{\delta_2-2\delta_1}}\right). \tag{S199}$$

We further obtain

$$C_\xi \, \exp\left(-\frac{\rho_U a}{1-\delta_2}(k+1)^{1-\delta_2} + \frac{\rho_U a}{1-\delta_2}2^{1-\delta_2}\right)\sum_{\ell=\bar{k}+1}^{k} \frac{\exp\left(\frac{\rho_U a}{1-\delta_2}\ell^{1-\delta_2}\right)}{\ell^{2\delta_2-2\delta_1}}$$

$$\leq C_\xi \, \exp\left(-\frac{\rho_U a}{1-\delta_2}(k+1)^{1-\delta_2} + \frac{\rho_U a}{1-\delta_2}2^{1-\delta_2}\right) \tag{S200}$$

$$\times \frac{2\delta_2-2\delta_1}{\rho_U a\delta_2}\left(\frac{\exp\left(\frac{\rho_U a}{1-\delta_2}(k+1)^{1-\delta_2}\right)}{(k+1)^{\delta_2-2\delta_1}} - \frac{\exp\left(\frac{\rho_U a}{1-\delta_2}\bar{k}^{1-\delta_2}\right)}{\bar{k}^{\delta_2-2\delta_1}}\right)$$

$$= \frac{2C_\xi(\delta_2-\delta_1)}{\rho_U a\delta_2}\frac{\exp\left(\frac{\rho_U a}{1-\delta_2}2^{1-\delta_2}\right)}{\exp\left(\frac{\rho_U a}{1-\delta_2}(k+1)^{1-\delta_2}\right)}\left(\frac{\exp\left(\frac{\rho_U a}{1-\delta_2}(k+1)^{1-\delta_2}\right)}{(k+1)^{\delta_2-2\delta_1}} - \frac{\exp\left(\frac{\rho_U a}{1-\delta_2}\bar{k}^{1-\delta_2}\right)}{\bar{k}^{\delta_2-2\delta_1}}\right) \tag{S201}$$

$$= \frac{2C_\xi(\delta_2-\delta_1)}{\rho_U a\delta_2}\left(\frac{\exp\left(\frac{\rho_U a}{1-\delta_2}2^{1-\delta_2}\right)}{(k+1)^{\delta_2-2\delta_1}} - \frac{\exp\left(\frac{\rho_U a}{1-\delta_2}2^{1-\delta_2}\right)}{\exp\left(\frac{\rho_U a}{1-\delta_2}(k+1)^{1-\delta_2}\right)}\frac{\exp\left(\frac{\rho_U a}{1-\delta_2}\bar{k}^{1-\delta_2}\right)}{\bar{k}^{\delta_2-2\delta_1}}\right) \tag{S202}$$

$$\leq \frac{2C_\xi(\delta_2-\delta_1)}{\rho_U a\delta_2}\left(\frac{\exp\left(\frac{\rho_U a}{1-\delta_2}2^{1-\delta_2}\right)}{(k+1)^{\delta_2-2\delta_1}}\right). \tag{S203}$$

Therefore, using (S189), (S191), and (S203), we rewrite (S172) as

$$F(p_{t_{k+1}}(\bar{\boldsymbol{w}})) \leq F(p_{t_0}(\bar{\boldsymbol{w}}))\exp\left(-\rho_U\sum_{\ell=0}^{k}\alpha_\ell\right) + C_\theta\,\exp\left(-\frac{\rho_U a}{1-\delta_2}(k+1)^{1-\delta_2}\right)$$

$$+ \sum_{\ell=0}^{\bar{k}}\frac{C_\xi}{\ell^{2\delta_2-2\delta_1}}\exp\left(-\rho_U\sum_{i=\ell+1}^{k}\alpha_i\right) + \sum_{\ell=\bar{k}+1}^{k}\frac{C_\xi}{\ell^{2\delta_2-2\delta_1}}\exp\left(-\rho_U\sum_{i=\ell+1}^{k}\alpha_i\right) \tag{S204}$$

$$
\leq F(p_{t_0}(\bar{\boldsymbol{w}})) \exp\left(-\rho_U \sum_{\ell=0}^{k} \alpha_\ell\right) + C_\theta \, \exp\left(-\frac{\rho_U a}{1-\delta_2}(k+1)^{1-\delta_2}\right)
$$

$$
+ \sum_{\ell=0}^{\bar{k}} \frac{C_\xi}{\ell^{2\delta_2-2\delta_1}} \exp\left(-\rho_U \sum_{i=\ell+1}^{k} \alpha_i\right) + \frac{2C_\xi\,(\delta_2-\delta_1)}{\rho_U a \delta_2} \left(\frac{\exp\left(\frac{\rho_U a}{1-\delta_2}2^{1-\delta_2}\right)}{(k+1)^{\delta_2-2\delta_1}}\right)
\tag{S205}
$$

$$
= F(p_{t_0}(\bar{\boldsymbol{w}})) \exp\left(-\rho_U \sum_{\ell=0}^{k} \alpha_\ell\right) + C_\theta \, \exp\left(-\frac{\rho_U a}{1-\delta_2}(k+1)^{1-\delta_2}\right)
$$

$$
+ \sum_{\ell=0}^{\bar{k}} \frac{C_\xi \exp\left(\rho_U \sum_{i=0}^{\ell} \alpha_i\right)}{\ell^{2\delta_2-2\delta_1}} \exp\left(-\rho_U \sum_{i=0}^{k} \alpha_i\right) + \frac{2C_\xi\,(\delta_2-\delta_1)}{\rho_U a \delta_2} \left(\frac{\exp\left(\frac{\rho_U a}{1-\delta_2}2^{1-\delta_2}\right)}{(k+1)^{\delta_2-2\delta_1}}\right)
\tag{S206}
$$

$$
= \left(F(p_{t_0}(\bar{\boldsymbol{w}})) + \sum_{\ell=0}^{\bar{k}} \frac{C_\xi \exp\left(\rho_U \sum_{i=0}^{\ell} \alpha_i\right)}{\ell^{2\delta_2-2\delta_1}}\right) \exp\left(-\rho_U \sum_{\ell=0}^{k} \alpha_\ell\right)
$$

$$
+ C_\theta \, \exp\left(-\frac{\rho_U a}{1-\delta_2}(k+1)^{1-\delta_2}\right) + \frac{2C_\xi\,(\delta_2-\delta_1)}{\rho_U a \delta_2} \left(\frac{\exp\left(\frac{\rho_U a}{1-\delta_2}2^{1-\delta_2}\right)}{(k+1)^{\delta_2-2\delta_1}}\right).
\tag{S207}
$$

Note that

$$
\sum_{\ell=0}^{\bar{k}} \frac{C_\xi \exp\left(\rho_U \sum_{i=0}^{\ell} \alpha_i\right)}{\ell^{2\delta_2-2\delta_1}} = \sum_{\ell=0}^{\bar{k}} \frac{C_\xi \exp(\rho_U a) \exp\left(\rho_U \sum_{i=1}^{\ell} \alpha_i\right)}{\ell^{2\delta_2-2\delta_1}}
\tag{S208}
$$

$$
\leq \sum_{\ell=0}^{\bar{k}} \frac{C_\xi \exp(\rho_U a) \exp\left(\frac{a\rho_U}{1-\delta_2}\left(\ell^{1-\delta_2}\right)\right)}{\ell^{2\delta_2-2\delta_1}},
\tag{S209}
$$

where the last inequality follows from

$$
\sum_{i=1}^{\ell} \alpha_i \leq \int_0^{\ell} \frac{a}{(x+1)^{\delta_2}}\,dx = \frac{a}{1-\delta_2}\left((\ell+1)^{1-\delta_2}-1\right)
\tag{S210}
$$

$$
\leq \frac{a}{1-\delta_2}\left(\ell^{1-\delta_2}+1^{1-\delta_2}-1\right) = \frac{a}{1-\delta_2}\left(\ell^{1-\delta_2}\right).
\tag{S211}
$$

Therefore (S207) can be written as

$$
F(p_{t_{k+1}}(\bar{\boldsymbol{w}})) \leq \left(F(p_{t_0}(\bar{\boldsymbol{w}})) + \sum_{\ell=0}^{\bar{k}} \frac{C_\xi \exp(\rho_U a) \exp\left(\frac{a\rho_U}{1-\delta_2}\left(\ell^{1-\delta_2}\right)\right)}{\ell^{2\delta_2-2\delta_1}}\right) \exp\left(-\rho_U \sum_{\ell=0}^{k} \alpha_\ell\right)
$$

$$
+ C_\theta \, \exp\left(-\frac{\rho_U a}{1-\delta_2}(k+1)^{1-\delta_2}\right) + \frac{2C_\xi\,(\delta_2-\delta_1)}{\rho_U a \delta_2} \, \exp\left(\frac{\rho_U a}{1-\delta_2}2^{1-\delta_2}\right) \frac{1}{(k+1)^{\delta_2-2\delta_1}}.
\tag{S212}
$$

Define $C_\rho$ as

$$
C_\rho = \frac{2\,(\delta_2-\delta_1)}{\rho_U \delta_2} \, \exp\left(\frac{\rho_U a}{1-\delta_2}2^{1-\delta_2}\right).
\tag{S213}
$$

Now (S212) can be written as

$$F(p_{t_{k+1}}(\bar{\boldsymbol{w}})) \leq \left( F(p_{t_0}(\bar{\boldsymbol{w}})) + \sum_{\ell=0}^{\bar{k}} \frac{C_\xi \exp\left(\rho_U a\right) \exp\left(\frac{a\rho_U}{1-\delta_2}\left(\ell^{1-\delta_2}\right)\right)}{\ell^{2\delta_2 - 2\delta_1}} \right) \exp\left(-\rho_U \sum_{\ell=0}^{k} \alpha_\ell\right)$$
$$+ C_\theta \exp\left(-\frac{\rho_U a}{1-\delta_2}(k+1)^{1-\delta_2}\right) + \frac{C_\xi C_\rho}{a} \frac{1}{(k+1)^{\delta_2 - 2\delta_1}}. \tag{S214}$$

From (S161) we have $a = \left(\frac{C_a}{n^{3\gamma}}\right)^{\frac{1}{3}}$ and from (S165) we have $C_\xi = a\bar{C}_\xi$, where

$$\bar{C}_\xi = \left( \frac{2C_a^{\frac{2}{3}} \bar{L}^4 C_{\bar{\boldsymbol{w}}}}{n^{2\gamma}} + \frac{2C_a^{\frac{2}{3}} \bar{L}^2 L^2 W_2}{n^{2\gamma}} + L^2 W_2 + \frac{2C_a^{\frac{1}{3}} \bar{L}^2 d_w}{n^\gamma} \right). \tag{S215}$$

Thus it follows from (S214) that

$$F(p_{t_{k+1}}(\bar{\boldsymbol{w}})) \leq \left( F(p_{t_0}(\bar{\boldsymbol{w}})) + \sum_{\ell=0}^{\bar{k}} \frac{C_\xi \exp\left(\rho_U a\right) \exp\left(\frac{a\rho_U}{1-\delta_2}\left(\ell^{1-\delta_2}\right)\right)}{\ell^{2\delta_2 - 2\delta_1}} \right) \exp\left(-\rho_U \sum_{\ell=0}^{k} \alpha_\ell\right)$$
$$+ C_\theta \exp\left(-\frac{\rho_U a}{1-\delta_2}(k+1)^{1-\delta_2}\right) + \bar{C}_\xi C_\rho \frac{1}{(k+1)^{\delta_2 - 2\delta_1}}. \tag{S216}$$

Considering the term $L^2 W_2$ in (S215), recall $W_2 = \frac{2n^2 a \left( \frac{2d_w}{(1-b\lambda_2(\mathcal{L}))^{\frac{1}{2}}} + na\mu_g \right)(\delta_2 - \delta_1)}{b^2 \lambda_2(\mathcal{L})^2 \delta_1} \exp\left(W_1 2^{1-\delta_1}\right)$ from Theorem 1, which, together with $a$ in (S154), leads to

$$W_2 = \frac{(\delta_2 - \delta_1) \exp\left(W_1 2^{1-\delta_1}\right)}{b^2 \lambda_2(\mathcal{L})^2 \delta_1} \left( \frac{4d_w}{n^{\gamma-2}(1-b\lambda_2(\mathcal{L}))^{\frac{1}{2}}} \left( \frac{\rho_U(3\delta_2 - 1)}{25L^4 \delta_2} \right)^{\frac{1}{3}} + \frac{2\mu_g}{n^{2\gamma-3}} \left( \frac{\rho_U(3\delta_2 - 1)}{25L^4 \delta_2} \right)^{\frac{2}{3}} \right) \tag{S217}$$

$$= \frac{(\delta_2 - \delta_1) \exp\left(W_1 2^{1-\delta_1}\right)}{n^{\gamma-2} b^2 \lambda_2(\mathcal{L})^2 \delta_1} \left( \frac{4d_w}{(1-b\lambda_2(\mathcal{L}))^{\frac{1}{2}}} \left( \frac{\rho_U(3\delta_2 - 1)}{25L^4 \delta_2} \right)^{\frac{1}{3}} + \frac{2\mu_g}{n^{\gamma-1}} \left( \frac{\rho_U(3\delta_2 - 1)}{25L^4 \delta_2} \right)^{\frac{2}{3}} \right) \tag{S218}$$

$$= \frac{\bar{C}_{W_2}}{n^{\gamma-2}}, \tag{S219}$$

where

$$\bar{C}_{W_2} = \frac{(\delta_2 - \delta_1) \exp\left(W_1 2^{1-\delta_1}\right)}{b^2 \lambda_2(\mathcal{L})^2 \delta_1} \left( \frac{4d_w}{(1-b\lambda_2(\mathcal{L}))^{\frac{1}{2}}} \left( \frac{\rho_U(3\delta_2 - 1)}{25L^4 \delta_2} \right)^{\frac{1}{3}} + \frac{2\mu_g}{n^{\gamma-1}} \left( \frac{\rho_U(3\delta_2 - 1)}{25L^4 \delta_2} \right)^{\frac{2}{3}} \right). \tag{S220}$$

Note that when $\gamma > 2$, $\bar{C}_{W_2}$ is bounded as $n$ increases. Substituting (S219) into (S215) yields

$$\bar{C}_\xi = \left( \frac{2C_a^{\frac{2}{3}} \bar{L}^4 C_{\bar{\boldsymbol{w}}}}{n^{2\gamma}} + \frac{2C_a^{\frac{2}{3}} \bar{L}^2 L^2 \bar{C}_{W_2}}{n^{3\gamma-2}} + \frac{L^2 \bar{C}_{W_2}}{n^{\gamma-2}} + \frac{2C_a^{\frac{1}{3}} \bar{L}^2 d_w}{n^\gamma} \right). \tag{S221}$$

Now substituting (S221) into (S216), we have

$$
F(p_{t_{k+1}}(\bar{\boldsymbol{w}})) \leq \left( F(p_{t_0}(\bar{\boldsymbol{w}})) + \sum_{\ell=0}^{\bar{k}} \frac{C_\xi \exp\left(\rho_U a\right) \exp\left(\frac{a\rho_U}{1-\delta_2}\left(\ell^{1-\delta_2}\right)\right)}{\ell^{2\delta_2 - 2\delta_1}} \right) \exp\left(-\rho_U \sum_{\ell=0}^{k} \alpha_\ell\right)
$$
$$
+ \left( \frac{2C_a^{\frac{2}{3}}\bar{L}^4 C_{\bar{\boldsymbol{w}}} C_\rho}{n^{2\gamma}} + \frac{2C_a^{\frac{2}{3}}\bar{L}^2 L^2 \bar{C}_{W_2} C_\rho}{n^{3\gamma-2}} + \frac{L^2 \bar{C}_{W_2} C_\rho}{n^{\gamma-2}} + \frac{2C_a^{\frac{1}{3}}\bar{L}^2 d_w C_\rho}{n^\gamma} \right) \frac{1}{(k+1)^{\delta_2 - 2\delta_1}}
$$
$$
+ C_\theta \exp\left(-\frac{\rho_U a}{1-\delta_2}(k+1)^{1-\delta_2}\right).
$$
(S222)

Note that the first and the third terms in (S222) are exponentially decaying while the second term is polynomial in $k$. However the polynomial term decreases with the number of agents, $n$. We further rewrite (S222) as

$$
F(p_{t_{k+1}}(\bar{\boldsymbol{w}})) \leq \left( F(p_{t_0}(\bar{\boldsymbol{w}})) + \bar{C}_{F_1} \right) \exp\left(-\rho_U \sum_{\ell=0}^{k} \alpha_\ell\right) + \frac{1}{n^{\gamma-2}} \frac{\bar{C}_{F_2}}{(k+1)^{\delta_2 - 2\delta_1}}
$$
$$
+ \bar{C}_{F_3} \exp\left(-\frac{\rho_U a}{1-\delta_2}(k+1)^{1-\delta_2}\right)
$$
(S223)

where

$$
\bar{C}_{F_1} = \sum_{\ell=0}^{\check{k}} \frac{C_\xi \exp\left(\rho_U a\right) \exp\left(\frac{a\rho_U}{1-\delta_2}\left(\ell^{1-\delta_2}\right)\right)}{\ell^{2\delta_2 - 2\delta_1}}, \qquad \check{k} = \left\lceil \left(\frac{2\delta_2 - 2\delta_1}{\rho_U a}\right)^{\frac{1}{1-\delta_2}} \right\rceil
$$
(S224)

$$
\bar{C}_{F_2} = \left( \frac{2C_a^{\frac{2}{3}}\bar{L}^4 C_{\bar{\boldsymbol{w}}} C_\rho}{n^{\gamma+2}} + \frac{2C_a^{\frac{2}{3}}\bar{L}^2 L^2 \bar{C}_{W_2} C_\rho}{n^{2\gamma}} + L^2 \bar{C}_{W_2} C_\rho + \frac{2C_a^{\frac{1}{3}}\bar{L}^2 d_w C_\rho}{n^2} \right)
$$
(S225)

$$
\bar{C}_{F_3} = W_3 (2a^3 \bar{L}^2 L^2 + L^2 a) \exp\left(\frac{\rho_U a(2-\delta_2)}{1-\delta_2}\right) \left( \sum_{\ell=0}^{\check{\ell}} \exp\left(\frac{\rho_U a}{1-\delta_2}\ell^{1-\delta_2} - W_1 \ell^{1-\delta_1}\right) \right.
$$
$$
\left. + \kappa^{-\frac{1}{1-\delta_1}} \frac{1}{1-\delta_1}\Gamma\left(\frac{1}{1-\delta_1}\right) \right)
$$
(S226)

$$
\kappa = -\left( \frac{\rho_U a}{1-\delta_2}\check{\ell}^{\delta_1-\delta_2} - W_1 \right), \qquad \check{\ell} = \left\lceil \left(\frac{\rho_U a}{(1-\delta_2)W_1}\right)^{\frac{1}{\delta_2-\delta_1}} \right\rceil
$$
(S227)

$$
C_\xi = a\left( \frac{2C_a^{\frac{2}{3}}\bar{L}^4 C_{\bar{\boldsymbol{w}}}}{n^{2\gamma}} + \frac{2C_a^{\frac{2}{3}}\bar{L}^2 L^2 W_2}{n^{2\gamma}} + L^2 W_2 + \frac{2C_a^{\frac{1}{3}}\bar{L}^2 d_w}{n^\gamma} \right)
$$
(S228)

$$
C_\rho = \frac{2(\delta_2 - \delta_1)}{\rho_U \delta_2} \exp\left(\frac{\rho_U a}{1-\delta_2}2^{1-\delta_2}\right)
$$
(S229)

$$
C_a = \frac{\rho_U(3\delta_2 - 1)}{25L^4\delta_2} \qquad \Rightarrow \qquad a = \left(\frac{\rho_U(3\delta_2 - 1)}{25n^{3\gamma}L^4\delta_2}\right)^{\frac{1}{3}}
$$
(S230)

$$
C_{\bar{\boldsymbol{w}}} = \max\left\{ \mathbb{E}\left[\|\bar{\boldsymbol{w}}(t_0)\|_2^2\right], \frac{2c_1 + \frac{4}{\rho_U}(c_2 + d_1)}{1 - \frac{24n^4 L^4\delta_2 a^3}{\rho_U(3\delta_2-1)}} \right\}
$$
(S231)

$$
\bar{C}_{W_2} = \frac{(\delta_2 - \delta_1)\exp\left(W_1 2^{1-\delta_1}\right)}{b^2\lambda_2(\mathcal{L})^2\delta_1} \left( 4d_w \left(\frac{\rho_U(3\delta_2-1)}{25L^4\delta_2}\right)^{\frac{1}{3}} + \frac{2\mu_g}{n^{\gamma-1}}\left(\frac{\rho_U(3\delta_2-1)}{25L^4\delta_2}\right)^{\frac{2}{3}} \right)
$$
(S232)

while $W_1$, $W_2$ and $W_3$ are defined in (S87), (S88) and (S89), respectively, and $c_1$, $c_2$ and $d_1$ are given in Lemma S6. This concludes the proof of Theorem 2.

∎

## S4.1 Lemmas used in the proof of Theorem 2

**Lemma S5.** *For $f_t(\bar{\boldsymbol{w}})$, $\tilde{f}_t(\bar{\boldsymbol{w}})$ and $\kappa(\bar{\boldsymbol{w}})$ defined in (S105), (S106) and (S112), respectively, we have*

$$\int \kappa(\bar{\boldsymbol{w}}) \left( \nabla_{\bar{\boldsymbol{w}}} \cdot [f_t(\bar{\boldsymbol{w}})] \right) d\bar{\boldsymbol{w}} = -\int \nabla \log \left( \frac{p_t(\bar{\boldsymbol{w}})}{p^*(\bar{\boldsymbol{w}})} \right)^\top f_t(\bar{\boldsymbol{w}}) \, d\bar{\boldsymbol{w}}, \tag{S233}$$

*and*

$$\int \kappa(\bar{\boldsymbol{w}}) \left( \nabla_{\bar{\boldsymbol{w}}} \cdot \left[ \tilde{f}_t(\bar{\boldsymbol{w}}) \right] \right) d\bar{\boldsymbol{w}} = -\int \nabla \log \left( \frac{p_t(\bar{\boldsymbol{w}})}{p^*(\bar{\boldsymbol{w}})} \right)^\top \tilde{f}_t(\bar{\boldsymbol{w}}) \, d\bar{\boldsymbol{w}}. \tag{S234}$$

**Proof:** This lemma is similar to [57, Lemma 10.4.1]. Here we use the identity for $x \in \mathbb{R}^d$, $b(x) : \mathbb{R}^d \mapsto \mathbb{R}$ and $\mathbf{a}(x) : \mathbb{R}^d \mapsto \mathbb{R}^d$:

$$\nabla_x \cdot [b(x)\mathbf{a}(x)] = (\nabla_x b(x))^\top \mathbf{a}(x) + b(x) \left( \nabla_x \cdot [\mathbf{a}(x)] \right). \tag{S235}$$

Thus we have

$$\kappa(\bar{\boldsymbol{w}}) \left( \nabla_{\bar{\boldsymbol{w}}} \cdot f_t(\bar{\boldsymbol{w}}) \right) = \nabla_{\bar{\boldsymbol{w}}} \cdot [\kappa(\bar{\boldsymbol{w}}) f_t(\bar{\boldsymbol{w}})] - (\nabla_{\bar{\boldsymbol{w}}} \kappa(\bar{\boldsymbol{w}}))^\top f_t(\bar{\boldsymbol{w}}). \tag{S236}$$

Note that

$$\int \nabla_{\bar{\boldsymbol{w}}} \cdot [\kappa(\bar{\boldsymbol{w}}) f_t(\bar{\boldsymbol{w}})] \, d\bar{\boldsymbol{w}} = \int \dots \int \sum_{i=1}^{d_w} \frac{\partial}{\partial \bar{\boldsymbol{w}}_i} \left( \kappa(\bar{\boldsymbol{w}}) f_t(\bar{\boldsymbol{w}}) \right) d\bar{\boldsymbol{w}}_1 \dots d\bar{\boldsymbol{w}}_{d_w} \tag{S237}$$

$$= \sum_{i=1}^{d_w} \int \dots \int \frac{\partial}{\partial \bar{\boldsymbol{w}}_i} \left( \kappa(\bar{\boldsymbol{w}}) f_t(\bar{\boldsymbol{w}}) \right) d\bar{\boldsymbol{w}}_1 \dots d\bar{\boldsymbol{w}}_{i-1} \, d\bar{\boldsymbol{w}}_i \, d\bar{\boldsymbol{w}}_{i+1} \dots d\bar{\boldsymbol{w}}_{d_w} \tag{S238}$$

$$= \sum_{i=1}^{d_w} \int \dots \int \left( (\kappa(\bar{\boldsymbol{w}}) f_t(\bar{\boldsymbol{w}})) \, |_{\bar{\boldsymbol{w}}_i=-\infty}^{\bar{\boldsymbol{w}}_i=+\infty} \right) d\bar{\boldsymbol{w}}_1 \dots d\bar{\boldsymbol{w}}_{i-1} \, d\bar{\boldsymbol{w}}_{i+1} \dots d\bar{\boldsymbol{w}}_{d_w} \tag{S239}$$

$$= 0. \tag{S240}$$

The last equality holds when $\left( (\kappa(\bar{\boldsymbol{w}}) f_t(\bar{\boldsymbol{w}})) \, |_{\bar{\boldsymbol{w}}_i=-\infty}^{\bar{\boldsymbol{w}}_i=+\infty} \right) = 0$ for all $i = 1, \dots, d_w$, which is satisfied under the condition that $p_t(\bar{\boldsymbol{w}}) \to 0$ as $\bar{\boldsymbol{w}}_i \to \pm\infty$. The same technical condition has been assumed in the literature, see e.g., one of the assumptions in [58, Theorem 3.1] and the "sufficiently fast decay at infinity" condition in [4, Appendix A.1].

It then follows that

$$\int \kappa(\bar{\boldsymbol{w}}) \nabla_{\bar{\boldsymbol{w}}} \cdot [f_t(\bar{\boldsymbol{w}})] \, d\bar{\boldsymbol{w}} = -\int \nabla \kappa(\bar{\boldsymbol{w}})^\top f_t(\bar{\boldsymbol{w}}) \, d\bar{\boldsymbol{w}}. \tag{S241}$$

Similar argument can be used to prove (S234). ∎

**Lemma S6.** *Let $\bar{w}^*$ denotes samples from the target-distribution $p^*$, i.e., $\bar{w}^* \sim p^*$ and $\bar{w}^*$ satisfies*

$$\mathbb{E} \left[ \|\bar{w}^*\|_2^2 \right] \le c_1. \tag{S242}$$

*Let $\bar{\boldsymbol{w}}(t)$ denotes samples from the distribution $p_t(\bar{\boldsymbol{w}})$, i.e., $\bar{\boldsymbol{w}}(t) \sim p_t(\bar{\boldsymbol{w}})$. Suppose that the KL-divergence between the initial distribution $p_{t_0}(\bar{\boldsymbol{w}})$ and the target distribution $p^*$, denoted as $F(p_{t_0}(\bar{\boldsymbol{w}}))$ is bounded by $c_2$, i.e.,*

$$F(p_{t_0}(\bar{\boldsymbol{w}})) \le c_2. \tag{S243}$$

*Also, suppose that $2\delta_2 - 2\delta_1 > 1$ and $a$ in (15) is chosen such that $\frac{24 n^4 L^4 \delta_2 a^3}{\rho_U (3\delta_2 - 1)} < 1$. Then there exists a $C_{\bar{\boldsymbol{w}}} > 0$ such that $\forall k \ge 0$,*

$$\mathbb{E} \left[ \|\bar{\boldsymbol{w}}(t_k)\|_2^2 \right] \le C_{\bar{\boldsymbol{w}}}, \tag{S244}$$

*where*

$$C_{\bar{\boldsymbol{w}}} = \max\left\{\mathbb{E}_{\bar{\boldsymbol{w}}(t_0)\sim p_{t_0}}\left[\|\bar{\boldsymbol{w}}(t_0)\|_2^2\right], \frac{2c_1 + \frac{4}{\rho_U}(c_2+d_1)}{1 - \frac{24n^4L^4\delta_2 a^3}{\rho_U(3\delta_2-1)}}\right\} \tag{S245}$$

*and $d_1$ is a positive constant satisfying*

$$d_1 \geq \sum_{k=0}^{\infty}(2\alpha_k^3\bar{L}^2L^2 + L^2\alpha_k)\mathbb{E}_{p(\tilde{\omega}(t_k))}\|\tilde{\omega}(t_k)\|_2^2 + 2\alpha_k^2\bar{L}^2 d_w. \tag{S246}$$

**Proof :** We prove the boundedness of $\mathbb{E}_{p_{t_k}(\bar{\boldsymbol{w}})}\left[\|\bar{\boldsymbol{w}}(t_k)\|_2^2\right]$ by induction. Assume that there exists a sufficiently large $C_{\bar{\boldsymbol{w}}} > 0$ such that

$$\mathbb{E}_{p_{t_n}(\bar{\boldsymbol{w}})}\left[\|\bar{\boldsymbol{w}}(t_n)\|_2^2\right] \leq C_{\bar{\boldsymbol{w}}}, \quad \forall\, n \leq k. \tag{S247}$$

We next show that

$$\mathbb{E}_{p_{t_{k+1}}(\bar{\boldsymbol{w}})}\left[\|\bar{\boldsymbol{w}}(t_{k+1})\|_2^2\right] \leq C_{\bar{\boldsymbol{w}}}. \tag{S248}$$

Following the proof of [19, Lemma 6], we couple $\bar{w}^*$ optimally with $\bar{\boldsymbol{w}}(t) \sim p_t(\bar{\boldsymbol{w}})$, i.e., $(\bar{\boldsymbol{w}}(t), \bar{w}^*) \sim \gamma \in \Gamma_{opt}(p_t(\bar{\boldsymbol{w}}), p^*)$. We then obtain

$$\mathbb{E}_{\bar{\boldsymbol{w}}(t_{k+1})\sim p_{t_{k+1}}}\left[\|\bar{\boldsymbol{w}}(t_{k+1})\|_2^2\right] = \mathbb{E}_{(\bar{\boldsymbol{w}}(t_{k+1}),\bar{w}^*)\sim\gamma}\left[\|\bar{w}^* + \bar{\boldsymbol{w}}(t_{k+1}) - \bar{w}^*\|_2^2\right] \tag{S249}$$

$$\leq 2\mathbb{E}_{\bar{w}^*\sim p^*}\|\bar{w}^*\|^2 + 2\mathbb{E}_{(\bar{\boldsymbol{w}}(t_{k+1}),\bar{w}^*)\sim\gamma}\|\bar{\boldsymbol{w}}(t_{k+1}) - \bar{w}^*\|^2 \tag{S250}$$

$$\leq 2c_1 + 2\mathcal{W}_2^2(p_{t_{k+1}}(\bar{\boldsymbol{w}}), p^*) \tag{S251}$$

$$\leq 2c_1 + \frac{4}{\rho_U}F(p_{t_{k+1}}(\bar{\boldsymbol{w}})), \tag{S252}$$

where $\mathcal{W}_2(\cdot, \cdot)$ denotes the Wasserstein metric between two distributions and the last inequality holds due to [5, Theorem 1].

From our analysis in (S153), we have

$$F(p_{t_{k+1}}(\bar{\boldsymbol{w}})) \leq \exp\left(-\rho_U\alpha_k\right)F(p_{t_k}(\bar{\boldsymbol{w}})) + \Big(2\alpha_k^3\bar{L}^4\mathbb{E}_{p_{t_k}(\bar{\boldsymbol{w}})}\|\bar{\boldsymbol{w}}(t_k)\|_2^2$$

$$+ (2\alpha_k^3\bar{L}^2L^2 + L^2\alpha_k)\mathbb{E}_{p(\tilde{\omega}(t_k))}\|\tilde{\omega}(t_k)\|_2^2 + 2\alpha_k^2\bar{L}^2 d_w\Big) \tag{S253}$$

$$\leq \exp\left(-\rho_U\alpha_k\right)F(p_{t_k}(\bar{\boldsymbol{w}})) + 2\alpha_k^3\bar{L}^4 C_{\bar{\boldsymbol{w}}}$$

$$+ (2\alpha_k^3\bar{L}^2L^2 + L^2\alpha_k)\mathbb{E}_{p(\tilde{\omega}(t_k))}\|\tilde{\omega}(t_k)\|_2^2 + 2\alpha_k^2\bar{L}^2 d_w. \tag{S254}$$

Let

$$g_k = (2\alpha_k^3\bar{L}^2L^2 + L^2\alpha_k)\mathbb{E}_{p(\tilde{\omega}(t_k))}\|\tilde{\omega}(t_k)\|_2^2 + 2\alpha_k^2\bar{L}^2 d_w. \tag{S255}$$

We rewrite (S254) as

$$F(p_{t_{j+1}}(\bar{\boldsymbol{w}})) \leq \exp\left(-\rho_U\alpha_j\right)F(p_{t_j}(\bar{\boldsymbol{w}})) + g_j + 2\alpha_j^3\bar{L}^4 C_{\bar{\boldsymbol{w}}}, \quad \forall j \leq k, \tag{S256}$$

from which we further obtain

$$F(p_{t_{k+1}}(\bar{\boldsymbol{w}})) \leq \exp\left(-\rho_U\alpha_k\right)F(p_{t_k}(\bar{\boldsymbol{w}})) + g_k + 2\alpha_k^3\bar{L}^4 C_{\bar{\boldsymbol{w}}} \tag{S257}$$

$$\leq \exp\left(-\rho_U\alpha_k\right)\left(\exp\left(-\rho_U\alpha_{k-1}\right)F(p_{t_{k-1}}(\bar{\boldsymbol{w}})) + g_{k-1} + 2\alpha_{k-1}^3\bar{L}^4 C_{\bar{\boldsymbol{w}}}\right)$$

$$+ g_k + 2\alpha_k^3\bar{L}^4 C_{\bar{\boldsymbol{w}}} \tag{S258}$$

$$\leq \exp\left(-\rho_U(\alpha_k+\alpha_{k-1})\right)F(p_{t_{k-1}}(\bar{\boldsymbol{w}})) + g_{k-1} + g_k + 2(\alpha_{k-1}^3 + \alpha_k^3)\bar{L}^4 C_{\bar{\boldsymbol{w}}} \tag{S259}$$

$$\leq \exp\left(-\rho_U\sum_{i=0}^{k}\alpha_i\right)F(p_{t_0}) + \sum_{i=0}^{k}g_i + 2C_{\bar{\boldsymbol{w}}}n^4L^4\sum_{i=0}^{k}\alpha_i^3 \tag{S260}$$

$$\leq F(p_{t_0}) + \sum_{i=0}^{\infty}g_i + 2C_{\bar{\boldsymbol{w}}}n^4L^4\sum_{i=0}^{k}\alpha_i^3 \tag{S261}$$

$$\leq c_2 + \sum_{i=0}^{\infty} g_i + 2C_{\bar{\boldsymbol{w}}} n^4 L^4 \sum_{i=0}^{k} \alpha_i^3. \tag{S262}$$

Here we used the relation $\bar{L} \leq nL$. Since $\alpha_k^3 = \frac{a^3}{(k+1)^{3\delta_2}}$ and $2\delta_2 > 1$, it follows from (S14) that

$$\sum_{i=0}^{k} \alpha_i^3 \leq a^3 + \int_0^{\infty} \frac{a^3}{(t+1)^{3\delta_2}} \, dt = a^3 + \frac{a^3}{3\delta_2 - 1} = \frac{3\delta_2}{3\delta_2 - 1} a^3. \tag{S263}$$

From (S255), we note that $g_k \sim O\left(\frac{1}{k^{2\delta_2 - 2\delta_1}}\right)$ because $\mathbb{E}_{p(\tilde{\omega}(t_k))}\left[\|\tilde{\omega}(t_k)\|_2^2\right] \sim O\left(\frac{1}{k^{\delta_2 - 2\delta_1}}\right)$ from (S96). Since $2\delta_2 - 2\delta_1 > 1$, $g_k$ is a summable sequence, that is, there exists a $d_1 > 0$ such that

$$\sum_{i=0}^{\infty} g_i \leq d_1. \tag{S264}$$

We obtain from (S262), (S263) and (S264) that

$$F(p_{t_{k+1}}) \leq c_2 + d_1 + \frac{6\delta_2 a^3 n^4 L^4}{3\delta_2 - 1} C_{\bar{\boldsymbol{w}}} \tag{S265}$$

which together with (S252) leads to

$$\mathbb{E}_{\bar{\boldsymbol{w}}(t_{k+1}) \sim p_{t_{k+1}}}\left[\|\bar{\boldsymbol{w}}(t_{k+1})\|_2^2\right] \leq 2c_1 + \frac{4}{\rho_U} c_2 + \frac{24 n^4 L^4 \delta_2 a^3}{\rho_U (3\delta_2 - 1)} C_{\bar{\boldsymbol{w}}} + \frac{4}{\rho_U} d_1. \tag{S266}$$

Since $a$ is chosen such that

$$\frac{24 n^4 L^4 \delta_2 a^3}{\rho_U (3\delta_2 - 1)} < 1, \tag{S267}$$

it follows that

$$\mathbb{E}_{\bar{\boldsymbol{w}}(t_{k+1}) \sim p_{t_{k+1}}}\left[\|\bar{\boldsymbol{w}}(t_{k+1})\|_2^2\right] \leq 2c_1 + \frac{4}{\rho_U} c_2 + \frac{24 n^4 L^4 \delta_2 a^3}{\rho_U (3\delta_2 - 1)} C_{\bar{\boldsymbol{w}}} + \frac{4}{\rho_U} d_1 \leq C_{\bar{\boldsymbol{w}}} \tag{S268}$$

for any $C_{\bar{\boldsymbol{w}}}$ such that

$$C_{\bar{\boldsymbol{w}}} \geq \frac{2c_1 + \frac{4}{\rho_U}(c_2 + d_1)}{1 - \frac{24 n^4 L^4 \delta_2 a^3}{\rho_U (3\delta_2 - 1)}}. \tag{S269}$$

One choice of $C_{\bar{\boldsymbol{w}}}$ is

$$C_{\bar{\boldsymbol{w}}} = \max\left\{\mathbb{E}_{\bar{\boldsymbol{w}}(t_0) \sim p_{t_0}}\left[\|\bar{\boldsymbol{w}}(t_0)\|_2^2\right], \frac{2c_1 + \frac{4}{\rho_U}(c_2 + d_1)}{1 - \frac{24 n^4 L^4 \delta_2 a^3}{\rho_U (3\delta_2 - 1)}}\right\}. \tag{S270}$$

∎

## S5  Proof of Corollary 1

From (S223) we have

$$F(p_{t_{k+1}}(\bar{\boldsymbol{w}})) \leq \left(F(p_{t_0}(\bar{\boldsymbol{w}})) + \bar{C}_{F_1}\right) \exp\left(-\rho_U \sum_{\ell=0}^{k} \alpha_\ell\right)$$

$$+ \frac{1}{n^{\gamma - 2}} \frac{\bar{C}_{F_2}}{(k+1)^{\delta_2 - 2\delta_1}} + \frac{\bar{C}_{F_3}}{\exp\left(\frac{\rho_U a}{1 - \delta_2}(k+1)^{1 - \delta_2}\right)} \tag{S271}$$

When $\delta_2 \in (0.5, \ 1)$, from Lemma S3 we have

$$\sum_{\ell=0}^{k} \alpha_\ell \geq \int_0^k \frac{a}{(x+1)^{\delta_2}} \, dx = \frac{a(k+1)^{1-\delta_2}}{1 - \delta_2} - \frac{a}{1 - \delta_2} \tag{S272}$$

Thus

$$\exp\left(-\rho_U \sum_{\ell=0}^{k} \alpha_\ell\right) \le \exp\left(\frac{a\rho_U}{1-\delta_2}\right) \exp\left(-\frac{a\rho_U}{1-\delta_2}(k+1)^{1-\delta_2}\right) \tag{S273}$$

and (S271) can be written as

$$\begin{aligned}
F(p_{t_{k+1}}(\bar{\boldsymbol{w}})) \le \ &\left(F(p_{t_0}(\bar{\boldsymbol{w}})) + \bar{C}_{F_1}\right) \exp\left(\frac{a\rho_U}{1-\delta_2}\right) \exp\left(-\frac{a\rho_U}{1-\delta_2}(k+1)^{1-\delta_2}\right) \\
&+ \frac{1}{n^{\gamma-2}} \frac{\bar{C}_{F_2}}{(k+1)^{\delta_2-2\delta_1}} + \bar{C}_{F_3} \exp\left(-\frac{a\rho_U}{1-\delta_2}(k+1)^{1-\delta_2}\right)
\end{aligned} \tag{S274}$$

Now define constants $Q_1$ and $Q_2$ as

$$Q_1 = \left(F(p_{t_0}(\bar{\boldsymbol{w}})) + \bar{C}_{F_1}\right) \exp\left(\frac{a\rho_U}{1-\delta_2}\right) + \bar{C}_{F_3} \tag{S275}$$

$$Q_2 = \frac{\bar{C}_{F_2}}{n^{\gamma-2}}. \tag{S276}$$

Thus we have

$$F(p_{t_{k+1}}(\bar{\boldsymbol{w}})) \le Q_1 \exp\left(-\frac{a\rho_U}{1-\delta_2}(k+1)^{1-\delta_2}\right) + \frac{Q_2}{(k+1)^{\delta_2-2\delta_1}} \tag{S277}$$

We would like to find a $k$ such that

$$F(p_{t_{k+1}}(\bar{\boldsymbol{w}})) \le \epsilon, \tag{S278}$$

which is satisfied if

$$Q_1 \exp\left(-\frac{a\rho_U}{1-\delta_2}(k+1)^{1-\delta_2}\right) \le \frac{\epsilon}{2} \tag{S279}$$

and

$$Q_2 \frac{1}{(k+1)^{\delta_2-2\delta_1}} \le \frac{\epsilon}{2}. \tag{S280}$$

From (S279) we have

$$k \ge \left(\frac{1-\delta_2}{a\rho_U} \log\left(\frac{2Q_1}{\epsilon}\right)\right)^{\frac{1}{1-\delta_2}} \tag{S281}$$

and from (S280) we have

$$k \ge \left(\frac{2Q_2}{\epsilon}\right)^{\frac{1}{\delta_2-2\delta_1}} \tag{S282}$$

Therefore, for all $k \ge k^*$, we have

$$F(p_{t_k}(\bar{\boldsymbol{w}})) \le \epsilon, \tag{S283}$$

where

$$k^* = \max\left\{\left(\frac{1-\delta_2}{a\rho_U} \log\left(\frac{2Q_1}{\epsilon}\right)\right)^{\frac{1}{1-\delta_2}}, \left(\frac{2Q_2}{\epsilon}\right)^{\frac{1}{\delta_2-2\delta_1}}\right\} \tag{S284}$$

This concludes the proof of Corollary 1.

∎

Table S1: Test accuracy (%) for different approaches after 10 epochs

|  | SGD | C-ULA | Agent 1 | Agent 2 | Agent 3 | Agent 4 | Agent 5 |
|---|---|---|---|---|---|---|---|
| MNIST | 98.15 | 98.16 | 98.52 | 98.52 | 98.39 | 98.45 | 98.47 |
| SVHN | 7.648 | 8.9313 | 14.897 | 13.44 | 15.346 | 13.506 | 15.934 |

## S6 Numerical Experiments

### S6.1 Parameter estimation for Gaussian mixture

For the centralized setting, 100 data samples are drawn from the mixture of Gaussians in Section 5.1. For D-ULA, these 100 samples were randomly divided into 5 data sets of 20 samples, one for each of the five agents in the network. Both C-ULA and D-ULA are run for 1000000 epochs using their respective batch gradients. Step-size $\alpha_k = \alpha_0/(b_1 + k)^{\delta_2}$ is varied from 0.01 to 0.0001 similar to [26] with consensus step-size, $\beta_k = \beta_0/(b_2 + k)^{\delta_1}$ in the interval [0.36, 0.24]. Figure S1 shows estimated posteriors from C-ULA and the proposed approach. Posteriors estimated by the D-ULA replicate the true posterior with samples from both the modes.

### S6.2 Bayesian logistic regression

Expressions for time-varying step-size $\alpha_k$ and $\beta_k$ are same as in Section S6.1 with $\alpha_0 = 0.004$, $b_1 = 230$, $\delta_2 = 0.55$ for C-ULA and $\alpha_0 = 0.00082$, $b_1 = 230$, $\delta_2 = 0.55$, $\beta_0 = 0.48$ $b_2 = 230$, $\delta_1 = 0.05$ for D-ULA. Data is processed in batches of 10 for both approaches with 10 epochs through the whole data set for 50 runs. Accuracy at each iteration averaged over 50 runs for C-ULA and the 5 agents in D-ULA is shown in Figure S2. The shaded region of the figure indicates 1 standard deviation. Zoomed version of the accuracy with centralized ULA shown in Figure S6.2.c indicates a faster convergence of D-ULA to 84.38 % in 1040 iterations when compared to C-ULA which converges to the final accuracy of 83.89 %.

### S6.3 Decentralized Bayesian learning for handwritten digit recognition

Variations in step-size, $\alpha_k$ and $\beta_k$ are similar to Section S6.1 with $\alpha_0 = 0.00024$, $b_1 = 230$, $\delta_2 = 0.55$ for stochastic gradient descent (SGD), $\alpha_0 = 0.00034$, $b_1 = 230$, $\delta_2 = 0.55$ for C-ULA, and $\alpha_0 = 0.00032$, $b_1 = 230$, $\delta_2 = 0.55$, $\beta_0 = 0.48$, $b_2 = 230$, $\delta_1 = 0.05$ for D-ULA. Data sets for training are processed in batches of 1024, 1024, and 256 images for SGD, C-ULA, and D-ULA, respectively. Tables S1 summarize MNIST and SVHN test accuracy after 10 epochs for SGD, C-ULA, and D-ULA. Figure S3 shows prediction probability density for MNIST and SVHN data sets using all the approaches considered. For SGD, prediction probability corresponding to all the class labels (0-9) are obtained for each test case and the maximum value evaluated across the class labels corresponds to the predicted probability for each individual test case. Density on the $y$ axis represents the normalized count of the predicted probabilities so that the cumulative density over the probability of predicted labels integrates to one. For C-ULA and D-ULA, prediction probability is the mean of samples over epochs after burn-in period and its maximum value over all the 10 class labels represents the predicted probability for each test case. We illustrate the performance of SGD and Bayesian methods, C-ULA and D-ULA for hand-written digit recognition using confusion matrices based on the true and predicted labels evaluated with MNIST and SVHN test samples. Figure S4 shows heat maps corresponding to the confusion matrices generated with predicted and actual labels from MNIST test samples for both SGD and C-ULA. Heat maps for D-ULA resembles same C-ULA and the plots are not included to avoid redundancy. Though, both the approaches indicate a high level of prediction accuracy across the test samples, confidence scores of the predictions obtained by the Bayesian approach indicate reliability of predictions. Figure S5 show average prediction probability scores for each MNIST labels. This is relevant in particular for OOD sample detection as shown in Figures S6 and S7. Figure S6 show predicted label across SVHN test sets wherein the prediction accuracy is fairly low as the test samples are out of the distribution. Approaches such as SGD provides a single prediction score along with predicted labels, where as the Bayesian approaches provide mean and standard deviation of the predictions as well. Expected values of prediction probabilities averaged across each labels are shown in Figure S7. Such distributions indicate reliability of the predicted scores and helps to detect OOD samples.

Next, the predicted labels and corresponding scores are shown for three test samples selected from MNIST and SVHN data sets. Here, we select one sample each from a high, medium, and low

(a) True posterior distribution

(b) Centralized setting

(c) Agent 1

(d) Agent 2

(e) Agent 3

(f) Agent 4

(g) Agent 5

Figure S1: True and estimated posteriors

(a) Centralized ULA

(b) Agent 1

(c) Agent 2

(d) Agent 3

(e) Agent 4

(f) Agent 5

Figure S2: Accuracy on test sets averaged over 50 runs

Table S2: Predicted labels and predicted scores for images with different confidence levels

| | True label | Confidence | | C-ULA | Agent 1 | Agent 2 | Agent 3 | Agent 4 | Agent 5 |
|---|---|---|---|---|---|---|---|---|---|
| MNIST | 4 | High | Predicted label | 4 | 4 | 4 | 4 | 4 | 4 |
| | | | Mean | 0.999 | 0.99970 | 0.9997 | 0.9997 | 0.9997 | 0.99971 |
| | | | Std. dev. | 0.0003 | 0.0002 | 0.0002 | 0.0002 | 0.0002 | 0.0002 |
| | 7 | Medium | Predicted label | 1 | 7 | 7 | 7 | 7 | 7 |
| | | | Mean | 0.617 | 0.8974 | 0.8912 | 0.8875 | 0.8994 | 0.8843 |
| | | | Std. dev. | 0.142 | 0.0730 | 0.0622 | 0.0737 | 0.0561 | 0.0682 |
| | 8 | Low | Predicted label | 7 | 7 | 7 | 7 | 7 | 7 |
| | | | Mean | 0.349 | 0.4612 | 0.4512 | 0.4157 | 0.4574 | 0.4853 |
| | | | Std. dev. | 0.124 | 0.1852 | 0.1896 | 0.1806 | 0.1835 | 0.1855 |
| SVHN | 8 | High | Predicted label | 8 | 8 | 8 | 8 | 8 | 8 |
| | | | Mean | 0.993 | 0.995 | 0.995 | 0.995 | 0.995 | 0.995 |
| | | | Std. dev. | 0.006 | 0.005 | 0.005 | 0.005 | 0.005 | 0.005 |
| | 5 | Medium | Predicted label | 8 | 8 | 8 | 8 | 8 | 8 |
| | | | Mean | 0.407 | 0.6024 | 0.6144 | 0.6033 | 0.6237 | 0.5892 |
| | | | Std. dev. | 0.143 | 0.1363 | 0.1451 | 0.1476 | 0.1289 | 0.1411 |
| | 4 | Low | Predicted label | 9 | 9 | 9 | 9 | 9 | 9 |
| | | | Mean | 0.369 | 0.4007 | 0.3835 | 0.3574 | 0.4034 | 0.4151 |
| | | | Std. dev. | 0.124 | 0.1824 | 0.1772 | 0.164 | 0.1768 | 0.1836 |

confidence cases. Table S2 summarizes predictions, mean, and standard deviation of the predicted scores for both C-ULA and D-ULA.

(a) SGD

(b) Centralized ULA

(c) Agent 1

(d) Agent 2

(e) Agent 3

(f) Agent 4

(g) Agent 5

Figure S3: Probability of predicted labels

(a) SGD

(b) Centralized ULA

Figure S4: Actual and predicted MNIST labels across test data for SGD and C-ULA

Figure S5: Probability heat map across MNIST test labels for C-ULA

(a) SGD

(b) Centralized ULA

Figure S6: Actual and Predicted SVHN labels across test data for C-ULA

Figure S7: Probability heat map across SVHN test labels for C-ULA