[Reviews · NeurIPS 2020]

Review 1

Summary and Contributions: The paper proposes and analyzes a decentralized Langevin algorithm to perform bayesian inference in a setting in which independent data sets are distributed among a network of n agents. The communication topology between the different agents is an undirected connected graph. The authors analyse the decentralized ULA which incorporates an average consensus process on the top of local ULA. The authors use time-varying stepsizes and two different time scales for the consensus and the local ULA. The authors show that the averaged sample converges to the true posterior asymptotically. PS: I was already convinced by the interest of the paper. I did not change my appreciation of the work.

Strengths: The problem of bayesian parameter inference in the distribueted setting has been addressed by several authors. The strength of the approach which is presented here stems from the fact that it is both simple (it is a bayesian version of the ULA algorithm) and theoretically justified. The use of decreasing stepsize allows to remove the bias. The proposed theoretical bounds are easy to interpret: one part is dominated by the consensus error while the other is dominated by the additive noise. The proposed algoritm offers improved accuracy and enhanced speed of convergence.

Weaknesses: You can blame the paper for being overly academic. But this is a first result and I find it quite normal that the authors focused on a "very well posed" problem. One can immediately imagine many possible open problems: introducing constraints on communication between agents, intermittent communication or even asynchrony among the agents. It would have been interesting in the application section to present a more realistic example closer to the type of problems the authors propose to deal with. We still have the impression that applications are still quite "remote" and toyish.

Correctness: It would be pretentious to pretend I read the proofs. The authors use a fairly standard methods for the analysis of ULA. There are, of course, many technical difficulties, but I haven't had time to study all the technical derivations. The results seem quite credible. The simulations are very unremarkable, but the results of Bayesian approaches are quite difficult to illustrate anyway. I'm not sure that the Bayesian estimator gives results that are significantly different from a MAP estimator in the examples that have been given? But that is not the focus of this work.

Clarity: The paper is very well written and is easy to read. The approach is not complicated because it is a combination of a consensus algorithm and Langevin dynamics. An "informed" reader will easily follow the reasoning and understand the essence of the results.

Relation to Prior Work: There is an extensive bibliography on Langevin algorithms and their analysis. There is comparatively less about the Monte Carlo methods distributed. It's the part of the literature I know the least. Isn't there a competing algorithm that it would be interesting to compare with?

Reproducibility: Yes

Additional Feedback:


Review 2

Summary and Contributions: The authors propose a decentralized approach to doing Langevin dynamics, with a novel formulation of the iterate sequence and convergence discussion using log-Sobolev inequality. Experimental results on synthetic and 'real-world' datasets show that the method is capable of learning accurate distribution for the former and providing usable point estimates for the latter.

Strengths: 1. The authors propose a novel iterate sequence (eqn (8)) in the distributed setting, and provide a detailed theoretical discussion of the workings of the algorithm. Convergence proof using the Sobolev inequality in the supplementary is pretty detailed as well, though I did not get a chance to verify it in detail. 2. The authors provide experimental results on both synthetic and real datasets; they demonstrate that the method learns accurate distributions for the mixture of Gaussians used in the former, and achieves good test accuracy for MNIST/SVHN datasets in the latter as well. Overall I think the main contribution of this work would be the iterate formulation in the distributed setting and the convergence analysis.

Weaknesses: 1. I think the distributed setting and all the subtleties that come with it should have been explored better. 2. For example, eqn (8) seems to use a collation of the w_{j} for all the nodes to calculate the next iterate at each. I was left wondering about the communication costs of an architecture needed for this to work, and potential issues with that. One obvious question would be, how would the iterate updates at each node be impacted by random noise injected into the w_{js} being passed around in the comms channels and/or random drops / missed updates. 3.The rationale behind weighting the signed difference between the w_{j}s by the communication adjacency matrix entries for use in eqn (8) was not fully clear to me; the authors could explain the motivation behind this better in the main text. 4. Eqn (2) is a bit dubious, as I don't think the conditional independence assumptions allow that particular factorization. 5. The only difference between the convergence discussion in \S4.1 and other works in the literature that use similar machinery seems to be the formulations for the extra constants/iterate weights in the distributed setting. This reduces the novelty/significance of that section somewhat, in my opinion. 6. While the authors' inclusion of both synthetic and MNIST results is commendable, the authors could describe some of their choices better; examples would be cost constraints behind using only five agents 100samples in the synthetic case, runtime plots comparing diff # agents with decay of suitable energy function for the synthetic case and test accuracy for the real datasets; discussions of overall communication cost/complexity would be useful too. The authors could also find better ways to compare the baseline centralized Langevin method with theirs; Figures 1 and 2 seem to be somewhat inefficient use of space. 7. On a similar vein, the test accuracies reported for MNIST are not that difficult to achieve with other methods, so the authors should discuss the advantages of the proposed method for this task. 8. The statement on line 258 (SGD being a MAP estimate) should be qualified better.

Correctness: I did not get a chance to verify the supplementary proofs in detail; regarding the discussions in the main text, I have some questions about the motivation behind the author's distributed iterate formulation in eqn(8) and the ideas behind eqn(2) that are mentioned above.

Clarity: The writing could use some improvement. 1. As mentioned earlier, the ideas/motivations behind some of the formulations could be described better. 2. The writing on lines 6 -- 8, 24--25, 69--71 could be improved. 3. The content on lines 34 -- 65 could be shortened and/or moved to a 'Related Work' section, as it's not directly related to the central contribution. 4. The writing on lines 140--149 could also be improved. For instance, I'm not sure if ignoring the noise term (as mentioned on line 140) is meaningful in this context.

Relation to Prior Work: Other than the concerns above, I think the connections to related work are sufficiently detailed.

Reproducibility: Yes

Additional Feedback: Please see the sections above. ===== Post rebuttal+discussion: The author's response addresses couple of my questions, but the rest remain unanswered. Based on their promise of including additional experimental results and clarifications in the final version, I'm changing my score to weak accept.


Review 3

Summary and Contributions: This paper proposes a distributed version of the Unadjusted Langevin Algorithm, in which the log density is smooth but potentially non-logconcave, and has a finite sum structure, each term being only available to one specific agent, which can communicate according to some arbitrary communication graph. The authors design a new algorithm inspired the consensus algorithm in optimization, and shows that the mean of iterates over the nodes converges to the target distribution in KL divergence.

Strengths: The problem of distributed sampling is interesting and important to the ML community, both theoretically and practically. The claims are clear and empirically demonstrated. Although the proposed algorithm is a natural combination of existing algorithms (Langevin Dynamics and consensus optimisation), its analysis appears to be non-trivial.

Weaknesses: The empirical setup for comparison to centralised-ULA is a bit imprecise, especially when it comes to step size choice.

Correctness: Claims and methodology are correct.

Clarity: The paper is well written

Relation to Prior Work: Relation to the Langevin literature is clearly discussed. It would be good to also mention the related distributed optimisation literature, e.g., is the obtained speed-up similar as the one obtained in optimisation?

Reproducibility: Yes

Additional Feedback: In your theoretical analysis, it would be good to explicitly track the dependence on the dimension of the problem. Does it appear in all constants C_{F_i}, i=1,2,3 ? I am not an expert in distributed optimisation, but is Assumption 3 standard in the literature? Numerical experiments: How did you choose the step-sizes for the different algorithms. Since this can largely affect the algorithms performance, the tuning procedure for each algorithm should be clearly stated in order to claim empirical outperformance. Parameter estimation for Gaussian mixture: It's not so clear to me that D-ULA outperforms C-ULA for this task, since the modes of the D-ULA samples do not seem to really be aligned with those of the true posterior. It would be more striking if you could compute some discrepancy measure, e.g., Wasserstein distance or TV distance using some binning, between the sample and true posterior, which should be doable since the problem is only in 2 dimensions. However, the fact that D-ULA is able to better distinguish the two modes is quite interesting and promising for applications to non-convex optimisation. Can you give some intuition about why this occurs? What initialisation do you use? If the initial distribution evenly covers both modes, I'm a bit surprised that C-ULA dos not distinguish them well. It would also be interesting to observe the improvement when the number of workers increases (as predicted by the theory). Due to these concerns, I am not fully convinced by the claim that D-ULA achieves "considerable improvement in the convergence rate" compared to C-ULA since this claim is not supported by the theory. Minor: l.212: "minimum number of iterations required" Is it really required ? Prefer statement like "number of iterations after which we guarantee that". Typo: l.170 admit ========== Post rebuttal: The author's response addressed some of my questions, but the other remained unanswered. I will therefore keep my score to "weak accept".


Review 4

Summary and Contributions: Decentralized and embarrassingly parallel methods have been a core of optimization. This paper is in a long line of research that tries to bridge the gap between frequentist optimization and Bayesian methods. In particular this methods proposes to combine the idea of local posteriors (often used in embarrassingly parallel methods) with Langevin Dynamics. They not only give a clear description of the algorithm, but also provide theoritical analysis of their method. Moreover they show some experimental results backing their theoretical analysis

Strengths: ->The method looks sound -> Paper is well written -> Fair theoretical analysis

Weaknesses: -> The experimental section is lacking. Real world experiments are only with 5 agents, which I think is not sufficient.

Correctness: I did not find an error with the claims in the paper

Clarity: The paper is well written

Relation to Prior Work: There are some prior work missing. The authors should add other work in the area that have combined advances in the optimization to Langevin dynamics. Examples include variance reduction methods and second order methods for Langevin dynamics.

Reproducibility: Yes

Additional Feedback: I am on the fence for this paper, please let me know if I missed something in my summarization *********** post author response *********** please ensure that the experiments promised in the rebuttal are added to the next version of the paper.

[Author Response · NeurIPS 2020]

We propose a decentralized Bayesian learning algorithm when the data set $\mathbf{X}$ is held disjointly over $n$ agents, i.e., $\mathbf{X} = \bigcup_{i=1}^{n} \boldsymbol{X}_i$ with $\boldsymbol{X}_i \bigcap \boldsymbol{X}_j = \emptyset$ for $j \neq i$. Thus the posterior satisfies $p(\boldsymbol{w}|\mathbf{X}) = \prod_{i=1}^{n} p(\boldsymbol{w}|\boldsymbol{X}_i)$. Similar formulations can be seen in almost all embarrassingly parallel MCMC algorithms (see [35,37,38] in main paper). We will add further discussions/references on various parallel MCMC schemes[1–4] in a Related Work section. However, they do not apply to the decentralized setting since they require a central node to combine the samples from individual Markov chains. In comparison, our formulation does not require a central node: each computing node $i$ reconstructs an approximate posterior from $\boldsymbol{X}_i$ and prior information (the 3rd term on the r.h.s of (8)) while interacting with their neighbors as dictated by the undirected communication graph $\mathcal{G}(\mathcal{V}, \mathcal{E})$ (the 2nd term on the r.h.s of (8), where $a_{i,j} = 1$ if the $i$-th node can receive $\boldsymbol{w}_j$ from $j$-th node and zero otherwise). We will clarify this point and mention similar techniques in consensus-optimization (e.g., decentralized SGD[6–9]). Though our proposed algorithm is built on ULA, analysis of even the centralized ULA (C-ULA) for non-log-concave target distributions requires restricting assumptions (see lines 51-60 & discussion on [13-20]). We will add discussions on variance reduced SGLD and second-order (underdamped) Langevin algorithms (e.g.,[10–13]). Here we relax aforementioned assumptions and our Assumption 3 is weaker than the uniform bound on gradient disagreement[7] and the bounded gradient assumption[6;9].

To the best of our knowledge, we propose the first-ever decentralized ULA (D-ULA) for general non-log-concave target distributions with time-varying step-sizes. The **3** main advantages of the proposed D-ULA include: **1)** it enables the individual computing nodes to approximate the posterior with an accuracy comparable to that of the C-ULA. **2)** by using decaying step-sizes, we are able to remove the constant bias term present in the KL-divergence and show that the rate of convergence is $\mathcal{O}\left((n^{1/3}(k+1)^{\delta_2 - 2\delta_1})^{-1}\right)$ (see (27)). In the final version we will discus how the convergence rate and the constants $C_{F_i}$ depends polynomially in problem dimension $d_w$. **3)** similar to D-SGD[6–9], D-ULA experiences speedup with the number of agents as shown by the $n^{1/3}$ in the denominator of (27). These advantages will be highlighted in the final version. Our analysis of D-ULA is **novel/non-trivial** compared to the existing non-convex consensus-optimization and non-log-concave ULA literature because: **(i)** Consensus analysis and results in Theorem 1 are novel since we use time-varying step-sizes $\alpha_k$ and $\beta_k$ and provide an explicit consensus rate in term of step-size decay rates (see (25)) (not just bounded consensus as in[7;8]). **(ii)** Compared to existing C-ULA analysis for non-log-concave target distributions, the continuous-time approximation to the D-ULA contains an additional consensus error term $\zeta(\cdot)$ (see (21)) that complicates the analysis. Requirements on the time-varying step sizes are also not straightforward to obtain as the existing literature is focused on fixed step-sizes. We will emphasize the novelty of our analysis in the final version. D-ULA requires the same number of communication rounds as the computation iterations to achieve a prescribed level of accuracy (Corollary 1). We hope this paper provides foundations to many open research problems, such as relaxing the synchronous, periodic communication requirement of D-ULA through local computation, compression, quantization, event-triggered and asynchronous communication as done in the D-SGD[6;9], and extensions of the proposed algorithm to SGLD and noisy, time-varying communication channel.

The goal of Bayesian learning is to estimate the epistemic uncertainty for assessing confidence in the model, which is not possible with MAP or ML point estimates as illustrated using the OOD detection example (Section 5.3). Though the histogram of probability of predicted labels across all MNIST test samples using SGD shows similar trend to that of Bayesian estimates (Fig. S3, Table 2), MAP cannot quantify the uncertainty associated with the predictions. To further illustrate this point, we will include the mean and standard deviation of prediction scores for individual test samples from both (in-distribution) MNIST and (OOD) SVHN datasets using Bayesian estimates in the final version. Though D-ULA replicates the true posterior with similar fidelity as C-ULA, our analysis proves the faster convergence of D-ULA as shown in Fig 2. For the GMM experiment, algorithm parameters were selected so as to have same step-sizes for both the distributed and centralized approach. However additional experiments have shown that C-ULA can distinguish both modes with further tuning of hyper-parameters. Step-sizes for each algorithm are obtained using grid search through feasible hyperparameter space, which is derived from theoretical results (Section 4). In response to reviewers' comments, we will include the following results in the final version 1) Results of all the experiments with more number of agents 2) Plots of empirical training loss and accuracy (for classification example) versus epochs for D-ULA with varying number of agents 3) An approximate discrepancy measure based on Wasserstein distance using estimated modes of the posterior distribution for GMM[5]. However, for complicated and intractable posteriors in regression and classification applications, we rely on the metrics based on test accuracy and OOD detection.

[1] Wang and Dunson, *Parallelizing MCMC . . .*, arXiv:1312.4605, 2013. [2] Neiswanger et al., *Asymptotically exact, embarrassingly parallel MCMC*, UAI, 2014. [3] Wang et al., *Parallelizing MCMC with random partition trees*, NIPS, 2015 [4] Chowdhury and Jermaine, *Parallel and distributed MCMC via shepherding . . .*, AISTATS, 2018. [5] Givens and Shortt, *A class of Wasserstein metrics . . .*, The Michigan Mathematical Journal, 1984. [6] Signh et. al., *SPARQ-SGD . . .* arXiv:1910.14280, 2019. [7] Lian et al., *Can decentralized algorithms outperform . . .*, NeurIPS, 2017. [8] Tang et al., *$D^2$: Decentralized Training. . .*, ICML, 2018. [9] Koloskova et. al., *Decentralized Deep Learning . . .*, ICLR 2020. [10] Dubey et. al., *Variance Reduction in SGLD*, NIPS, 2016. [11] Chatterji et. al., *On the Theory of Variance Reduction . . .*, ICML, 2018. [12] Cheng et. al., *Underdamped Langevin MCMC. . .*, COLT, 2018. [13] Şimşekli et. al., *Fractional Underdamped Langevin. . .*, arXiv:2002.05685, 2020.


[Meta-Review · NeurIPS 2020]

The paper adresses the important problem of Bayesian inference in a distributed setting, via a decentralized Langevin algorithm. Although the method is a natural extension of existing algorithms, its simplicity is an advantage, and the theoretical analysis is nontrivial. After considering the author's response, all reviewers agreed that the paper will make a nice contribution to Neurips.